# Dysregulated naive B cells and de novo autoreactivity in severe COVID-19

Matthew C. Woodruff[1,2,11], Richard P. Ramonell[3,11], Natalie S. Haddad[4], Fabliha A. Anam[1,2], Mark E. Rudolph[5], Tiffany A. Walker[6], Alexander D. Truong[7], Aditeeya N. Dixit[7], Jenny E. Han[6], Monica Cabrera-Mora[7], Martin C. Runnstrom[7], Regina Bugrovsky[1,2], Jennifer Hom[1,2], Erin C. Connolly[8], Igor Albizua[9], Vidhi Javia[7], Kevin S. Cashman[1,2], Doan C. Nguyen[7], Shuya Kyu[7], Ankur Singh Saini[1,2], Michael Piazza[10], Christopher M. Tipton[1,2], Arezou Khosroshahi[1,2], Greg Gibson[8], Greg S. Martin[7], Cheryl L. Maier[9], Annette Esper[7], Scott A. Jenks[1,2], F. Eun-Hyung Lee[7]✉ & Ignacio Sanz[1,2]✉

Severe SARS-CoV-2 infection[1] has been associated with highly inflammatory immune activation since the earliest days of the COVID-19 pandemic[2–5]. More recently, these responses have been associated with the emergence of self-reactive antibodies with pathologic potential[6–10], although their origins and resolution have remained unclear[11]. Previously, we and others have identified extrafollicular B cell activation, a pathway associated with the formation of new autoreactive antibodies in chronic autoimmunity[12,13], as a dominant feature of severe and critical COVID-19 (refs. [14–18]). Here, using single-cell B cell repertoire analysis of patients with mild and severe disease, we identify the expansion of a naive-derived, low-mutation IgG1 population of antibody-secreting cells (ASCs) reflecting features of low selective pressure. These features correlate with progressive, broad, clinically relevant autoreactivity, particularly directed against nuclear antigens and carbamylated proteins, emerging 10–15 days after the onset of symptoms. Detailed analysis of the low-selection compartment shows a high frequency of clonotypes specific for both SARS-CoV-2 and autoantigens, including pathogenic autoantibodies against the glomerular basement membrane. We further identify the contraction of this pathway on recovery, re-establishment of tolerance standards and concomitant loss of acute-derived ASCs irrespective of antigen specificity. However, serological autoreactivity persists in a subset of patients with postacute sequelae, raising important questions as to the contribution of emerging autoreactivity to continuing symptomology on recovery. In summary, this study demonstrates the origins, breadth and resolution of autoreactivity in severe COVID-19, with implications for early intervention and the treatment of patients with post-COVID sequelae.

In 2019, the novel betacoronavirus SARS-CoV-2 emerged from Wuhan, China, resulting in the COVID-19 pandemic[1]. With reported mortality of around 2%, early characterizations of severe disease emphasized the proinflammatory cytokine IL-6 and invoked the possibility of cytokine storms[2,3]. These observations, alongside the observed efficacy of high-dose steroids in these patients, were highly suggestive of immune responses not only responsible for viral clearance but potentially contributing to disease pathology[4,5]. Profound alterations in the immune compartment were quickly identified as correlates of these inflammatory responses, with distinct patient immunotypes having increased frequencies of circulating plasmablasts yet lacking

evidence of T follicular help (Tfh)[19]. This was bolstered by the identification of collapsed germinal centre environments in patients that had succumbed to COVID-19 (ref. [14]).

Deep analysis of B cell activation pathways by our group and others has led to a strong emphasis on the extrafollicular (EF) pathway as a common feature of severe disease[14,15,17]. Characterized by the induction of T-bet-driven double-negative 2 (CD27⁻, IgD⁻, CD11c⁺, CD21⁻ (DN2)) B cells, expansion of CD19⁺ antibody-secreting cells (ASCs) and depression of mutation frequencies in the ASC repertoire, these responses are highly similar to those we had identified previously in patients with active severe systemic lupus erythematosus (SLE)[13,20]. In these patients,

[1]Department of Medicine, Division of Rheumatology, Lowance Center for Human Immunology, Emory University, Atlanta, GA, USA. [2]Emory Autoimmunity Center of Excellence, Emory University, Atlanta, GA, USA. [3]Department of Medicine, Division of Pulmonary, Allergy and Critical Care Medicine, University of Pittsburgh, Pittsburgh, PA, USA. [4]MicroB-plex, Atlanta, GA, USA. [5]Exagen Inc., Vista, CA, USA. [6]Department of Medicine, Division of General Internal Medicine, Emory University, Atlanta, GA, USA. [7]Department of Medicine, Division of Pulmonary, Allergy, Critical Care and Sleep Medicine, Emory University, Atlanta, GA, USA. [8]School of Biological Sciences, Georgia Institute of Technology, Atlanta, GA, USA. [9]Department of Pathology and Laboratory Medicine, Center for Transfusion and Cellular Therapies, Emory University School of Medicine, Atlanta, GA, USA. [10]Nicoya, Kitchener-Waterloo, Ontario, Canada. [11]These authors contributed equally: Matthew C. Woodruff and Richard P. Ramonell. ✉e-mail: F.E.Lee@emory.edu; Ignacio.sanz@emory.edu

EF response activation results in the de novo generation of naive-derived autoreactivities despite the presence of chronic preformed auto-immune memory, correlated with disease severity[12]. At the time of publication of our study, evidence of autoreactivity was mounting in severe disease, with observations of autoantibody-linked blood clotting[6], anti-interferon antibodies[7], connective-tissue-disease-associated interstitial lung disease[8] and generalized observations of clinical autoreactivity[9], including our findings of expanded *IGHV4-34* B cells[15,21]. These observations have been bolstered by the reporting of broad autoreactivity in these patients, frequently targeting critical immune components[10], with serological kinetics strongly suggesting the onset of new autoreactivity[11]. However, the developmental origins of these autoreactivities, their connection with the underlying de novo antiviral response and their ultimate resolution have remained unknown.

## Viral-specific ASCs in severe COVID-19

Previous work established robust expansion of the ASC compartment as a hallmark of severe COVID-19 (refs. [15,19]). Retrospective analysis of previously collected data from 25 (nine healthy donors (HD), seven outpatients (OUT-C) and nine intensive care unit (ICU) patients (ICU-C)) individuals showed that such expansion also includes the more mature CD19-negative ASC fraction that we first reported to contain the long-lived plasma cells in the human bone marrow and that has not been previously measured in COVID-19 infection or other acute immune responses in humans[22] (Extended Data Fig. 1a–c and Supplemental Tables 1 and 2). Consistent with previous findings, ASC expansion in the ICU-C cohort was directly correlated with expansion of DN2 B cells, an important intermediate in the naive-derived EF B cell response pathway (Extended Data Fig. 1a, d)[13,15].

Although ASC expansion correlates with increased serological IgG response to the SARS-CoV-2 spike protein receptor-binding domain (RBD) in patients with severe disease[15], the circulating ASC compartment's direct contribution to that response has not been assessed. Using a new in vitro method that optimizes overnight antibody secretion from peripheral blood mononuclear cell (PBMC)-purified ASCs into the culture supernatant (medium enriched in newly synthesized antibodies; MENSA[23]), we found that ICU-C patients had higher frequencies of blood ASCs secreting IgG RBD-specific antibodies, confirming the relevance of early circulating ASCs to the antiviral response as opposed to non-specific cellular expansion (Fig. 1a). Indeed, overall IgG-switched RBD-targeted MENSA titres were directly correlated with ASC expansion across the COVID-19+ cohorts (Fig. 1b).

## Low selective pressure in expanded ASCs

In SLE, naive-derived EF ASC expansions result in new autoreactive clones[12]. With considerable literature pointing to the presence of autoreactivity as a feature of severe COVID-19 (refs. [6,7,10]), it was important to understand the contribution of ASCs to both antiviral and autoantigen targeting. However, direct binding studies of these IgG+ cells are hampered by the propensity of the cells to downregulate surface B cell receptor (BCR), in contrast to their IgM+ counterparts (Fig. 1c). Thus, antigen-specific flow-based study of this population would incompletely assess the ASC contribution to the overall antigen-specific response, and broad analysis of this cellular compartment independent of BCR expression and antigen-specific probing is required.

To study the nature of the ASC compartment in these patients, six of ten recruited ICU patients without dexamethasone treatment, alongside four patients with mild disease and three demographically matched HD, were selected for single-cell VDJ repertoire analysis. More than 17,000 ASCs were sequenced at acute infection time points between 4 and 18 days after symptom onset, reflecting almost 9,000 independent ASC clonotypes across all individuals (Supplemental Table 3). Clonality of the library was consistent with previous

descriptions of oligoclonal ASC expansion[15], with up to 13% of clonotypes representing more than 3% of the total repertoire (Supplemental Table 3). Isotype analysis demonstrated a consistent expansion of IgG1 in the ICU-C cohort relative to the dominance of IgA found in steady-state HD in this study and previous publications[24] (Fig. 1d and Extended Data Fig. 2a, b). Concomitant IgM+ expansions in some patients, alongside clonal connectivity between IgM and IgG1 ASCs in the ICU-C group, indicated that the IgG1 compartment might reflect the newly minted Ag-specific ASC pool (Extended Data Fig. 2a, c). An intermediate phenotype was observed in the OUT-C group with IgG1 increases that did not reach statistical significance (Extended Data Fig. 2b). Emphasis of IgG1 clonotypes was consistent with enrichment of total serological IgG1 in the ICU-C cohort, and with retrospective analysis of published single-cell transcriptomics data collected from bronchoalveolar lavage fluid (BALF) of ten intubated patients, which identified substantial IgG1 expression in the plasmablast population (Extended Data Fig. 3)[25].

The expanded IgG1+ ASC compartment of ICU-C patients was distinguished by reduced mutation frequency relative to OUT-C and HD controls (Fig. 1e, f). Notably, mutation reduction was largely concentrated on the IgG1 compartment, with 10–70% of all IgG1 ASCs expressing VH germline sequences and overall mutation frequencies significantly decreased in comparison with the rest of the class-switched ASC compartment (Fig. 1e–g). Consistent with these observations, an analysis of the selective pressure on the antibody complementarity-determining regions, as determined by Bayesian estimates of antigen-driven selection (BASELINe)[26], demonstrated selective reduction of the IgG1 in the ICU-C cohort versus other class-switched compartments (Fig. 1h). In SLE, the increased incorporation of autoreactivity-prone IgHV4-34 clonotypes into the antigen-selected CD27+ B cell compartment is a bellwether of reduced selective pressure and is often a result of naive-derived EF B cell responses[12]. A similar phenomenon was reflected in the repertoire of the ICU-C cohort, with increased frequency of IGHV4-34+ cells emerging specifically in the IgG1+ ASC compartment (Fig. 1i), aligning with our previous observations of increased IGHV4-34 serology in these patients[15].

## Uncoupled ASC and memory compartments

To more deeply understand the origins and persistence of the low-mutation IgG1 ASC compartment, the contemporaneous CD27+ memory B cells were sorted and analysed in three surviving patients from the original ICU cohort (Supplementary Table 3). Consistent with the expected properties of established memory B cells, class-switched CD27+ cells were more polyclonal and had high levels of SHM[12] (Supplemental Table 3). In contrast to their matched ASC counterparts, IgG1-expressing memory clonotypes showed increased selective pressures and decreased frequency of IgG1 clonotypes expressing unmutated BCRs (Extended Data Fig. 4a–c).

Formal connectivity analysis of the IgG1 ASC compartment and contemporaneous memory in the ICU-C cohort showed low levels of clonal sharing in two of three patients and no significant differences compared with steady-state HD, who in the absence of known immune perturbation are presumed to be devoid of continuing memory activation (Extended Data Fig. 4d). Moreover, in the two patients that showed active connections between memory and ASC compartments, the connections were dominated by higher-mutation clonotypes (>1%) (Extended Data Fig. 4e). Indeed, across the dataset, only four low-mutation clonotypes were identified as shared between the emerging ASC and memory compartments. Overall, our findings indicate uncoupling and separate selection pressures between the IgG1 ASC and memory B cell repertoires (Extended Data Fig. 4a–c) and are consistent with the emergence during acute severe infection of a memory-independent, newly generated ASC compartment with reduced selective pressure.

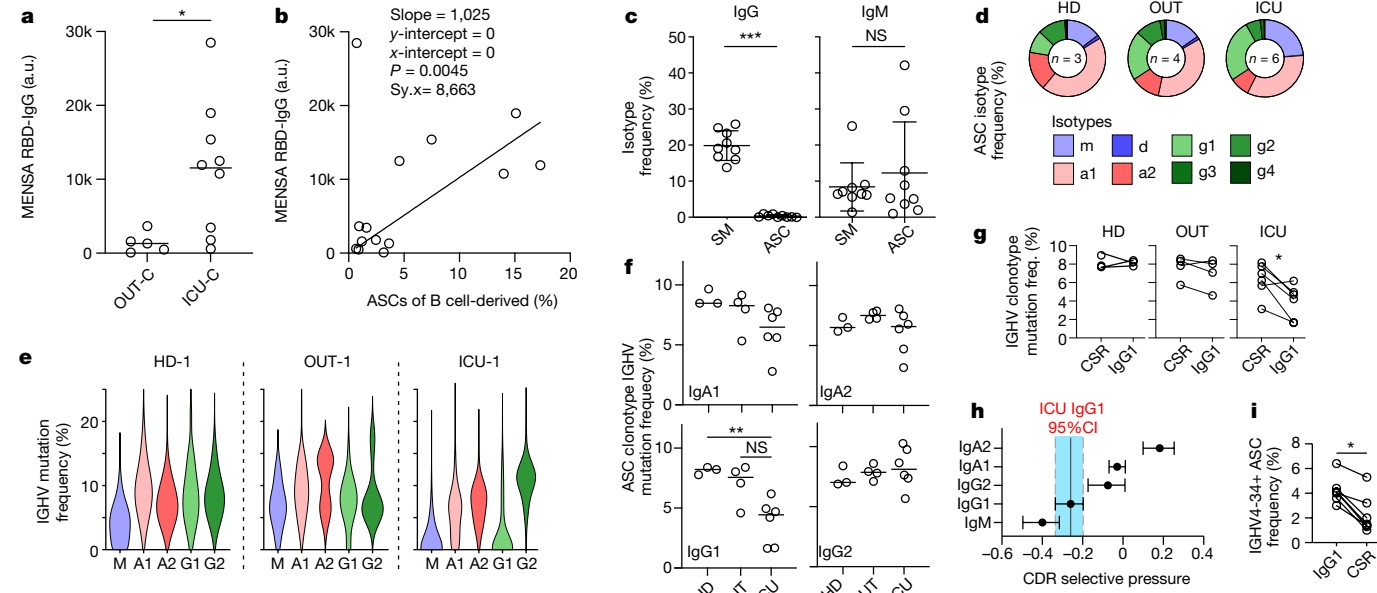

**Fig. 1 | Expansion of low-selection IgG1 ASC compartment is a hallmark of severe COVID-19. a,b**, MENSA samples from OUT-C (*n* = 7) or ICU-C (*n* = 9) patients were analysed for IgG reactivity against the SARS-CoV-2 RBD. RBD-specific IgG antibody in MENSA samples collected from OUT-C and ICU-C patients (**a**). Linear correlation of RBD-specific IgG antibody in MENSA samples versus ASC frequency of B cell-derived cells in OUT-C and ICU-C patients (**b**). IgG$^+$ and IgM$^+$ frequency of total switch memory (SM) or ASC populations from the ICU-C cohort (**c**). **d–i**, ASCs from the HD (*n* = 3), OUT-C (*n* = 4) and ICU-C (*n* = 6) cohorts were sorted for single B cell repertoire sequencing and subsequent analysis. Average ASC isotype compositions of HD, OUT-C and ICU-C individuals (**d**). Representative ASC mutation frequency distributions by isotype in HD-1, OUT-1 and ICU-1 individuals (**e**). *IGHV* gene nucleotide mutation frequencies of the indicated ASC isotypes in HD, OUT-C and ICU-C individuals (**f**). *IGHV* gene nucleotide mutation frequencies of IgG1 versus other class-switched ASCs from the indicated cohort (**g**). BASELINe selection analysis of CDR selection in ICU-C ASCs, grouped by isotype. Bars represent 95% confidence intervals (CI) in the group (**h**). IGHV4-34$^+$ ASC frequency in IgG1 versus other class-switched ASCs (**i**). In **a**, **c**, **g** and **i**, statistical significance was determined using two-tailed *t*-test between the indicated groups. In **g** and **i**, paired analyses were used. In **f**, statistical significance was determined using analysis of variance with Tukey's multiple-comparisons testing between all groups. In **a–i**, *\*P* ≤ 0.05; *\*\*P* ≤ 0.01; *\*\*\*P* ≤ 0.001. In **a**, **c**, and **f**, summary statistics are mean ± s.d. In **h**, summary statistics are mean ± 95% CI. a.u., arbitrary units; freq., frequency; NS, not significant.

## Clinical autoreactivity in COVID-19

The developing ASC response characteristics observed at both the cellular and repertoire levels were highly similar to previous observations in patients with active SLE[12,15]. To understand whether COVID-19 responses also correlated with autoreactivity, plasma collected from 27 ICU-C, 18 OUT-C, 20 SLE and 14 HD individuals was assessed through testing of more than 30 clinically relevant autoantigens by Exagen, Inc. and analysed for autoreactivity associated with connective tissue disorders. Broad tolerance breaks were identified across the ICU-C cohort against a variety of targets including rheumatoid factor (RF; 2/27), phospholipids (3/27), nuclear antigens (11/27) and glomerular basement membrane (GBM; 2/27) (Table 1). Most ICU patients had at least one positive test, with some patients testing positive for up to seven independent autoantigens (Fig. 2a). Higher 'densities' of autoreactivity were significantly increased in ICU-C individuals, with three or more autoreactivities being found exclusively in this cohort (Fig. 2a and Extended Data Fig. 5a).

Autoreactivity screening identified significant emergence of two autoreactivities: antinuclear antibodies (ANAs) and anticarbamylated protein responses (CarP) (Fig. 2b, c). Although ANAs have been well characterized in clinical autoimmunity, they can also be present in up to 15% of healthy subjects at immunofluorescence titres less than 1:80 (ref. [27]). By contrast, more than 40% of the ICU-C cohort showed ANA reactivities at titres greater than 1:160 (Table 1). Anti-CarP antibodies, which are associated with tissue damage in rheumatoid arthritis and SLE[28,29], were specific to the ICU-C cohort and present in more than 40% of patients (Fig. 2c and Table 1). Titres of a-CarP were directly correlated with the overall number of tolerance breaks across the cohort (Fig. 2d and Extended Data Fig. 5b). Despite similarities in B cell activation profiles, other canonical reactivities associated with SLE, including Sm/RNP, Ro, La and even double-stranded DNA (dsDNA), were universally negative (Table 1).

To understand specificity to COVID-19, another 28 plasma samples from ICU patients with acute respiratory distress syndrome (ARDS) as a result of confirmed bacterial pneumonia were assessed (Table 1). Notably, the autoreactivity profiles of these patients were highly similar to those of patients with critical COVID-19, strongly suggesting that the autoimmune phenomena described in COVID-19 so far may be generalizable to other severe pulmonary infections (Fig. 2e and Table 1). Identification of similar self-targets, particularly anti-CarP and anti-GBM titres, indicates that clinical tests available at present could be used to identify these phenomena in real time across a host of human infectious diseases (Table 1).

To validate the ICU cohort, a retrospective study of 52 independent critically ill patients who had received autoantibody testing as part of routine clinical care at the discretion of their treating physicians was undertaken. More than 50% of patients had at least one positive test, with ANAs as the most common autoreactive feature (a-CarP antibody testing was not performed) (Extended Data Fig. 6). Among ICU patients, disease severity was correlated with tolerance breaks—patients with the highest levels of C-reactive protein (CRP; a surrogate of disease severity in COVID-19 (ref. [30])) had both increased numbers and increased intensities of autoreactive tests (Fig. 2f–h). Although longitudinal testing for this cohort was limited, seven patients were tested 2 weeks after the initial draw, with three of seven testing positive for ANAs on initial assessment (Fig. 2i). In alignment with published work describing building serological autoreactivity in immune-targeted autoantibodies[11], all three patients showed stable or increasing ANA titres despite decreased CRP, suggesting building autoreactivity profiles beyond the resolution of biomarkers of clinical illness. Combining the datasets and

**Table 1 | Summary of positive autoreactive tests**

| Autoreactive Target | HD (*n*=14) | OUT (*n*=18) | ICU (*n*=27) | ICU-PASC (*n*=40) | ARDS (*n*=29) | SLE (*n*=20) |
|---|---|---|---|---|---|---|
| dsDNA | 0 | 0 | 0 | 1 (3%) | | 18 (90%) |
| ANA titre | 2 (14%) | 1 (6%) | 11 (41%) | 16 (40%) | 18 (62%) | 20 (100%) |
| Sm | 0 | 0 | 0 | 0 | 0 | 6 (30%) |
| Ro52 | 0 | 0 | 1 (4%) | 0 | 1 (3%) | 12 (60%) |
| Ro60 | 1 (7%) | 0 | 0 | 0 | 0 | 13 (65%) |
| La | 0 | 0 | 0 | 0 | 0 | 3 (15%) |
| Jo | 0 | 0 | 0 | 0 | 0 | 0 |
| Ribonucleoprotein | 0 | 0 | 0 | 0 | 0 | 14 (70%) |
| Ribosomal protein | 0 | 0 | 0 | 1 (3%) | 0 | 6 (30%) |
| RNA Pol 3 IgG | 0 | 0 | 2 (7%) | 4 (10%) | 2 (7%) | 8 (40%) |
| RF IgM | 0 | 1 (6%) | 2 (7%) | 2 (5%) | 4 (14%) | 8 (40%) |
| RF IgA | 0 | 0 | 0 | 0 | 2 (7%) | 4 (20%) |
| Citulinated protein | 0 | 0 | 1 (4%) | 0 | 3 (10%) | 3 (15%) |
| Prothrombin IgM | 0 | 2 (11%) | 4 (15%) | 1 (3%) | 0 | 4 (20%) |
| Prothrombin IgG | 0 | 0 | 0 | 1 (3%) | 0 | 7 (35%) |
| Cardiolipin IgM | 0 | 0 | 0 | 0 | 1 (3%) | 0 |
| Cardiolipin IgA | 0 | 0 | 0 | 0 | 2 (7%) | 1 (5%) |
| Cardiolipin IgG | 0 | 2 (11%) | 2 (7%) | 1 (3%) | 2 (7%) | 1 (5%) |
| B2GP1 IgM | 0 | 0 | 0 | 0 | 0 | 0 |
| B2GP1 IgA | 0 | 0 | 0 | 0 | 0 | 1 (5%) |
| B2GP1 IgG | 0 | 1 (6%) | 1 (4%) | 2 (5%) | 1 (3%) | 3 (15%) |
| MPO | 0 | 0 | 0 | 0 | 0 | 0 |
| PR3 | 0 | 0 | 0 | 0 | 1 (3%) | 0 |
| ANCA | 0 | 0 | 3 (11%) | 4 (10%) | 1 (3%) | 12 (60%) |
| p70 | 0 | 0 | 0 | 0 | 1 (3%) | 9 (45%) |
| Carbamylated protein | 0 | 0 | 11 (41%) | 10 (25%) | 11 (38%) | 14 (70%) |
| GBM | 0 | 0 | 2 (7%) | 0 | 1 (3%) | 0 |

supplementing them with a further 50 ICU patient plasma samples, a cross-sectional analysis of ANA reactivity as a function of the day after COVID-19 symptom onset demonstrated a clear and significant emergence of autoreactivity that can be identified between days 10 and 15 following symptomatic severe infection (Fig. 2j, k).

## Self-reactivity in antiviral response

In addition to autoimmune serologies and repertoire features of IgG1 ASC, the contribution of IgG1 to autoreactivity in ICU-C was supported by the identification of IgG1-specific ANA reactivity in that cohort that could not be identified in IgG2 (Extended Data Fig. 7a). To investigate this possibility, two patients (ICU-1 and ICU-2) were identified for individual clonotype assessment and monoclonal antibody (mAb) production and testing. These patients were representative of the overall cohort, with low ASC mutation frequencies and high incorporation of autoreactive-prone *IGHV4-34* clonotypes (Extended Data Fig. 7b, c). In patient ICU-1, low-mutation ASCs had more connections to the CD27⁻ B cell fraction than the memory compartment, and high levels of IgM ASC connectivity to IgG1 ASCs in both patients were suggestive of recent development (Extended Data Fig. 7d, e).

Clonotypes were selected from this ASC compartment (54 and 53 clonotypes from ICU-1 and ICU-2, respectively) on the basis of their inclusion of an IgG1 member, low mutation frequency (<1%), and presence in the ASC compartment, CD27⁻ compartment or both. In addition to all expanded clonotypes (more than five members), all *IGHV4-34*-expressing members were included in the screening analysis.

mAbs were screened against several SARS-CoV-2 antigens including S1, RBD, amino-terminal domain (NTD), S2, ORF-3 and nucleocapsid (Fig. 3a)[31], with more than 65% showing binding to one of the tested target antigens (Fig. 3a, b). Despite similar frequencies in antiviral targeting, responses against nucleocapsid and the spike NTD differed between patients ICU-1 and ICU-2, indicating potential differences in the response microenvironment. Despite their naive origin and low (or absent) levels of somatic hypermutation (SHM), many of the resulting antibodies had high affinity, with $K_D$ values in the low nanomolar range (Supplemental Table 4). The top binders to spike and nucleocapsid had affinities of $K_D = 2.82 \times 10^{-9}$ and $K_D = 9.93 \times 10^{-10}$, respectively, in the range of several published neutralizing antibodies[32] (Fig. 3a and Extended Data Fig. 8). Of interest, *IGHV4-34*-expressing clones were generally viral targeted (Fig. 3a). Overall, these data confirm this compartment as enriched for antigen-specific ASCs contributing to the emerging antiviral response.

However, despite the dominance of SARS-CoV2-specific ASC, ~30% of the clones tested did not have clear specificity for the tested proteins, and many showed low binding (Fig. 3a). Given these findings, combined with the low selective pressures, it was important to understand whether these antibodies also contained autoreactive potential. To this end, mAbs were screened for ANA binding as an established method for broad human B cell autoreactivity assessment[33]. In accordance with patients' ANA serum positivity, 16% of all 107 mAbs showed ANA reactivity, equally distributed between the two patients (Fig. 3c). Specific antigen targeting was heterogeneous, with individual reactivities identified against cytoplasmic, nuclear, membrane, cytoskeleton and

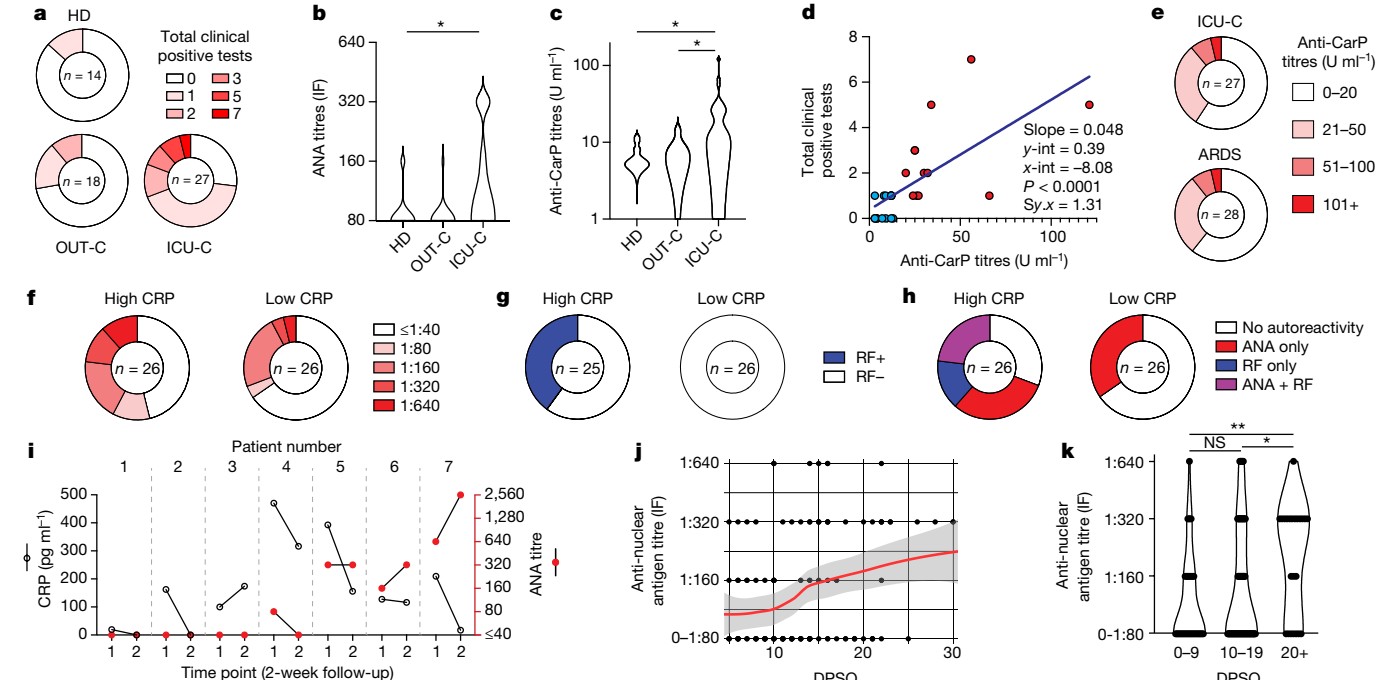

**Fig. 2 | Characterizing clinical autoreactivity profiles in COVID-19. a–e**, HD, OUT-C, ICU-C and ARDS patient frozen plasma was tested against a variety of autoantigens in Exagen's clinical laboratory. Frequency of total positive clinical tests across the HD, OUT-C and ICU-C cohorts (**a**). Distribution of ANA titres across the HD, OUT-C and ICU-C cohorts (**b**). Distribution of anti-CarP titres across the HD, OUT-C and ICU-C cohorts (**c**). Linear regression of anticarbamylated protein titres versus total number of patient autoreactive breaks across the ICU-C cohort. Patients with positive anti-CarP titres are highlighted in red (**d**). Frequency of anti-CarP responses, broken down by titre in HD, OUT-C, ICU-C and bacterially induced ARDS cohorts (**e**). **f**, Frequency of ANA titres in high versus low CRP patients in the independent ICU cohort. **g**, Frequency of RF-positive tests in high versus low CRP patients in the independent ICU cohort. **h**, Frequency of ANA- and RF-positive tests in high versus low CRP patients in the independent ICU cohort. **i**, Two-week follow-up testing of seven patients from the independent ICU cohort. CRP and ANA titres are shown. **j,k**, Immunofluorescence (IF) ANA titres were assessed for the combined patient cohorts (Fig. 2a,f), alongside a further 50 ICU patients (total $n = 129$). ANA reactivity as a function of time after symptom onset. Red line indicates LOESS regression with 95% CI (**j**). Time-point-binned assessment of IF ANA reactivity (**k**). In **b**, **c** and **k**, statistical significance was determined using analysis of variance with Tukey's multiple-comparisons testing between all groups. $*P \leq 0.05$; $**P \leq 0.01$; $***P \leq 0.001$.

Golgi antigens (Extended Data Fig. 9a). Further screening against the highly disease-specific GBM antigen resulted in several positive hits in the patient with anti-GBM serum reactivity (ICU-1; 4/54 or 8%) with three of four also showing antiviral affinity (Fig. 3d). Binding to human naive B cells, a feature of *IGHV4-34* antibodies in SLE linked to reactivity against the naive B cell surface, was also tested[34,35]. Consistent with autoreactive potential against B cells and other lupus antigens, 10 of 30 VH4-34 antibodies demonstrated B cell binding, with four of them showing reactivity to SARS-CoV-2 antigens as confirmed through surface plasmon resonance (Fig. 3a and Extended Data Fig. 9b)[34].

In total, 65% (15/23) of mAbs with identified autoreactivity had some affinity to a screened viral antigen, highly similar to the overall antiviral reactivity of the total antibody pool. Autoreactivity was independent of SHM, with more than half of identified self-targeted antibodies (14/23) having germline BCRs (Fig. 3a). Crossreactivity between self-antigens and the RBD (highly specific to SARS-CoV-2) further confirmed the naive origins of these autoreactive responses, and the heterogeneity of antiviral targets associated with self-reactivity strongly favours a model in which relaxed selective pressure in the ASC compartment, rather than dominant molecular mimicry driven by a specific viral protein, is likely to be responsible for the emergence of autoreactivity observed in this cohort.

## Uneven autoreactive recovery in COVID-19

To understand the evolution of the low-mutation ASC compartment in acute disease resolution, patients ICU-1–3 were recruited for follow-up between 6 and 10 months after symptom onset (Supplemental Table 3).

All three patients showed a contraction of the overall IgG1 ASC compartment from the acute phase of disease, with two showing reductions of more than 50% (Fig. 4a, b), down to frequencies comparable with those observed in steady-state HD (Fig. 1d). Of more than 900 independent IgG1 ASC clonotypes identified in the acute phase of disease, only two could be detected in the recovery phase in both memory and ASC compartments. None of the 107 characterized clones was persistent at recovery, irrespective of antiviral targeting. IgG1 ASC mutation frequencies were increased at recovery to HD steady-state levels (Fig. 4c), and the nature of these mutations reflected a normalization of selective pressures at levels similar to those of other contemporaneous class-switched ASC compartments (Fig. 4d). Renewed censoring of *IGHV4-34* clonotypes in the ASC compartment across all three patients and reductions in *IGHV4-34*[+] IgG antibodies in the plasma at recovery time points further confirmed the re-establishment of tolerance standards (Fig. 4e, f).

However, despite universal signs of a return to 'normal' tolerance environments in the ASC compartment, the resolution of clinical autoreactivities was more complex. Although two of three patients (ICU-1 and ICU-2) showed evidence of resolving autoreactivity in the blood across several target antigens (including high titres of anti-GBM antibodies), one of the two seemed to have increased reactivity against cardiolipin 7 months following disease onset (Extended Data Fig. 10a–c). Further, patient ICU-3 showed increased reactivity against both RF and CarP antigens at 10 months after symptom onset versus the acute phase of infection, indicating that in a subset of patients, clinical autoreactivity may persist well beyond the acute phase of infection (Extended Data Fig. 10c). To assess this possibility, plasma from 40 ICU-recovered

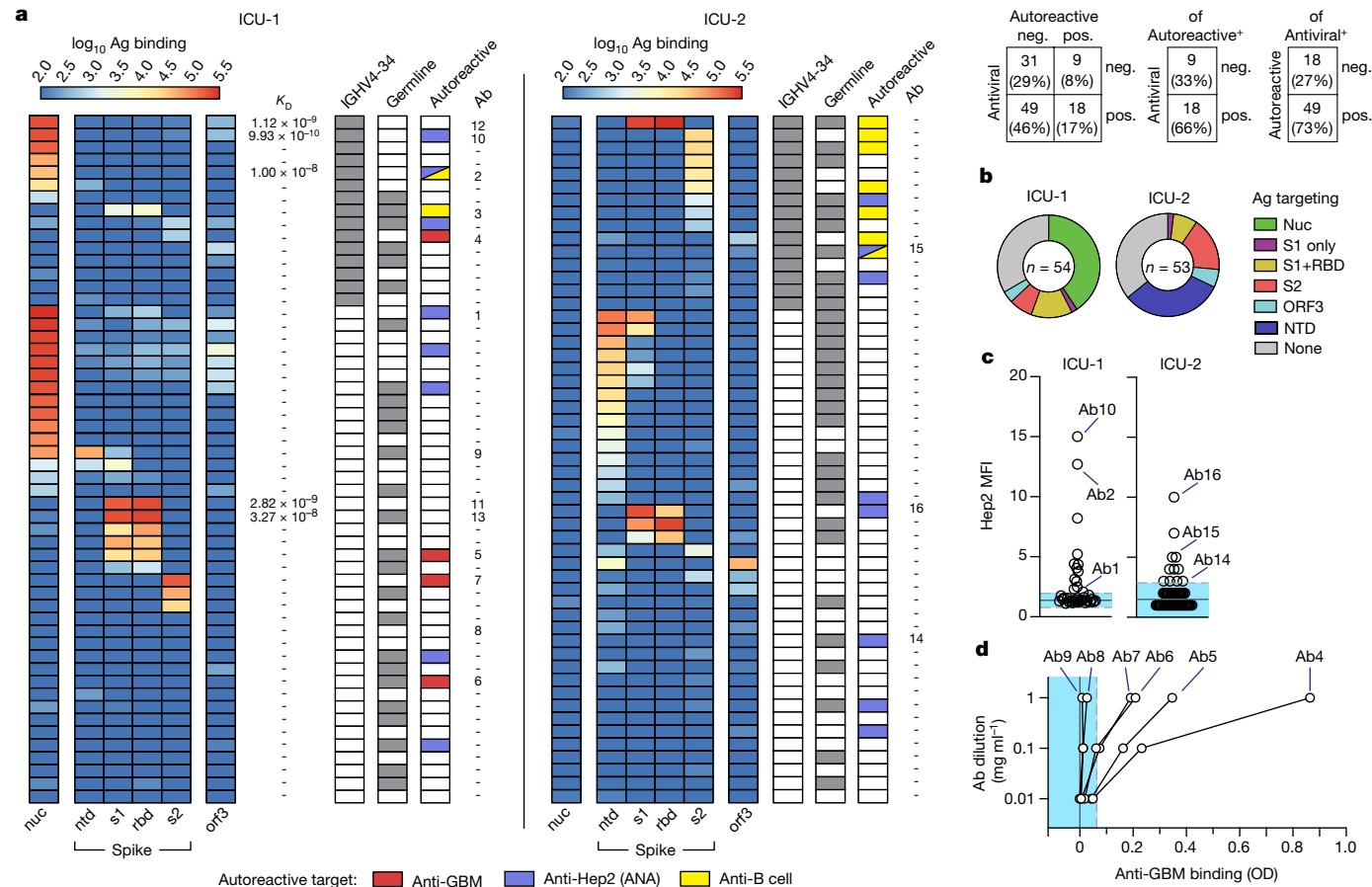

**Fig. 3 | IgG1 clonotypes are both antiviral and autoreactive. a**, Overview of clonotype (mAb) testing from patients ICU-1 and ICU-2 (total *n* = 107). Clonotypes were selected from the IgG1+ low-selection compartment described in Figs. 1,2. Left: heatmaps of mAb (rows) binding to indicated antigens (columns). Middle: $K_D$, antibody affinities confirmed through HT-SPR; IGHV4-34, clonotype encodes IGHV4-34 receptor; germline, clonotype shows germline heavy and light chain configurations; autoreactive, clonotype shows autoreactivity against indicated autoantigen. Right: Ab designation to aid tracking throughout Fig. 4. **b**, SARS-CoV-2 antigen targeting across all 107 mAbs. **c**, MFI measurements of Hep2 cell line reactivity by synthesized mABs using immunofluorescence. Selected mAb designations are indicated (Fig. 4a). **d**, Anti-GBM ELISA testing of isolated mAbs (optical density, OD). Selected clonotype designations are indicated (Fig. 4a). In **c** and **d**, summary statistics are mean negative test value ±3 s.d. pos., positive; neg., negative.

patients with no history of autoimmune disorders was collected from postacute sequelae of COVID-19 (PASC) clinics and combined with that of existing cohorts of acute patients for cross-sectional longitudinal analysis. Consistent with individual patient reactivities, an early emergence of ANA reactivity was observed that persisted at significant albeit tapering levels over the following year (Fig. 4g). Of the 20 PASC patients available more than 100 days after symptom onset, seven (35%) showed ANA reactivity. Similarly, anti-CarP responses remained elevated, albeit at decreased levels, in a large fraction of patients (ICU-C 35%, PASC 25%) in the recovery phase of COVID-19 (Fig. 4h), further stressing the need for continued follow-up of these patients to understand the long-term implications of tolerance breaks on continuing symptomology and chronic autoimmune manifestations.

## Discussion

Although several studies have detailed the presence of autoantibodies in COVID-19, their mechanisms of generation, chronic pathogenic potential and eventual resolution remain to be understood. New recent information has clearly documented the appearance of de novo serological autoreactivity in patients hospitalized with severe infection; however, the precise cellular source of such autoreactivity remains unidentified. Indeed, both the naive and memory B cell compartments of healthy individuals contain large proportions of autoreactive or

polyreactive cells that could be triggered to produce autoantibodies in the context of severe inflammation through a combination of antigen-specific and non-specific stimuli[33,36–38]. Here, we assign that phenomenon, at least in large part, to a transient naive-derived ASC compartment through mechanisms that by and large involve antigen-mediated triggering. These cells, enriched in autoreactive potential, emerge during the acute phase of severe COVID-19 and regress gradually during the recovery phase in most but not all patients. This compartment is characterized by a predominance of IgG1 ASCs expressing antibodies with low levels or absence of SHM distributed in a pattern consistent with low antigenic selection pressure. Emergence of this population is correlated with increased clinical autoreactivity against a variety of self-antigens, routinely including nuclear antigens and carbamylated proteins. Whereas the re-establishment of indicators of selective pressure in the ASC repertoire was consistent among all patients, the presence of autoantibodies in the serum persisted into the recovery phase in many patients experiencing continuing symptoms well into the recovery phase of disease, raising significant questions as to their contributions to postacute sequelae.

The origins of autoreactivity in COVID-19 have been an important area of debate owing to their prognostic potential. Early reporting of autoreactivity against type-I interferons in critically ill patients indicates that if these autoantibodies predate the infection, they may help to predict those at high risk[7]. Here, we demonstrate definitively

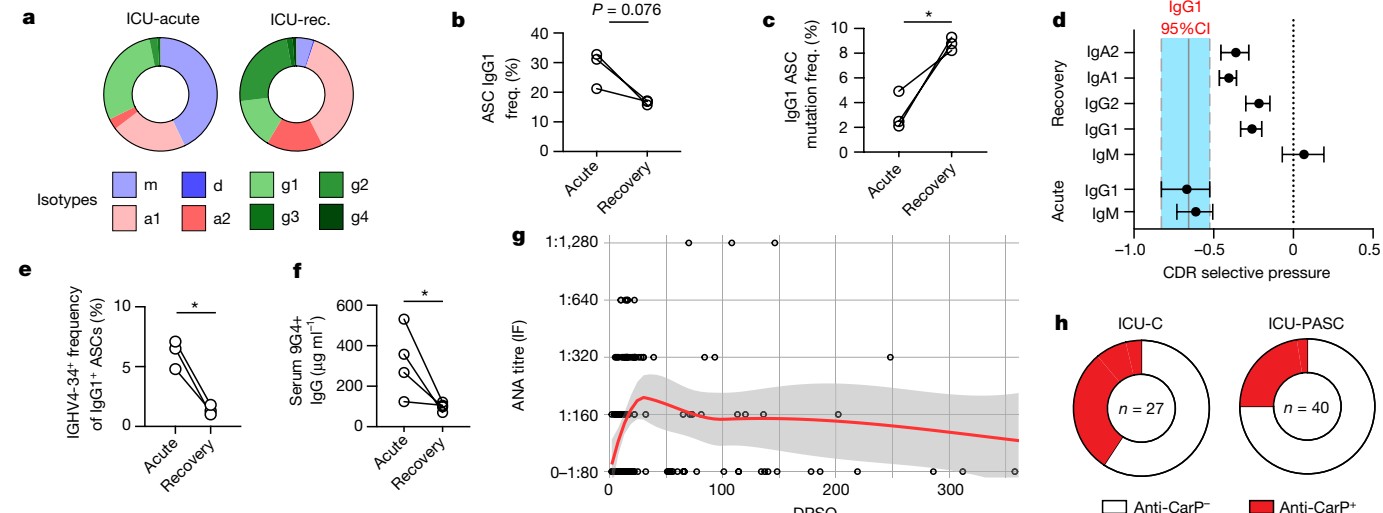

**Fig. 4 | Relaxed peripheral tolerance resolves in the repertoire on recovery.** **a**, Average isotype frequencies at acute and recovery time points from ICU-C patient cohort (180–300 days post symptom onset (DPSO), $n = 3$). **b**, IgG1 ASC isotype frequency in acute and recovery ICU-C cohorts. **c**, *IGHV* nucleotide mutation frequency in IgG1 ASCs in acute versus recovery samples in ICU-C cohort. **d**, ASC selective pressure comparisons of selected isotypes from acute or recovery ICU-C cohort. Bars represent 95% CI in the group. **e**, *IGHV4-34*[+] ASC frequency in IgG1 ASCs in acute versus recovery samples in acute and recovery ICU-C cohorts. **f**, ELISA assessment of *IGHV4-34*[+] IgG plasma antibody concentration in acute and recovery ICU-C cohorts ($n = 4$). **g**, IF ANA titres were assessed for the combined acute patient cohorts (Fig. 2j), alongside 45 ICU patients at the indicated recovery time points (total $n = 174$). ANA reactivity was assessed as a function of time after symptom onset. Red line indicates LOESS regression with 95% CI. **h**, Frequency of anti-CarP positive reactivity in acute ($n = 27$) versus recovery ($n = 40$) ICU-C cohorts. In **b**, **c**, **e** and **f**, statistical significance was determined using paired two-tailed *t*-test between the indicated groups. *$P \le 0.05$; **$P \le 0.01$. rec., recovery.

that autoantibodies of substantial affinity can be generated, de novo, at the earliest phases of the humoral immune response. Indeed, the identification of RBD-specific clonotypes with germline BCR configurations and autoreactive targeting confirms that autoreactivity and antiviral targeting can be generated simultaneously in the robust EF responses identified in severe COVID-19. Thus, although preformed autoantibodies are likely to have an important role in specific aspects of infection severity, they are unlikely to account for the robust autoreactive phenotypes identified routinely in these patients. Instead, the current work establishes experimentally that the early emergence of isotype-switched autoreactivity is not a proxy for pre-existing autoreactive memory.

Although emphasized in COVID-19, autoreactivity following severe viral infection has been well documented in mice, with various potential mechanisms proposed. Early work by Roosnek and Lanzavecchia described efficient non-cognate antigen presentation by autoreactive B cells as a mechanism for autoreactivity induction[39]. That model was invoked a decade later to explain the significant fraction of autoreactive clonotypes emerging from lymphocytic choriomeningitis infection[40]. Molecular mimicry, an independent model of tolerance breakdown, has also been postulated as the source of autoreactivity in viral infection. Best described in rheumatic fever, in which antistreptococcal antibodies crossreact with cardiac myosin, different types of molecular mimicry have been invoked across a variety of infectious diseases[41–45]. In SLE, it has been suggested that peptide homology between Epstein–Barr virus and ribonucleoproteins could lead to B cell epitope spreading and disease development[46]. Our experimental data do indeed demonstrate a degree of crossreactivity between SARS-CoV2 antigens and a variety of self-antigens, an observation that would be likely to expand with more extensive testing against more comprehensive self-antigen arrays. However, specifically measuring the degree to which molecular mimicry accounts for such crossreactivity would require extensive molecular and structural studies of various antigens and antigen–mAb structures outside the scope of the current work. In a strict sense, whereas molecular mimicry would require the sharing of a linear or conformational epitope between different antigens, crossreactivity at large may be mediated by binding to separate antigens devoid of shared epitopes through separate parts of the antigen-binding site, a promiscuity that is enhanced by the large and heavily charged CDR3 frequently enriched in autoreactive B cells.

Although our experimental approach does not address these mechanisms directly, the identification of extensive ASC crossreactivity between viral and self-antigens indicates that the most robust manifestation of molecular mimicry—a specific pathogenic protein driving autoreactivity against a consensus self-antigen—may not be the primary driver of the autoreactivity emerging in COVID-19. This postulate is consistent with the lack of correlation between individual viral targets and specific self-antigens, as autoreactivity could be identified in clones targeting all tested components of SARS-CoV-2. Further, the same broad autoantigens (naive B cells, for example) could be targeted by antibodies with specificity to nucleocapsid or RBD, or could have no discernable affinity to the dominant viral antigens tested (Fig. 3a). Instead, the data presented here are most consistent with a model by which the highly inflammatory milieu created by severe COVID-19 would promote the unopposed expansion of a positively selected naive compartment endowed with substantial germline-encoded autoreactivity through the EF response pathway[12,47]. This scenario would result in the rapid conversion of autoreactive activated T-bet[+] naive B cells and their intermediary DN2 effectors into functional autoantibody-producing ASCs, a mechanism strongly driven by Th1-like cytokines prominently including IFN gamma, which is highly correlated with COVID-19 severity[15,48], as we and others have documented in acute SLE[12,13,49].

This model, in which the initially expanded autoreactivity would be enriched for self-reactivities not subject to strong central tolerance and readily present in the naive compartment, might help explain the enduring tolerance against some antigens, such as dsDNA and MPO, which would be abundant in the local milieu of severe COVID-19 owing to strong neutrophil activation and neutrophil extracellular trap formation[50]. This was also true of individual ANA antigens including La, Sm and Ro, which are associated with SLE but remained negative

throughout acute infection. This profile would be consistent with broad expansion of the autoreactivity previously documented in human naive B cells, which is enriched for ENA-negative ANA[+] reactivity[33] and normally censored in the germinal centre[51]. Accordingly, our studies highlight the immunological consequences of uncensored EF expansion of autoreactive naive B cells in severe COVID-19 infection and indicate that common, clinically testable autoreactivities including ANA and anti-CarP reactivity may be useful in identifying these phenomena in a variety of severe infectious diseases in real time[14]. The pathologic potential of individual reactivities that emerges in these patients remains to be established; however, the generation of autoantibodies associated with autoimmune diseases with antibody-mediated pathology, including anti-CarP and GBM, strongly suggests a pathogenic role.

A critical finding of this study is the restoration of normal features in the IgG1 ASC compartment after recovery, including size contraction and increased levels of somatic mutation and selection pressure on par with contemporaneous memory cells of the same subclass. These changes indicate a dynamic process of acute expansion of naive-derived IgG1 ASCs enriched in autoreactivity that dominates during severe infection and subsequently subsides. However, despite clear resolution at the cellular level, kinetic analysis of autoreactive serology presents a more subtle picture, with general declines in autoreactivity that nonetheless may persist at significant levels for several months. In some patients, such as ICU-3, autoreactivity may even increase postinfection; it will be important to know whether these features are associated with the future emergence of chronic autoimmunity.

This mixed picture is consistent with established properties of the EF response, not only in the dominant generation of short-lived plasmablasts but also in their less-appreciated potential to generate long-lived plasma cells and to contribute to memory responses[52]. Although the absence of acute phase clonotypes from IgG1 memory at recovery argues against robust memory incorporation, this finding is tempered by the necessarily restricted depth of repertoire tracking afforded by single-cell analysis and the lack of direct examination of tissue-based memory and plasma cells. Hence, combinations of large correlative clinical studies and more extensive cellular studies will be required to understand whether acute relaxations of tolerance do indeed result in an increased susceptibility to chronic autoimmunity in a small subset of patients. These studies could help identify a therapeutic window wherein, as in autoimmunity, infectious disease treatment could be tailored to diminish the generation and survival of autoreactive B cells. Further, interfering with the maturation of autoreactive naive B cells using anti-BAFF or similar therapies[53], depletion of emerging pathogenic autoantibodies using anti-ASC agents[54] or strategies aimed at cycling the patient's IgG fraction[55] may improve recovery outcomes. This current study informs that important future work, characterizing the immunologic underpinning of emerging primary autoreactivity in COVID-19 and identifying potential avenues for monitoring those responses, in real time, in a clinical setting.

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

## Methods

### Human participants

All research was approved by the Emory University Institutional Review Board (nos. IRB00058507, IRB00057983 and IRB00058271) and was performed in accordance with all relevant guidelines and regulations. Written informed consent was obtained from all participants or, if they were unable to provide informed consent, from designated healthcare surrogates. Healthy individuals ($n = 20$) were recruited using promotional materials approved by the Emory University Institutional Review Board. Individuals with COVID-19 ($n = 19$) were recruited from Emory University Hospital, Emory University Hospital Midtown and Emory St. Joseph's Hospital (all Atlanta, USA). All nonhealthy individuals were diagnosed with COVID-19 by PCR amplification of SARS-CoV-2 viral RNA obtained from nasopharyngeal or oropharyngeal swabs. Individuals with COVID-19 were included in the study if they were between 18 and 80 years of age, were not immunocompromised and had not been given oral or intravenous corticosteroids during the preceding 14 days. Peripheral blood was collected in either heparin sodium tubes (for PBMCs) or serum tubes (for serum) (both BD Diagnostic Systems). Baseline individual demographics are included in Supplementary Table 1. Study data were collected and managed using REDCap electronic data capture tools hosted at Emory University.

Banked frozen plasma from patients with ARDS ($n = 28$) was obtained as previously described[56].

### PBMC cell isolation and plasma collection

Peripheral blood samples were collected in heparin sodium tubes and processed within 6 h of collection. PBMCs were isolated by density gradient centrifugation at 1,000$g$ for 10 min. Aliquots from the plasma layer were collected and stored at −80 °C until use. PBMCs were washed twice with RPMI at 500$g$ for 5 min. Viability was assessed using trypan blue exclusion, and live cells were counted using an automated hemocytometer.

### Flow cytometry

Isolated PBMCs ($2 \times 10^6$) were centrifuged and resuspended in 75 µl FACS buffer (phosphate-buffered saline (PBS) and 2% fetal bovine serum (FBS)) and 5 µl Fc receptor block (BioLegend, no. 422302) for 5 min at room temperature. For samples stained with anti-IgG, it was observed that Fc block inappropriately interfered with staining, so a preincubation step of the anti-IgG alone for 5 min at 22 °C was added before the addition of the block. Next, 25 µl of antibody cocktail (Supplementary Table 3) was added (100 µl staining reaction), and samples were incubated for 20 min at 4 °C. Cells were washed in PBS and resuspended in a PBS dilution of Zombie NIR fixable viability dye (BioLegend, no. 423106). Cells were washed and fixed in 0.8% paraformaldehyde for 10 min at 22 °C in the dark before a final wash and resuspension for analysis.

Cells were analysed on a Cytek Aurora flow cytometer using Cytek SpectroFlo software. Up to $3 \times 10^6$ cells were analysed using FlowJo v.10 (Treestar) software.

### Analysis software

Computational analysis was carried out in R (v.3.6.2; release 12 Dec 2019). Heat maps were generated using the pheatmap library (v.1.0.12), with data prenormalized (log-transformed $z$ scores calculated per feature) before plotting. Custom plotting, such as that for mutation frequency violin plots, was performed using the ggplot2 library for base analysis, followed by postprocessing in Adobe Illustrator. Alluvial plotting was performed using the ggalluvial package with postprocessing in Adobe Illustrator. Clonotype connectivity analysis was carried out using the R-based 'vegan' package and then visualized with 'pheatmap' before postprocessing in Adobe Illustrator. Statistical analyses were performed directly in R or in GraphPad Prism (v.8.2.1).

Analyses of the single-cell VDJ annotated sequences were performed using the Immcantation tool suite (http://www.immcantation.org) v.4.1.0 pipeline in Docker. This suite contains SHazaM for statistical analysis of SHM patterns as described in Gupta et al., 2015 and BASELINe for analysis of selection pressure as described in Yaari et al.[26]. Visualizations were generated in R using the SHazaM package (v.1.0.2) and then postprocessed in Adobe Illustrator.

### Flow cytometry and sorting of B cell subsets for repertoire sequencing

Frozen cell suspensions were thawed at 37 °C in RPMI with 10% fetal calf serum and then washed and resuspended in FACS buffer (PBS with 2% fetal calf serum). The cells were incubated with a mix of fluorophore-conjugated antibodies for 30 min on ice. The cells were washed in PBS and then incubated with live/dead fixable aqua dead cell stain (Thermo Fisher) for 10 min at 22 °C. After a final wash in FACS buffer, the cells were resuspended in FACS buffer at $10^7$ cells per ml for cell sorting on a three-laser BD FACS (BD Biosciences).

For single-cell analysis, total ASCs were gated as CD3⁻CD14⁻CD16⁻CD19⁺CD38⁺CD27⁺ single live cells, whereas naive B cells were gated as CD3⁻CD14⁻CD16⁻CD19⁺CD27⁻IgD⁺CD38⁺ single live cells.

For bulk sequencing preparations, B cells were enriched using StemCell's Human Pan-B Cell Enrichment Kit (no. 19554; negative selection of CD2, CD3, CD14, CD16, CD36, CD42b, CD56, CD66b and CD123). CD138⁺ ASCs were enriched further using CD138⁺ selection beads according to the manufacturer's instructions (Miltenyi Biotec, no. 130-051-301).

### Single-cell V(D)J repertoire library preparation and sequencing

Cells were counted immediately using a hemocytometer and adjusted to 1,000 cells per microlitre to capture 10,000 single cells per sample loaded in the 10× Genomics Chromium device according to the manufacturer's standard protocol (Chromium Next GEM Single Cell V(D)J Reagent Kits, v.1.1). The 10× Genomics v2 libraries were prepared using the 10x Genomics Chromium Single Cell 5′ Library Construction Kit per the manufacturer's instructions. Libraries were sequenced on an Illumina NovaSeq (paired-end; 2 × 150 bp; read 1:26 cycles; i7 index: 8 cycles, i5 index: 0 cycles; read 2: 98 cycles) such that more than 70% saturation could be achieved with a sequence depth of 5,000 reads per cell.

### Carbodiimide coupling of microspheres to SARS-CoV-2 antigens

Two SARS-CoV-2 proteins were coupled to MagPlex microspheres of different regions (Luminex). Nucleocapsid protein expressed from *Escherichia coli* (N-terminal His6) was obtained from Raybiotech (230-01104-100) and the RBD of spike protein expressed from HEK293 cells was obtained from the laboratory of J. Wrammert63 at Emory University. Coupling was carried out at 22 °C following standard carbodiimide coupling procedures. Concentrations of coupled microspheres were confirmed by Bio-Rad T20 Cell Counter.

### Luminex proteomic assays for measurement of anti-antigen antibody

Approximately 50 µl of coupled microsphere mix was added to each well of 96-well clear-bottom black polystyrene microplates (Greiner Bio-One) at a concentration of 1,000 microspheres per region per well. All wash steps and dilutions were accomplished using 1% BSA, 1× PBS assay buffer. Sera were assayed at 1:500 dilution and surveyed for antibodies against nucleocapsid protein or RBD. After a 1-h incubation in the dark on a plate shaker at 800 rpm, wells were washed five times in 100 µl of assay buffer using a BioTek 405 TS plate washer before applications of 3 µg ml⁻¹ PE-conjugated goat antihuman IgA, IgG and/or IgM (Southern Biotech). After 30 min of incubation at 800 rpm in the dark, wells were washed three times in 100 µl assay buffer, resuspended in 100 µl assay buffer and analysed using a Luminex FLEXMAP

3D instrument running xPONENT 4.3 software. Mean fluorescence intensity (MFI) using combined or individual detection antibodies (anti-IgA, anti-IgG or anti-IgM) was measured using the Luminex xPONENT software. The background value of the assay buffer was subtracted from each serum sample result to obtain MFI minus background (net MFI).

## Statistical analysis

Statistical analysis was carried out using Prism (GraphPad). For each experiment, the type of statistical testing, summary statistics and levels of significance can be found in the figures and corresponding legends. All measurements shown were taken from distinct samples.

## High-throughput surface plasmon resonance

High-throughput surface plasmon resonance (HT-SPR) data were collected through single-cycle kinetic analysis against either SARS-CoV-2 nucleocapsid or spike trimer (S2P). mAbs were prescreened for antigen binding through Luminex-based multiplex binding assessment (above), and select antibodies were analysed for binding affinity. All data were collected with 1:1 referencing collected in real time on a Nicoya Alto HT-SPR instrument with eight referenced channels running in parallel on carboxyl-coated sensors. Ligand binding and regeneration conditions for each antigen were as follows.

**S2P.** SARS-CoV-2 spike trimer was resuspended in Tris acetate buffer, pH 4.5, and immobilized on an EDC/NHS-activated carboxyl sensor for 5 min at 50 µg ml$^{-1}$. Regeneration of the sensor was performed using glycine HCl, pH 2.5, for 1 min.

**Nucleocapsid.** SARS-CoV-2 nucleocapsid protein was resuspended in Tris acetate buffer, pH 6, and immobilized onto an EDC/NHS-activated carboxyl sensor for 5 min at 50 µg ml$^{-1}$. Regeneration of the sensor was performed using glycine HCl, pH 3, for 1 min.

All single-curve kinetics were performed with five threefold analyte dilutions with final concentrations between 222 nM and 914 pM. Analytes were run in phosphate-buffered saline (0.05% Tween), with interactions collected at 25 °C.

## B cell binding assay

Two to three million HD PBMCs were incubated with 5 µg of mAb at 40 °C for 30 min. The cells were washed with 30× volume FACS buffer (1× PBS, 2% FBS) and subsequently stained with antibodies against CD3, CD19, CD27, IgD and IgG, as well as with Zombie NIR. Staining was completed with 0.8% paraformaldehyde for fixation. Flow cytometry analysis was performed on a CytoFLEX (BD Biosciences). Dead cells and doublets were excluded. The mean fluorescence intensity (MFI) of mAb (IgG) was determined on a naive B cell population.

## mAb selection and production

mAb were selected for production from the single-cell repertoire data obtained from patient ICU-1. Individual cells were clustered into clonotypes and then assessed for clonotype size, nucleotide mutation frequency, isotype and connectivity between sorted populations. Through progressive filtering, clonotypes were selected that met the following criteria:
(1) contained at least one IgG1 member;
(2) had at least one member with a mutation frequency of <1%;
(3) had at least one member in the ASC compartment or the CD27$^-$ compartment or contained members in both.

With those criteria met, all expanded clonotypes (>5 individual cells identified in the clonotype) and all *IGHV4-34*$^+$ members were selected for mAb production and screening (55 clonotypes in all). The most frequently repeated BCR sequence from each clonotype was provided to Genscript for antibody production on a standard IgG1 backbone.

## Clinical autoreactivity testing

For autoimmune biomarker analysis, frozen plasma was shipped on dry ice to Exagen, Inc., which has a clinical laboratory accredited by the College of American Pathologists and certified under the Clinical Laboratory Improvement Amendments. Thawed plasma was aliquoted and distributed for the following tests: antinuclear antibodies (ANA) were measured using enzyme-linked immunosorbent assays (ELISA) (QUANTA Lite; Inova Diagnostics) and indirect immunofluorescence (IFA) (NOVA Lite; Inova Diagnostics); anti-dsDNA antibodies were also measured by ELISA and were confirmed by IFA with *Crithidia luciliae*; extractable nuclear antigen autoantibodies (anti-Sm, anti-SS-B/La IgG, anti-Scl-70 IgG, anti-U1RNP IgG, anti-RNP70 IgG, anti-CENP IgG, anti-Jo-1 IgG and anti-CCP IgG) as well as RF IgA and IgM were measured using the EliA test on the Phadia 250 platform (ThermoFisher Scientific); IgG, IgM and IgA isotypes of anticardiolipin and anti-β2-glycoprotein, as well as anti-Ro52, anti-Ro60, anti-GBM, anti-PR3 and anti-MPO, were measured using a chemiluminescence immunoassay (BIO-FLASH; Inova Diagnostics); anti-CarP, anti-RNA-pol-III, and the IgG and IgM isotypes of anti-PS/PT were measured by ELISA (QUANTA Lite; Inova Diagnostics), whereas C-ANCA and P-ANCA were measured by IFA (NOVA Lite; Inova Diagnostics). All assays were performed following the manufacturer's instructions.

## BALF plasma cell gene expression

To assess the constant region gene expression in BALF-derived ASCs, data were retrospectively analysed from the UCSC data browser available at https://www.nupulmonary.org/covid-19-ms1. In brief, these data are representative of ten ICU patients whose BALF was collected within 48 h of intubation, with total isolated cells sequenced using the 10× single-cell transcriptomics platform. Patient information and full methods are available in the associated manuscript[25].

## MENSA generation

Medium enriched for newly synthesized antibodies (MENSA) was generated by isolating, washing and culturing ASC-containing PBMC from blood using a modified procedure previously described (REF). PBMC were isolated by centrifugation (1,000g; 10 min) using Lymphocyte Separation Media (Corning) and Leucosep tubes (Greiner Bio-One). Five washes with RPMI-1640 (Corning) were performed to remove serum immunoglobulins (800g; 5 min), with erythrocyte lysis (3 ml; 3 min) after the second wash and cell counting after the fourth. Collected PBMCs were cultured at 10$^6$ cells ml$^{-1}$ in R10 medium (RPMI-1640, 10% Sigma FBS, 1% Gibco antibiotic/antimycotic) on a 12-well sterile tissue-culture plate for 24 h at 37 °C and 5% CO$_2$. After incubation, the cell suspension was centrifuged (800g; 5 min), and the supernatant (MENSA) was separated from the PBMC pellet, aliquoted and stored at −80 °C for testing.

## COVID-19 multiplex immunoassay

SARS-CoV-2 antigens were coupled to MagPlex microspheres of spectrally distinct regions by carbodiimide coupling and tested against patient samples as previously described. Results were analysed on a Luminex FLEXMAP 3D instrument running xPONENT 4.3 software. MFI using combined or individual PE-conjugated detection antibodies (anti-IgA/anti-IgG/anti-IgM) was measured using the Luminex xPONENT software on enhanced PMT setting. The background value of assay buffer or R10 medium was subtracted from the serum and plasma or MENSA results, respectively, to obtain an MFI minus background (net MFI). Serum and plasma samples were tested at 1:500 dilution, and MENSA was tested undiluted.

## Selection of antigens

**MENSA and serum samples.** Four recombinant SARS-CoV-2 antigens were used in this study. The nucleocapsid protein (catalogue no. Z03480; expressed in *E. coli*), the S1 domain (amino acids 16–685; catalogue no. Z03485; expressed in HEK293 cells) of the spike protein, and the S1-RBD

(catalogue no. Z03483; expressed in HEK293 cells) were purchased from GenScript. The S1-NTD (amino acids 16–318) was custom synthesized by GenScript. Each protein was expressed with an N-terminal His6-tag to facilitate purification (>85% pure) and appeared as a predominant single band in sodium dodecyl sulfate polyacrylamide gel electrophoresis analysis.

**mAb testing.** RBD (catalogue no. Z03483; expressed in HEK293 cells) and nucleocapsid protein (catalogue no. Z03480; expressed in *E. coli*) were purchased from GenScript (same as the first version). S1 (catalogue no. S1N-C52H3; HEK293), S2 (catalogue no. S2N-C52H5; HEK293) and S1-NTD (catalogue no. S1D-C52H6; HEK293) were purchased from ACROBiosystems. The carboxyl terminus sequence of ORF3a (accession: QHD43417.1, amino acids 134–275 plus N-terminal His6-Tag) was sent to Genscript for custom protein expression in *E. coli*.

## Reporting summary

Further information on research design is available in the Nature Research Reporting Summary linked to this article.

## Data availability

Source data for Figs. 1, 2 and 4 are provided with the paper. All flow cytometry(FCM) and sequencing data presented here are publicly available in alignment with current requirements for public disclosure before peer review. All FCM data presented and analysed in this manuscript (Fig. 1) are publicly available in the FlowRepository at http://flowrepository.org/id/FR-FCM-Z2XF/.

56. Nirappil, F. J. et al. Characteristics and outcomes of HIV-1-infected patients with acute respiratory distress syndrome. *J. Crit. Care* **30**, 60–64 (2015).

**Acknowledgements** This work was supported by National Institutes of Health grants UL TR000424 (Emory Library IT), U54-CA260563-01 Emory SeroNet (I.S., F.E.L.), U19-AI110483 Emory Autoimmunity Center of Excellence (I.S.), P01-AI125180-01 (I.S., F.E.L.), R37-AI049660 (I.S.), 1R01AI12125 (F.E.L.), 1U01AI141993 (F.E.L) and T32-HL116271-07 (R.P.R.) and by the Bill and Melinda Gates Foundation (INV-002351, F.E.L.). It was additionally supported by the Lupus Research Alliance (Distinguished Innovator Award, I.S.). Clinical autoreactivity testing was provided by Exagen, Inc. We thank S. Auld, W. Bender, L. Daniels, B. Staitieh, C. Swenson and A. Truong for their expertise and support of our research. We also thank the nurses, staff and providers in the 71 ICU in Emory University Hospital Midtown and the 2E ICU in Emory Saint Joseph's Hospital. We thank S. Rey, S. Demers, M. Hammons, A. Sace and R. LaFon and the Sanz/Lee clinical coordination and sample processing teams for aid in sample identification, collection, preparation and serological screening. Finally, we acknowledge L. Morales-Nebreda for her guidance in the use of previously published datasets.

**Author contributions** M.C.W., R.P.R., S.A.J., F.E.L. and I.S. conceived and directed this study. M.C.W. and R.B. performed flow cytometry, mAb and serum autoreactivity testing and mAb affinity testing used in the study. M.C.-M. and A.S.S. performed single-cell sorting and sequencing. R.P.R., N.S.H. and F.A.A. performed serum and mAb screening against viral antigens. M.C.W., J.H., E.C.C., M.P. and C.M.T. analysed and compiled the resulting data. R.P.R., M.C.R. and A.K. conducted chart review and identified patient samples for study inclusion, and T.A.W., A.D.T., A.N.D., J.E.H., I.A., V.J., K.S.C., D.C.N., S.K., G.S.M., C.L.M. and A.E. provided critical patient samples for the study. M.E.R. and M.P. oversaw critical collaborations for patient autoreactivity screening and mAb affinity testing, respectively. E.C.C. and G.G. provided critical feedback and support in data analysis and interpretation. M.C.W. and I.S. wrote the manuscript, with all authors providing editorial support.

**Competing interests** F.E.L. is the founder of MicroB-plex, Inc., and has research grants with Genentech. M.E.R. is employed by Exagen, Inc. M.P. is employed by Nicoya.

**Additional information**
**Correspondence and requests for materials** should be addressed to F. Eun-Hyung Lee or Ignacio Sanz.

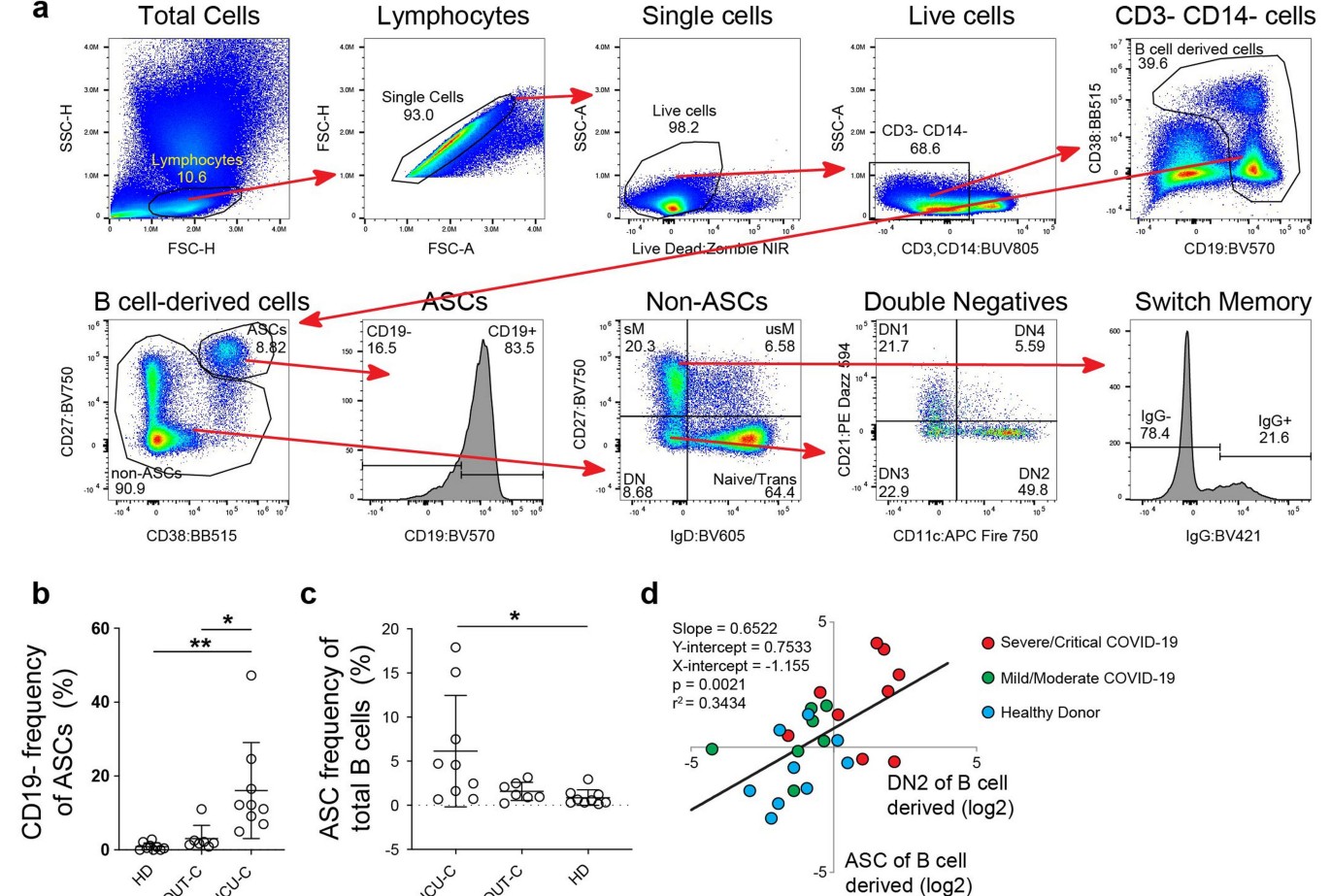

**Extended Data Fig. 1 | EF B cell activation in COVID-19. (a–d)** PBMCs were isolated from HD (n = 9), OUT-C (n = 7), or ICU-C (n = 10) patients and analyzed by spectral flow cytometry. (**a**) Progressive gating strategy for flow cytometry. Label above plot indicated pre-gating population from previous plot. (**b**) CD19⁻ ASC frequency of total ASCs. (**c**) ASC frequency of total B cell-derived cells.

(**d**) Linear regression analysis of log2-transformed DN2 vs ASC frequencies of total B cell-derived cells. (**b, c**) Statistical significance was determined using ANOVA with Tukey's multiple-comparisons testing between all groups. *P ≤ 0.05; **P ≤ 0.01; ***P ≤ 0.001.

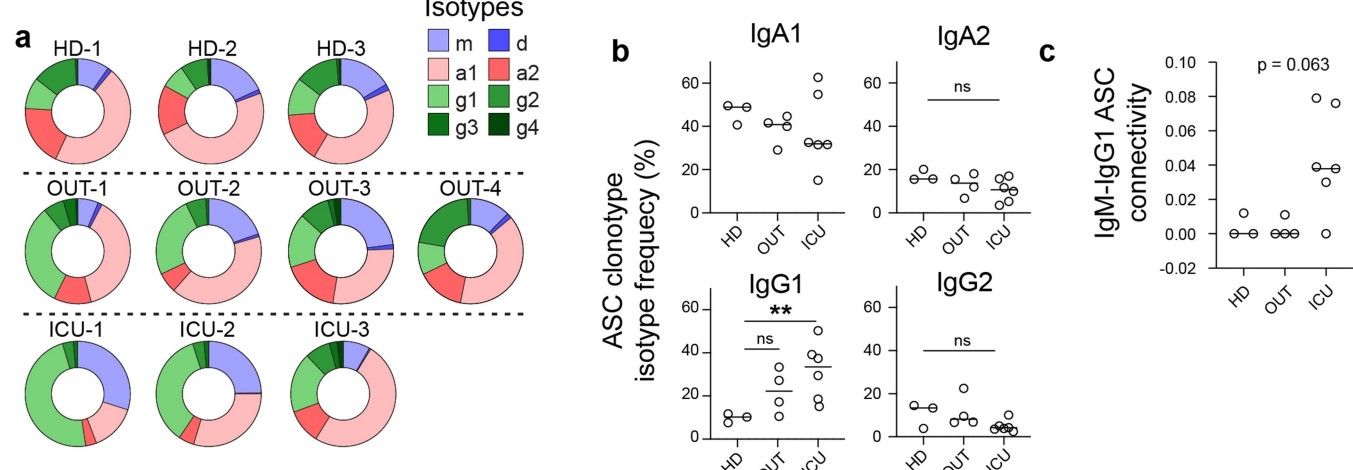

**Extended Data Fig. 2 | IgM-connected IgG1 ASC expansion in severe/critical COVID-19.** (**a–c**) ASCs from HD (n = 3), OUT-C (n = 4), or ICU-C (n = 6) patient cohorts were sorted for single B cell repertoire sequencing and subsequent analysis. (**a**) Isotype frequencies of individual patients within the ICU-C, OUT-C, and HD cohorts (**b**) ASC subclass frequencies by indicated isotype in HD, OUT-C, and ICU-C cohorts. (**c**) Clonotype connectivity between IgM and IgG1 ASCs in HD, OUT-C and ICU-C cohorts. (**b**, **c**) Statistical significance was determined using ANOVA with Tukey's multiple-comparisons testing between all groups. *P ≤ 0.05; **P ≤ 0.01; ***P ≤ 0.001.

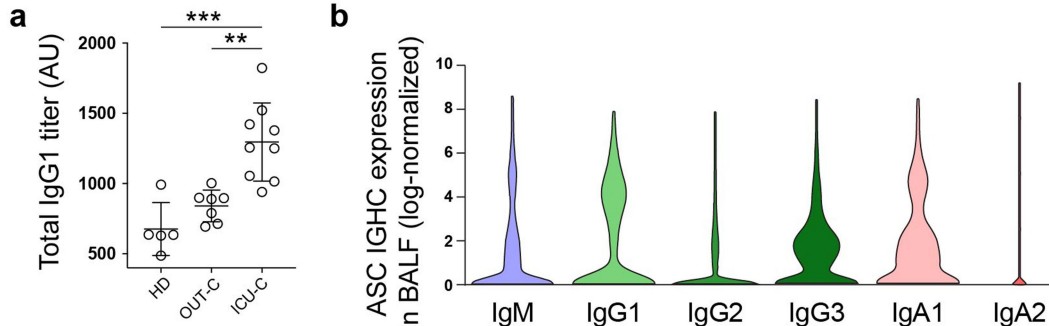

**Extended Data Fig. 3 | IgG1 ASCs are present in the BAL.** (**a**) Statistical significance was determined using ANOVA with Tukey's multiple-comparisons testing between all groups. *P ≤ 0.05; **P ≤ 0.01; ***P ≤ 0.001. (**a**) Bulk IgG1 assessment in HD, OUT-C, or ICU-C cohorts. (**b**) Gene expression of indicated constant region in ASCs identified in the bronchoalveolar fluid from 10 ICU patients. Retrospective analysis of data collected by Grant et. al.

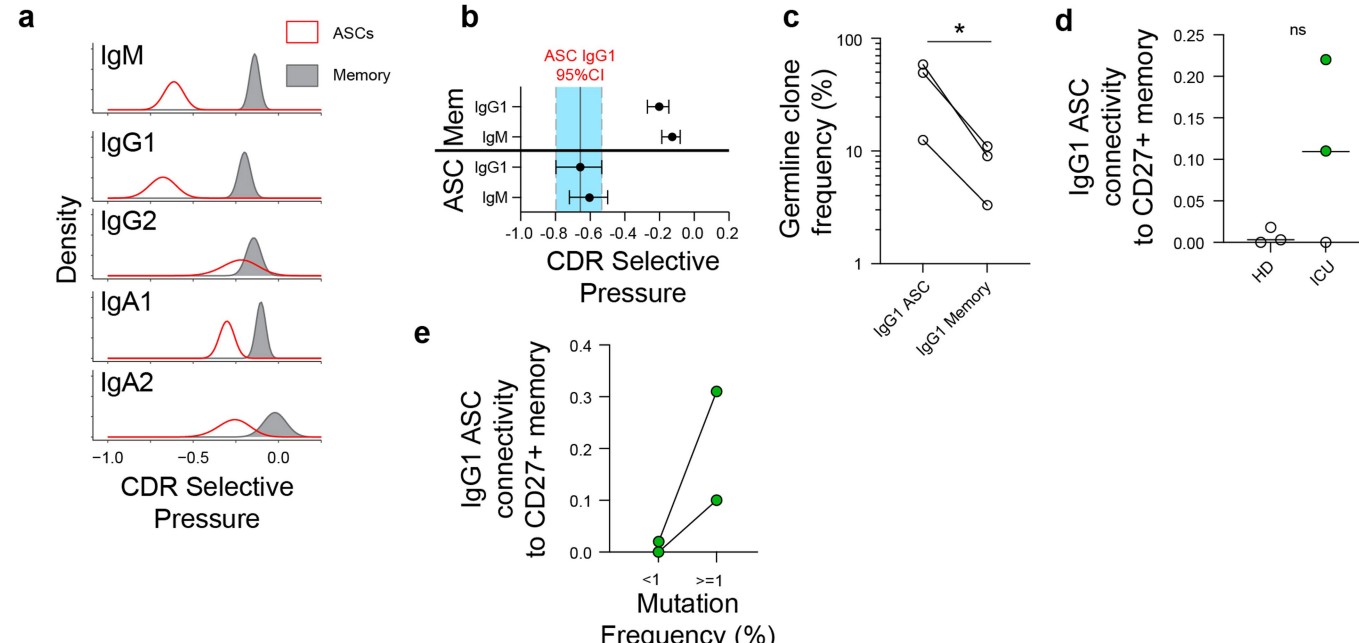

**Extended Data Fig. 4 | Low-mutation IgG1 ASCs are uncoupled from the contemporaneous memory.** (**a-e**) Single cell VDJ analysis of ASCs and memory compartments from ICU-C patients (n = 3) (**a,b**) BASELINe selection analysis of CDR selection in ASC vs. memory B cell populations, grouped by isotype (n = 4). (**b**) Statistical selective pressure comparisons of selected isotypes. Bars = 95% CI (**c**) Frequency of clonotypes whose most expanded member maintains germline heavy and light chain BCR configuration from IgG1[+] ASC or CD27[+] memory compartments. (**d**) Clonotype connectivity between IgG1[+] ASCs and the contemporaneous CD27[+] memory compartment. Patients displaying any connectivity highlighted in green. (**e**) Relative clonal connectivity between mutated (>=1% mutation) versus unmutated (<1%) IgG1[+] ASCs and the contemporaneous memory. Only two patients showing active connection between the compartments [2d] are evaluated. (**c**) Statistical significance was determined using paired two-tailed t-testing between indicated groups. *P ≤ 0.05; **P ≤ 0.01.

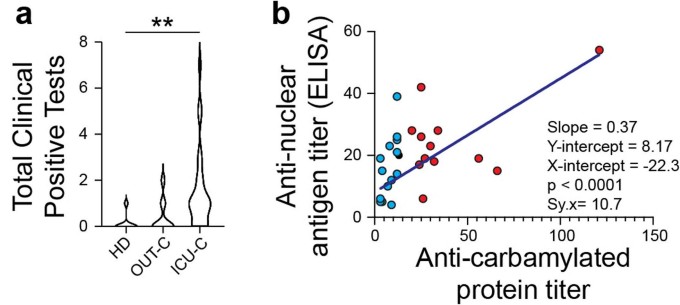

**Extended Data Fig. 5 | Severe COVID-19 correlates with increased autoreactivity against multiple autoantigens.** (**a**, **b**) HD, OUT-C, and ICU-C patient frozen plasma was tested against a variety of autoantigens in Exagen's clinical laboratory. (**a**) Distribution of total positive clinical tests across the HD, OUT-C, and ICU-C cohorts. (**b**) Linear regression of anti-carbamylated protein titers vs. anti-nuclear antigen titers across the ICU-C cohort. Patients with positive anti-CarP titers are highlighted in red.

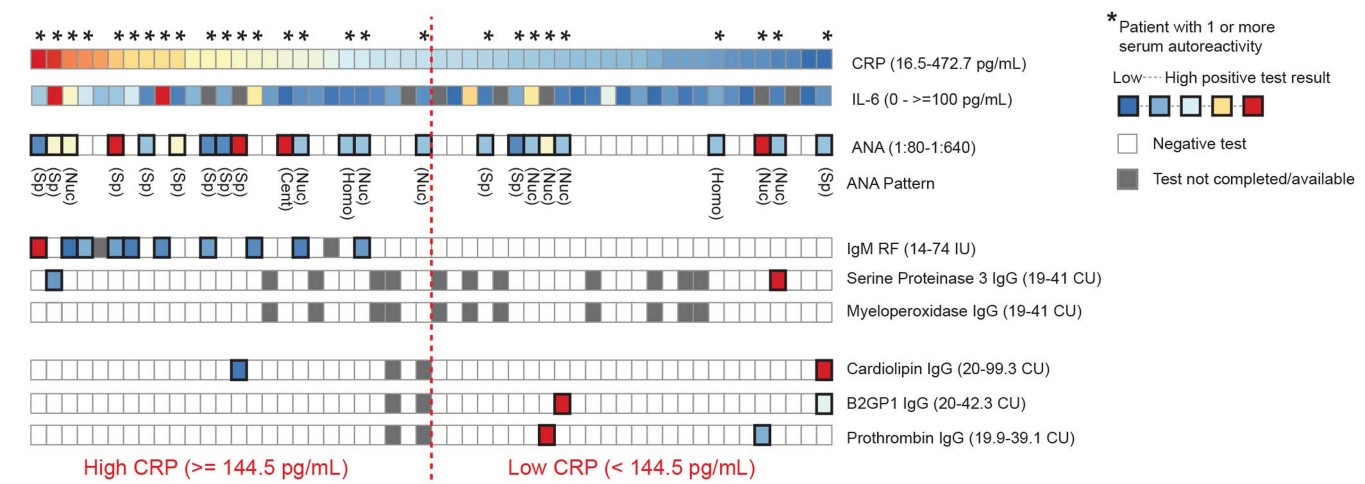

**Extended Data Fig. 6 | Autoreactivity against clinical autoantigens correlates with inflammation.** Heatmap display of Emory pathology-confirmed clinical results of 52 SARS-CoV-2 ICU patients with US NIH "severe" or "critical" clinical designations. Patients are organized by ascending CRP values (range 16.5-472.7). Individual testing scale values are indicated following the test name.

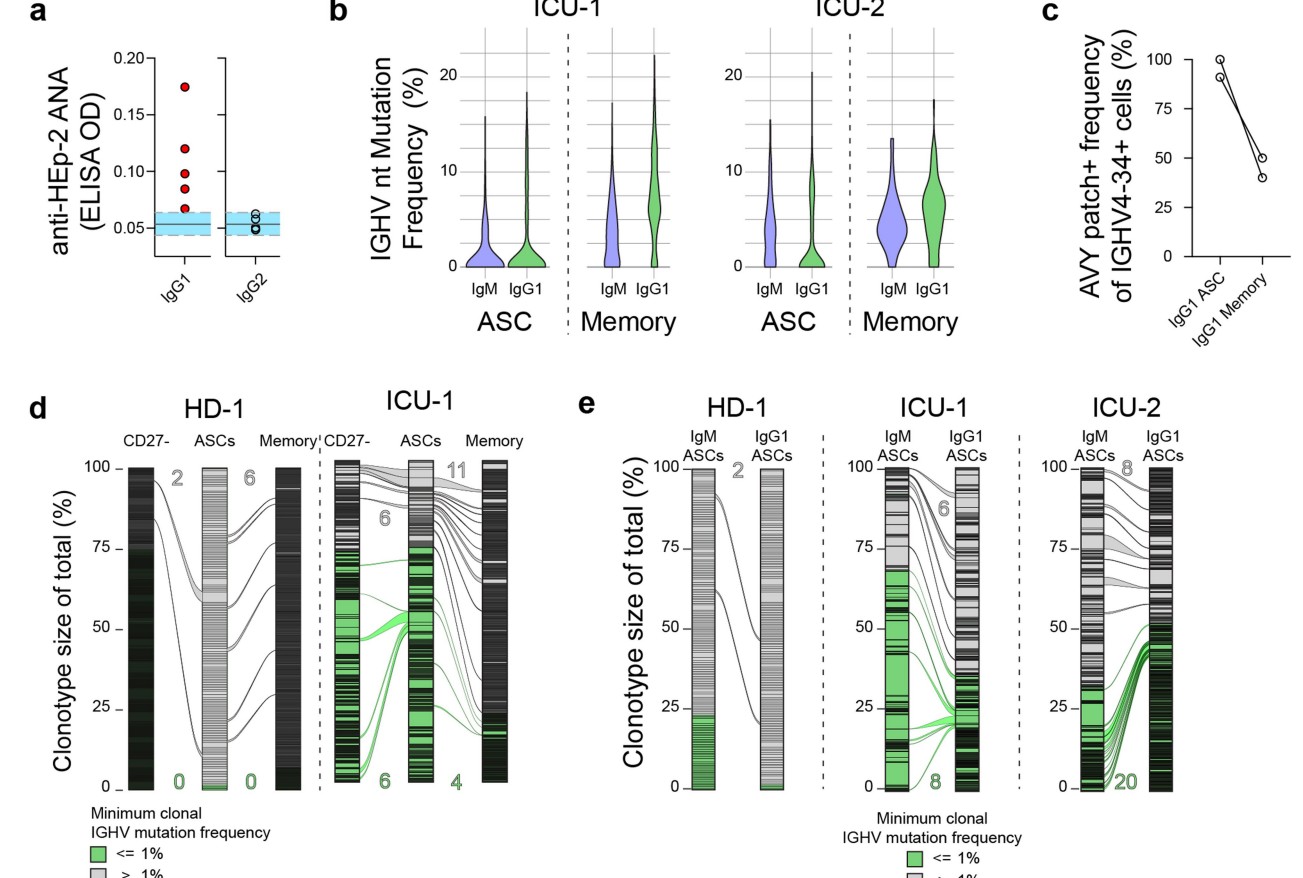

**Extended Data Fig. 7 | Phenotypes of patients selected for antibody screening. (a)** ANA ELISA testing of 5 ICU-C patients with positive clinical testing as determined by Exagen, Inc. [Fig. 2a]. ELISAs were developed with isotype specific IgG1 and IgG2 secondary probes. Red dots indicate positive tests **(b)** Mutation frequency distributions of ICU-1 and ICU-2 ASC and CD27+ memory compartments of indicated isotypes. **(c)** Frequency of autoreactivity-mediating 'AVY' patch integrity in IgG1 ASCs versus IgG1 memory in patients ICU-1 and ICU-2. **(d)** Alluvial plots showing clonotype connectivity between

IgG1 ASCs to the CD27- or memory compartments. Individual clonotypes represented by vertical banding, with the height of the band reflective by the number of cells incorporated into the clonotype. Clonotypes with minimum mutation frequencies <= 1% are highlighted in green. **(e)** Alluvial plots showing clonotype connectivity between IgG1 ASCs to the IgM ASC compartment. Clonotypes with minimum mutation frequencies <= 1% are highlighted in green.

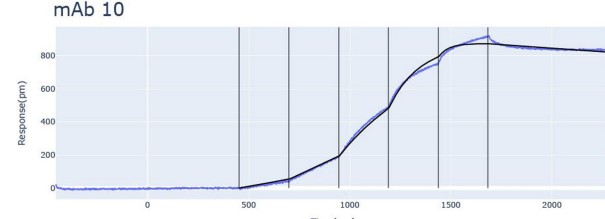

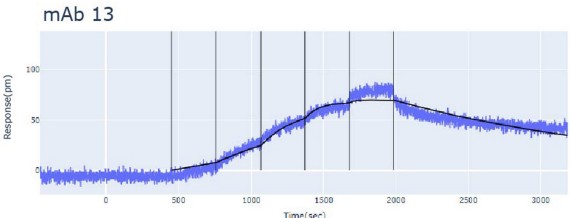

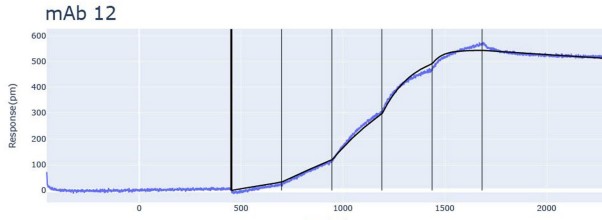

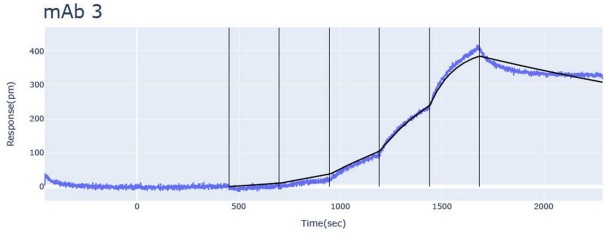

| mAb ID | Target | ka (1/(M*s) | kd (1/s) | KD (M) |
|--------|--------|-------------|----------|--------|
| mAb 11 | S2P | 1.68E+05 (±2.57E+04) | 4.59E-04 (±1.05E-04) | 2.82E-09 (±1.00E-09) |
| mAb 13 | S2P | 2.61E+04 (±7.45E+03) | 8.36E-04 (±2.25E-04) | 3.27E-08 (±8.39E-09) |
| mAb 10 | Nucleocapsid | 1.18E+05 (±3.7E+04) | 1.15E-04 (±2.52E-05) | 9.93E-10 (±9.55E-11) |
| mAb 12 | Nucleocapsid | 1.50E+05 (±5.70E+04) | 1.60E-04 (±7.34E-05) | 1.12E-09 (±4.83E-10) |
| mAb 3 | Nucleocapsid | 4.32E+04 (±9.44E+03) | 4.2E-04 (±9.97E-05) | 1.00E-08 (±3.04E-09) |

**Extended Data Fig. 8 | SPR-based affinity testing of naïve-derived, low mutation monoclonal antibodies.** Representative raw data (blue), and model fitting (black) are displayed for each of the 5 antibodies tested for affinity via HT-SPR. Summary table displays the target, on rate (Ka), off rate (Kd), and affinity (KD), with associated standard deviations in parentheses.

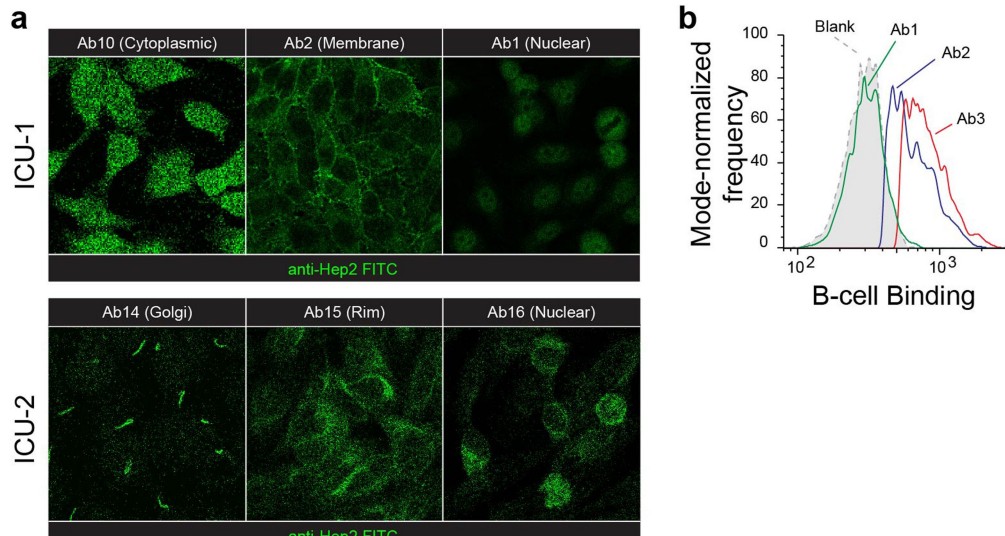

**Extended Data Fig. 9 | Autoantigen reactivity screening of naïve-derived, low mutation monoclonal antibodies. (a)** Representative staining patterns from select mAbs with reactivity against the Hep2 cell line as identified in [Fig. 2c]. Select clonotype designations indicated [Fig. 2a] **(b)** Naive B cell binding of two monoclonal antibodies as identified in [Fig. 2a].

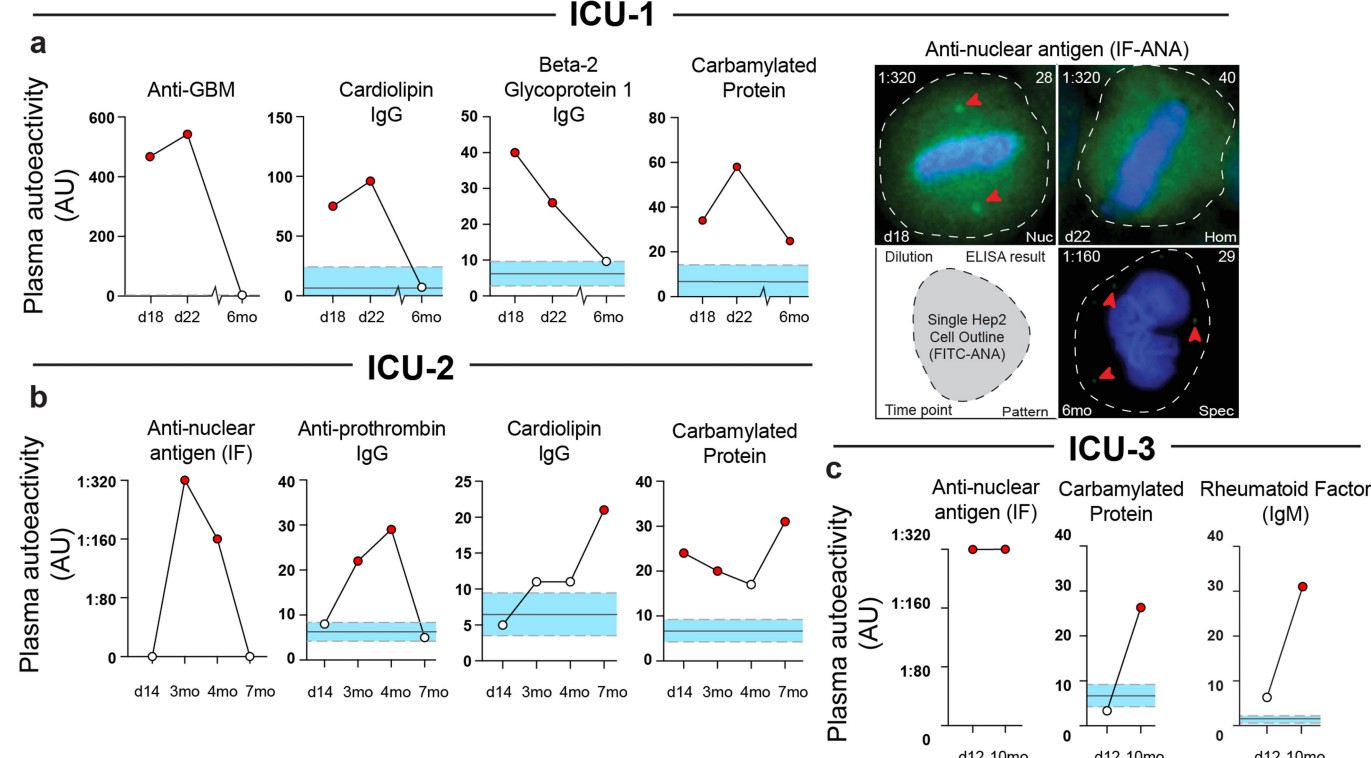

**Extended Data Fig. 10 | Longitudinal clinical autoreactivity profiles of patients ICU1-3. (a–c)** Samples obtained at all time points from patients ICU-1:3 were sent to Exagen, Inc. for broad autoreactivity testing in their clinical laboratory. All clinical positive tests for each patient are displayed. Red dots indicate a positive clinical value. (**a**) Clinical positive tests for patient ICU-1 at indicated time points. (**b**) Clinical positive tests for patient ICU-2 at indicated time points. (**c**) Clinical positive tests for patient ICU-3 at indicated time points.

# Reporting Summary

## Statistics

For all statistical analyses, confirm that the following items are present in the figure legend, table legend, main text, or Methods section.

| n/a | Confirmed | |
|---|---|---|
| ☐ | ☒ | The exact sample size (*n*) for each experimental group/condition, given as a discrete number and unit of measurement |
| ☐ | ☒ | A statement on whether measurements were taken from distinct samples or whether the same sample was measured repeatedly |
| ☐ | ☒ | The statistical test(s) used AND whether they are one- or two-sided<br>*Only common tests should be described solely by name; describe more complex techniques in the Methods section.* |
| ☒ | ☐ | A description of all covariates tested |
| ☒ | ☐ | A description of any assumptions or corrections, such as tests of normality and adjustment for multiple comparisons |
| ☐ | ☒ | A full description of the statistical parameters including central tendency (e.g. means) or other basic estimates (e.g. regression coefficient) AND variation (e.g. standard deviation) or associated estimates of uncertainty (e.g. confidence intervals) |
| ☐ | ☒ | For null hypothesis testing, the test statistic (e.g. *F*, *t*, *r*) with confidence intervals, effect sizes, degrees of freedom and *P* value noted<br>*Give P values as exact values whenever suitable.* |
| ☒ | ☐ | For Bayesian analysis, information on the choice of priors and Markov chain Monte Carlo settings |
| ☒ | ☐ | For hierarchical and complex designs, identification of the appropriate level for tests and full reporting of outcomes |
| ☐ | ☒ | Estimates of effect sizes (e.g. Cohen's *d*, Pearson's *r*), indicating how they were calculated |

*Our web collection on statistics for biologists contains articles on many of the points above.*

## Software and code

Policy information about availability of computer code

| | |
|---|---|
| Data collection | Patient data was collected using RedCAP data capture software. Flow cytometry data was collected on a Cytek Aurora flow cytometer using Cytek SpectroFlo software. Luminex data (including antigen specific data) was analyzed using a Luminex FLEXMAP 3D® instrument (Luminex; Austin, TX, USA) running xPonent 4.3 software. Repertoire data was sequenced by Novogene, and then processed through the 10x VDJ repertoire pipeline. Resulting sequences of high confidence were then mapped using IMGT's V-quest B cell receptor mapping software. |
| Data analysis | Computational analysis was carried out in R (v3.6.2; release 12 Dec 2019). Heat maps were generated using the pheatmap library (v1.0.12), with data pre-normalized (log-transformed z-scores calculated per feature) before plotting. Custom plotting, such as mutation frequency violin plots, was performed using the ggplot2 library for base analysis, and then post-processed in Adobe Illustrator. Alluvial plotting was performed using the ggalluvial package with post-processing in Adobe Illustrator. Clonotype connectivity analysis was carried out using the R-based 'vegan' package, and then visualized through 'pheatmap' before post-processing in Adobe Illustrator. Statistical analyses were performed directly in R, or in GraphPad Prism (v8.2.1).<br><br>Analyses on the single cell VDJ annotated sequences were performed using the Immcantation tool suite (http://www.immcantation.org) version 4.1.0 pipeline in Docker. This suite contains SHazaM for statistical analysis of somatic hypermutation (SHM) patterns as described in (Gupta et al., 2015), and BASELINe (Bayesian estimation of Antigen-driven SELectIoN) for analysis of selection pressure as described in (Yaari et al., 2012). Visualizations were generated in R using the SHazaM package (version 1.0.2) and then post-processed in Adobe Illustrator. |

For manuscripts utilizing custom algorithms or software that are central to the research but not yet described in published literature, software must be made available to editors and reviewers. We strongly encourage code deposition in a community repository (e.g. GitHub). See the Nature Portfolio guidelines for submitting code & software for further information.

## Data

Policy information about availability of data

All manuscripts must include a data availability statement. This statement should provide the following information, where applicable:

- Accession codes, unique identifiers, or web links for publicly available datasets
- A description of any restrictions on data availability
- For clinical datasets or third party data, please ensure that the statement adheres to our policy

All FCM and sequencing data presented here are publicly available in alignment with current requirements for public disclosure before peer review. All FCM data presented and analyzed in this manuscript (Fig. 1) are publicly available in the FlowRepository at http://flowrepository.org/id/FR-FCM-Z2XF/.

# Field-specific reporting

Please select the one below that is the best fit for your research. If you are not sure, read the appropriate sections before making your selection.

☐ Life sciences ☐ Behavioural & social sciences ☐ Ecological, evolutionary & environmental sciences

For a reference copy of the document with all sections, see nature.com/documents/nr-reporting-summary-flat.pdf

# Life sciences study design

All studies must disclose on these points even when the disclosure is negative.

| | |
|---|---|
| Sample size | *Describe how sample size was determined, detailing any statistical methods used to predetermine sample size OR if no sample-size calculation was performed, describe how sample sizes were chosen and provide a rationale for why these sample sizes are sufficient.* |
| Data exclusions | *Describe any data exclusions. If no data were excluded from the analyses, state so OR if data were excluded, describe the exclusions and the rationale behind them, indicating whether exclusion criteria were pre-established.* |
| Replication | *Describe the measures taken to verify the reproducibility of the experimental findings. If all attempts at replication were successful, confirm this OR if there are any findings that were not replicated or cannot be reproduced, note this and describe why.* |
| Randomization | *Describe how samples/organisms/participants were allocated into experimental groups. If allocation was not random, describe how covariates were controlled OR if this is not relevant to your study, explain why.* |
| Blinding | *Describe whether the investigators were blinded to group allocation during data collection and/or analysis. If blinding was not possible, describe why OR explain why blinding was not relevant to your study.* |

# Behavioural & social sciences study design

All studies must disclose on these points even when the disclosure is negative.

| | |
|---|---|
| Study description | *Briefly describe the study type including whether data are quantitative, qualitative, or mixed-methods (e.g. qualitative cross-sectional, quantitative experimental, mixed-methods case study).* |
| Research sample | *State the research sample (e.g. Harvard university undergraduates, villagers in rural India) and provide relevant demographic information (e.g. age, sex) and indicate whether the sample is representative. Provide a rationale for the study sample chosen. For studies involving existing datasets, please describe the dataset and source.* |
| Sampling strategy | *Describe the sampling procedure (e.g. random, snowball, stratified, convenience). Describe the statistical methods that were used to predetermine sample size OR if no sample-size calculation was performed, describe how sample sizes were chosen and provide a rationale for why these sample sizes are sufficient. For qualitative data, please indicate whether data saturation was considered, and what criteria were used to decide that no further sampling was needed.* |
| Data collection | *Provide details about the data collection procedure, including the instruments or devices used to record the data (e.g. pen and paper, computer, eye tracker, video or audio equipment) whether anyone was present besides the participant(s) and the researcher, and whether the researcher was blind to experimental condition and/or the study hypothesis during data collection.* |
| Timing | *Indicate the start and stop dates of data collection. If there is a gap between collection periods, state the dates for each sample cohort.* |
| Data exclusions | *If no data were excluded from the analyses, state so OR if data were excluded, provide the exact number of exclusions and the rationale behind them, indicating whether exclusion criteria were pre-established.* |
| Non-participation | *State how many participants dropped out/declined participation and the reason(s) given OR provide response rate OR state that no participants dropped out/declined participation.* |

| Randomization | *If participants were not allocated into experimental groups, state so OR describe how participants were allocated to groups, and if allocation was not random, describe how covariates were controlled.* |
|---|---|

# Ecological, evolutionary & environmental sciences study design

All studies must disclose on these points even when the disclosure is negative.

| Study description | *Briefly describe the study. For quantitative data include treatment factors and interactions, design structure (e.g. factorial, nested, hierarchical), nature and number of experimental units and replicates.* |
|---|---|
| Research sample | *Describe the research sample (e.g. a group of tagged Passer domesticus, all Stenocereus thurberi within Organ Pipe Cactus National Monument), and provide a rationale for the sample choice. When relevant, describe the organism taxa, source, sex, age range and any manipulations. State what population the sample is meant to represent when applicable. For studies involving existing datasets, describe the data and its source.* |
| Sampling strategy | *Note the sampling procedure. Describe the statistical methods that were used to predetermine sample size OR if no sample-size calculation was performed, describe how sample sizes were chosen and provide a rationale for why these sample sizes are sufficient.* |
| Data collection | *Describe the data collection procedure, including who recorded the data and how.* |
| Timing and spatial scale | *Indicate the start and stop dates of data collection, noting the frequency and periodicity of sampling and providing a rationale for these choices. If there is a gap between collection periods, state the dates for each sample cohort. Specify the spatial scale from which the data are taken* |
| Data exclusions | *If no data were excluded from the analyses, state so OR if data were excluded, describe the exclusions and the rationale behind them, indicating whether exclusion criteria were pre-established.* |
| Reproducibility | *Describe the measures taken to verify the reproducibility of experimental findings. For each experiment, note whether any attempts to repeat the experiment failed OR state that all attempts to repeat the experiment were successful.* |
| Randomization | *Describe how samples/organisms/participants were allocated into groups. If allocation was not random, describe how covariates were controlled. If this is not relevant to your study, explain why.* |
| Blinding | *Describe the extent of blinding used during data acquisition and analysis. If blinding was not possible, describe why OR explain why blinding was not relevant to your study.* |

Did the study involve field work? ☐ Yes ☐ No

## Field work, collection and transport

| Field conditions | *Describe the study conditions for field work, providing relevant parameters (e.g. temperature, rainfall).* |
|---|---|
| Location | *State the location of the sampling or experiment, providing relevant parameters (e.g. latitude and longitude, elevation, water depth).* |
| Access & import/export | *Describe the efforts you have made to access habitats and to collect and import/export your samples in a responsible manner and in compliance with local, national and international laws, noting any permits that were obtained (give the name of the issuing authority, the date of issue, and any identifying information).* |
| Disturbance | *Describe any disturbance caused by the study and how it was minimized.* |

# Reporting for specific materials, systems and methods

We require information from authors about some types of materials, experimental systems and methods used in many studies. Here, indicate whether each material, system or method listed is relevant to your study. If you are not sure if a list item applies to your research, read the appropriate section before selecting a response.

## Materials & experimental systems

| n/a | Involved in the study |
|---|---|
| ☐ | ☒ Antibodies |
| ☒ | ☐ Eukaryotic cell lines |
| ☒ | ☐ Palaeontology and archaeology |
| ☒ | ☐ Animals and other organisms |
| ☐ | ☒ Human research participants |
| ☒ | ☐ Clinical data |
| ☒ | ☐ Dual use research of concern |

## Methods

| n/a | Involved in the study |
|---|---|
| ☐ | ☐ ChIP-seq |
| ☐ | ☒ Flow cytometry |
| ☐ | ☐ MRI-based neuroimaging |

# Antibodies

| | |
|---|---|
| Antibodies used | Target; Fluorophore; Panel; Clone; Vendor; Cat#; Dilution (ul/100ul)<br>CD62L BV480 v1, v2 DREG-56 BD 566174 5 ul<br>CD86 PerCP-Cy5.5 v1, v2 IT2.2 Biolegend 305419 5 ul<br>CD27 BV750 v1, v2, ICS O323 Biolegend 302849 2.5 ul<br>CD19 BV570 v1, v2, ICS HIB19 Biolegend 302235 2.5 ul<br>CD45 Spark NIR 685 v2 2D1 Biolegend 368552 1.25 ul<br>CD1c BV510 v2 L161 Biolegend 331534 1.25 ul<br>IgM BV711 v1, v2, ICS MHM-88 Biolegend 314539 1.25 ul<br>CXCR3 A647 v1, v2, ICS G025H7 Biolegend 353711 1.25 ul<br>CXCR4 PerCP-e710 v1, v2 12G5 eBioscience 46-9999-41 1.25 ul<br>CCR7 A488 v1 G043H7 Biolegend 353205 1.25 ul<br>CD24 PerCP v1, v2, ICS ML5 Biolegend 311113 1.25 ul<br>CD3 BUV 805 v1, v2, ICS UCHT1 BD 612896 0.6 ul<br>CD11c APC-Fire750 v1, v2, ICS S-HCL-3 Biolegend 371509 0.6 ul<br>CD138 APC-R700 v1, v2 MI15 BD 566051 0.6 ul<br>HLA-DR BV650 v1, v2 L243 Biolegend 307649 0.6 ul<br>CD95 BV785 v1, v2 DX2 Biolegend 305645 0.6 ul<br>CD14 BUV805 v1, v2 M5E2 BD 612902 0.6 ul<br>CD23 APC v2 EBVCS-5 Biolegend 338514 0.3 ul<br>CD69 BUV 737 v1, v2 FN50 BD 612817 0.3 ul<br>IgD BV605 v1, v2, ICS IA6-2 Biolegend 348231 0.3 ul<br>CD21 PE-Dazzle594 v1, v2, ICS Bu32 Biolegend 354921 0.3 ul<br>CD38 BB515 v1, v2, ICS HIT2 BD 564499 0.3 ul<br>CXCR5 PE v1, v2, ICS J252D4 Biolegend 356903 0.3 ul<br>CD40 A532 v1, v2 5C3 Novus NBP1-43416AF523 0.3 ul<br>PD-1 PE-Cy7 v1, v2 EH12.2H7 Biolegend 239917 0.3 ul<br>IgG BV421 v1, v2 M1310G05 Biolegend 410703 0.15 ul<br>CD10 PE-Cy5 v1, v2 HI10a Biolegend 312205 0.15 ul<br>CD25 e450 v1 BC96 eBioscience 48-0259-41 5 ul<br>CD1d BV510 v1 51.1 Biolegend 350313 2.5 ul<br>ICOS-L APC v1 2D3 Biolegend 309407 5 ul<br>B220 Spark NIR 685 v1 RA3-6B2 Biolegend 103268 2.5 ul<br>T-bet APC ICS 4B10 Biolegend 644814 1.25 ul<br>Viability Zombie NIR v1,2 NA Biolegend 423106 0.2 ul |
| Validation | All antibodies have been validated by the manufacturer for use in targeting human proteins as indicated above. |

# Human research participants

Policy information about studies involving human research participants

| | |
|---|---|
| Population characteristics | Population characteristics are fully described in Supplementary table 1 of the manuscript. |
| Recruitment | Written informed consent was obtained from all participants or, if they were unable to provide informed consent, obtained from designated healthcare surrogates. Healthy donors (n = 36) were recruited using promotional materials approved by the Emory University Institutional Review Board. Subjects with COVID-19 (n = 19) were recruited from Emory University Hospital, Emory University Hospital Midtown and Emory St. Joseph's Hospital, all in Atlanta, GA, USA. All non-healthy donor subjects were diagnosed with COVID-19 by PCR amplification of SARS-CoV-2 viral RNA obtained from nasopharyngeal or oropharyngeal swabs. Subjects with COVID-19 were included in the study if they were 18 to 80 years of age, not immunocompromised, and had not been given oral or intravenous corticosteroids within the preceding 14 days. |
| Ethics oversight | All research was approved by the Emory University Institutional Review Board (Emory IRB numbers IRB00058507, IRB00057983, and IRB00058271) and was performed in accordance with all relevant guidelines and regulations. |

Note that full information on the approval of the study protocol must also be provided in the manuscript.

# ChIP-seq

## Data deposition

☐ Confirm that both raw and final processed data have been deposited in a public database such as GEO.

☐ Confirm that you have deposited or provided access to graph files (e.g. BED files) for the called peaks.

| | |
|---|---|
| Data access links<br>*May remain private before publication.* | *For "Initial submission" or "Revised version" documents, provide reviewer access links. For your "Final submission" document, provide a link to the deposited data.* |
| Files in database submission | *Provide a list of all files available in the database submission.* |

| Genome browser session (e.g. UCSC) | *Provide a link to an anonymized genome browser session for "Initial submission" and "Revised version" documents only, to enable peer review. Write "no longer applicable" for "Final submission" documents.* |
|---|---|

## Methodology

| Replicates | *Describe the experimental replicates, specifying number, type and replicate agreement.* |
|---|---|
| Sequencing depth | *Describe the sequencing depth for each experiment, providing the total number of reads, uniquely mapped reads, length of reads and whether they were paired- or single-end.* |
| Antibodies | *Describe the antibodies used for the ChIP-seq experiments; as applicable, provide supplier name, catalog number, clone name, and lot number.* |
| Peak calling parameters | *Specify the command line program and parameters used for read mapping and peak calling, including the ChIP, control and index files used.* |
| Data quality | *Describe the methods used to ensure data quality in full detail, including how many peaks are at FDR 5% and above 5-fold enrichment.* |
| Software | *Describe the software used to collect and analyze the ChIP-seq data. For custom code that has been deposited into a community repository, provide accession details.* |

# Flow Cytometry

## Plots

Confirm that:

☒ The axis labels state the marker and fluorochrome used (e.g. CD4-FITC).

☒ The axis scales are clearly visible. Include numbers along axes only for bottom left plot of group (a 'group' is an analysis of identical markers).

☒ All plots are contour plots with outliers or pseudocolor plots.

☒ A numerical value for number of cells or percentage (with statistics) is provided.

## Methodology

| Sample preparation | Peripheral blood samples were collected in heparin sodium tubes and processed within 6 hours of collection. PBMCs were isolated by density gradient centrifugation at 1000 x g for 10 minutes. Aliquots from the plasma layer were collected and stored at -80C until use. PBMCs were washed 2 times with RPMI at 500 x g for 5 minutes. |
|---|---|
| Instrument | Cells were analyzed on a Cytek Aurora flow cytometer (V3; 16V-14B-10YG-8R) |
| Software | Cells were analyzed on a Cytek Aurora flow cytometer using Cytek SpectroFlo software. Up to 3 x 106 cells were analyzed using FlowJo v10 (Treestar) software. |
| Cell population abundance | NA |
| Gating strategy | Gating strategy is provided in supplementary figure 1. |

☒ Tick this box to confirm that a figure exemplifying the gating strategy is provided in the Supplementary Information.

# Magnetic resonance imaging

## Experimental design

| Design type | *Indicate task or resting state; event-related or block design.* |
|---|---|
| Design specifications | *Specify the number of blocks, trials or experimental units per session and/or subject, and specify the length of each trial or block (if trials are blocked) and interval between trials.* |
| Behavioral performance measures | *State number and/or type of variables recorded (e.g. correct button press, response time) and what statistics were used to establish that the subjects were performing the task as expected (e.g. mean, range, and/or standard deviation across subjects).* |

## Acquisition

| | |
|---|---|
| Imaging type(s) | *Specify: functional, structural, diffusion, perfusion.* |
| Field strength | *Specify in Tesla* |
| Sequence & imaging parameters | *Specify the pulse sequence type (gradient echo, spin echo, etc.), imaging type (EPI, spiral, etc.), field of view, matrix size, slice thickness, orientation and TE/TR/flip angle.* |
| Area of acquisition | *State whether a whole brain scan was used OR define the area of acquisition, describing how the region was determined.* |

Diffusion MRI  ☐ Used  ☐ Not used

## Preprocessing

| | |
|---|---|
| Preprocessing software | *Provide detail on software version and revision number and on specific parameters (model/functions, brain extraction, segmentation, smoothing kernel size, etc.).* |
| Normalization | *If data were normalized/standardized, describe the approach(es): specify linear or non-linear and define image types used for transformation OR indicate that data were not normalized and explain rationale for lack of normalization.* |
| Normalization template | *Describe the template used for normalization/transformation, specifying subject space or group standardized space (e.g. original Talairach, MNI305, ICBM152) OR indicate that the data were not normalized.* |
| Noise and artifact removal | *Describe your procedure(s) for artifact and structured noise removal, specifying motion parameters, tissue signals and physiological signals (heart rate, respiration).* |
| Volume censoring | *Define your software and/or method and criteria for volume censoring, and state the extent of such censoring.* |

## Statistical modeling & inference

| | |
|---|---|
| Model type and settings | *Specify type (mass univariate, multivariate, RSA, predictive, etc.) and describe essential details of the model at the first and second levels (e.g. fixed, random or mixed effects; drift or auto-correlation).* |
| Effect(s) tested | *Define precise effect in terms of the task or stimulus conditions instead of psychological concepts and indicate whether ANOVA or factorial designs were used.* |

Specify type of analysis:  ☐ Whole brain  ☐ ROI-based  ☐ Both

| | |
|---|---|
| Statistic type for inference<br>(See Eklund et al. 2016) | *Specify voxel-wise or cluster-wise and report all relevant parameters for cluster-wise methods.* |
| Correction | *Describe the type of correction and how it is obtained for multiple comparisons (e.g. FWE, FDR, permutation or Monte Carlo).* |

## Models & analysis

| n/a | Involved in the study |
|---|---|
| ☐ | ☐ Functional and/or effective connectivity |
| ☐ | ☐ Graph analysis |
| ☐ | ☐ Multivariate modeling or predictive analysis |

| | |
|---|---|
| Functional and/or effective connectivity | *Report the measures of dependence used and the model details (e.g. Pearson correlation, partial correlation, mutual information).* |
| Graph analysis | *Report the dependent variable and connectivity measure, specifying weighted graph or binarized graph, subject- or group-level, and the global and/or node summaries used (e.g. clustering coefficient, efficiency, etc.).* |
| Multivariate modeling and predictive analysis | *Specify independent variables, features extraction and dimension reduction, model, training and evaluation metrics.* |

