## [Peer Review File · Nature]

Manuscript Title: Relaxed peripheral tolerance drives broad de novo autoreactivity in severe COVID-19

Reviewer Comments & Author Rebuttals

Reviewer Reports on the Initial Version:

Referees' comments:

Referee #1 (Remarks to the Author):

In this study from Woodruff et al., the authors have focused on antibody secreting cells in patients with severe COVID-19 and their association with an extra-follicular B cell response. They have examined the repertoire of these cells and focused on the propensity of some of these cells to generate autoantibodies.

Viral (and other) infections have long been known to lead to autoantibody generation. One of the landmark studies that aimed at elucidating the underlying basis of this phenomenon was published by Rolf Zinkernagel in 2002 (PMID: 12627229), and there have been many other studies showing that viral infections lead to a break in tolerance, but the model from Zinkernagel remains the only one with strong published support from multiple groups. These models are neither considered nor discussed in this manuscript

Sanz and colleagues (and others) have shown in earlier studies that extrafollicular B cells may contribute to autoantibody generation in autoimmune diseases. There have been a number of reports describing a plethora of autoantibodies in acute COVID-19, and also a few reports about the activation of virus-specific extrafollicular B cells. These authors previously made a useful contribution (reference 8) showing that extrafollicular B cell profiles and repertoires in severe COVID-19 were similar to those seen in autoimmunity. Based on their previous work and other studies in the field it has already been frequently inferred that extrafollicular B cell activation in infectious settings and autoimmunity breaks tolerance. This is in contrast to germinal center B cell responses where presumably tolerance checkpoints are generally functional and apoptosis and failure of selection by self-antigen may be major mechanisms by which self-reactive post-GC B cells fail to accumulate.

The studies presented in this current manuscript are technically solid and well executed. Extensive studies have been undertaken, ASCs have been more comprehensively looked at and some new information has certainly been provided. This study would have been far more valuable and interesting had the authors directly attempted to adequately consider and test at least one of the two major models that exist in the field. Is the model from Zinkernagel, already supported by data in the mouse LCMV context, wherein auto-reactive B cells promiscuously present viral antigens to T cells, also applicable to COVID-19? Very likely it is and there is little reason to believe this is not the case, but this really was not tested, though it could have been. The alternative major model is molecular mimicry. The one example provided in this manuscript that makes the likely rare case for molecular mimicry is not well developed.

A related question that arises and which I do not expect these authors to address is - why do only a very small fraction of COVID-19 patients exhibit clinically relevant and persistent auto-antibody responses?

Some specific concerns:

1. In Figure 1, the point is made that some ASCs downregulate IgG and so antigen specific BCR capture does not accurately capture all ASCs. ASCs are derived from activated B cells and overall, the results in the figure are consistent with what is already known in the field.

In Figure 1C, a correlation is shown between the fraction of B cells that are DN2 B cells and the fraction of cells that are ASCs. While these results suggest a connection between (essentially activated) extra-follicular B cells and ASCs, isn't it likely that if SARS-CoV-2 antigen-specific memory B cells had been gated on that they would also correlate with ASC numbers?

2. In Figure 1D, using the terminology "GC B cells" is problematic and should be avoided. Yes, most of the memory B cells studied here are likely GC derived. The authors however seem to discount long-established evidence that class switching and some SHM can occur in the absence of germinal centers as established by Takemori, Rajewsky and others (PMID 24610726, 23027924, 12354385), and that switched memory B cells can often be extrafollicular in origin. This is likely to be the case for antigen-specific memory B cells in COVID-19 and the authors do not assess these cells in their studies. The comparisons are likely largely being made between mainly virus-specific ASCs that have been compared to a large variety of historically accumulated memory B cells. Nevertheless, the point that extrafollicular B cell responses dominate in COVID-19 is well taken – but again this has been established before.

3. In Figure 2, the comparisons are between 3 healthy controls and 7 severe COVID-19 patients. The authors emphasize that ASCs in severe COVID-19 are largely IgG1+ cells; it is known that virus specific responses in this disease are biased towards the most prominent IgG isotype, IgG1. Given that circulating ASCs from healthy controls will likely largely reflect responses to commensal microbes at mucosal sites, while ASCs in severely ill COVID-19 patients disproportionately reflect an immune response to an acute viral infection that is known to cause viremia, the shift towards the most prominent IgG isotype in the context of an acute and severe viral infection (and this is already known from other studies) is not particularly remarkable. Clearly some non-IgG1 class switched ASCs are also likely to be virus-specific, so the other class switched cells are possibly just relatively more mucosal commensal specific and these distinctions are not air-tight. The increase in IgHV4-34 cells in the recently activated pool is interesting but given the breadth of antigens recognized by antibodies containing this VH gene it is not too surprising that there might be some expansion in COVID-19 of B cells that rearrange this V gene.

4. There are over a dozen reports that SHM is very limited in activated SARS-CoV-2 antigen-specific B cells including plasmablasts. So, while the SHM results in Figure 3 are also largely to be expected and further confirm that extrafollicular B cells selectively expand in COVID-19, the inclusion of and comparison with memory B cell SHM responses should be questioned. ASCs in the circulation are

likely to represent recently activated cells. For the comparison with memory B cells to be “fair” the authors should have used appropriately labeled probes for a few SARS-CoV-2 antigens and purified antigen-specific memory B cells. The comparison made in Figure 3 between ASCs and memory B cells are not really meaningful.

5. The AVY patch comparisons in Figure 3c are again biased, though they are consistent with the fact that the majority of accumulated memory B cells against a very wide range of antigens likely largely emanated from germinal centers and in these cells the AVY patch in the fairly promiscuous rearranged VH4-34 gene is likely to be mutated.

6. SHM has been well established to be limited in activated antigen-specific B cells in COVID-19. Is it surprising then that the expansion of B cells to SARS-CoV-2 antigens in individuals infected for the first time or in non-vaccinated individuals are largely naïve B cell derived? The connectivity data nonetheless represents novel information, though it does not necessarily yield new insights.

7. What could enhance these studies? I have two suggestions. One example for bi-specificity comes from VH4-34 that is already overrepresented in the naïve B cell population and contributes to the binding of many different autoantigens. It is not too surprising that some BCRs containing rearranged VH4-34 find an epitope in one of the 29 proteins of SARS-CoV-2. While I do not believe that autoimmunity in COVID-19 will be significantly linked to molecular mimicry (see below) nevertheless it would be useful to fine map this epitope and establish whether or not a specific nucleocapsid-specific BCR that contains rearranged VH4-34 is truly autoreactive.

The other alternative would be to study a few specific autoreactive BCRs in detail -say against carbamylated proteins or GBM. Even though SHM is low in these BCRs, it is not zero. It is theoretically possible that the unmutated common ancestor (UCA) for each BCR remains autoreactive, and that possibility could be tested. It could be shown whether these UCAs do or do not interact with any known protein of SARS-CoV-2. These results would lend support to the Zinkernagel model and suggest that as for other pathogens, antigen overload can lead to auto-reactive naïve B cells capturing and presenting viral peptides non-specifically and thus induce a promiscuous T-dependent B cell response that triggers autoimmunity.

Shiv Pillai

Referee #2 (Remarks to the Author):

It is now increasingly recognized that aberrant immune responses to SARS-CoV-2 can be a major contributor to the pathogenesis that accompanies the development of severe disease in COVID-19

patients. The Sanz group has been at the forefront in demonstrating that one of the key features observed in many critically ill COVID-19 patients is the inappropriate activation of a population of T-bet driven B cells known to accumulate in SLE (which they term DN2s, are enriched in VH4-34, and share similarities to murine ABCs) as well as a marked expansion of ASCs. Another recently recognized aspect of SARS-CoV2 related pathogenesis is the observation that disease can be accompanied by the production of a wide-range of autoantibodies (from anti-PL Abs to anti-IFN Abs to Abs targeting the exoproteome) a feature that again preferentially accompanies severe disease. Given that DN2s are believed to differentiate into ASCs primarily via an EF route and to be major autoAb producers in SLE, the authors in this manuscript employ several elegant and novel approaches to extend their original analysis of COVID-19 patients described in their Nature Immunology manuscript to establish closer connections between these B cells subsets and the production of both antiviral and potentially pathogenic autoantibodies (primarily of the IgG1 isotype). In particular, their analysis aims to uncover the “origins, breadth, and resolution” of the transient autoreactivity that is observed in severe COVID-19.

While the production of autoantibodies has long been recognized to transiently accompany many infections in addition to COVID-19 (e.g. malaria, Chikungunya virus, hepatitis, Lyme disease, etc) and hence this particular aspect of the story is not as innovative, the impact and novelty of the manuscript lies in the in-depth analysis of the ASC compartment and the combined molecular and serological approaches utilized to understand the mechanism of the appearance of this phenomenon during COVID-19. Furthermore, some of the findings, such as the detection of anti-carbamylated antibodies in severe COVID-19 patients, could potentially have important clinical implications. Despite a great degree of enthusiasm for the work, however, the studies need to be strengthened in several areas.

1) While the authors take advantage of the patient cohort described in their Nature Immunology manuscript most of the studies deal with a very small (and at times extremely small) subset of these patients. Thus, some of their analysis could be profoundly skewed by the demographics and clinical characteristics of the subjects on which the in-depth analysis is performed. The authors should thus provide additional clinical information on the patients investigated in each of the figures. More detailed information about timing of the analysis in each figure would also be helpful to the reader. For instance, which ones are ICU-1 and ICU-2 in Fig 2a? How do their demographics compare to the 3 HD controls? Was there a change in the clinical course of the ICU-1 patient that might have contributed to the change in the ANA pattern from nuclear to homogenous in a 4-day span and the increase in other autoAb titers in Fig. 4d?

2) The finding of broad autoreactivity is consistent with other recent studies and obviously of great clinical interest. The ability of the investigators to link this finding to the EF-derived IgG1 ASCs is thus one of the most interesting aspects of the manuscript. However, while the investigators provide some evidence for these connections additional information should be included to further support this claim. Specifically,

a. The authors should assess whether the isotype of the plasma autoantibodies detected in their ICU COVID patients is indeed IgG1. This should be easily testable for at least some of the autoantibodies like the anti-CarP antibodies.

b. The presence of anti-CarP (and potentially anti-GBM) antibodies could be an important biomarker

not only of acute peripheral tolerance breaks but also of the extent of lung damage in these patients. However, the authors cannot exclude that presence of these antibodies could be preexisting in some of these patients based on their age, smoking status etc. Hence availability of (any) pre-infection samples would be very valuable to assess this possibility. One would furthermore presume that the ICU-COVID (but not the OUT-C) patients were on mechanical ventilation, which could be a key contributing feature to the findings. Thus, samples from non-COVID ICU patients on mechanical ventilation should be included as a control. Ideally one would also assess samples from intubated patients with other viral/bacterial infections to evaluate whether these findings are specific for COVID-19 patients. Again providing additional information on the clinical course of these patients (e.g. did they have any evidence of acute renal damage?) would be very helpful.

c. While the authors do attempt to broaden their findings to an additional cohort of patients, this analysis, as they admit, primarily included clinically validated autoantibodies like ANA and RF (which is often an IgM and seems to be the main difference between high and low CRP patients) and thus falls short of supporting their key claim that this autoreactivity is linked to IgG1 producing EF-derived ASCs. Given the very small number of ICU-COVID patients in which their in-depth analysis was conducted, gaining additional support for their main findings (e.g. by assessing the production/kinetics of IgG1 autoantibodies) in additional cohorts of COVID-19 patients (which unfortunately are not lacking) of different disease severity is very important.

3) More experimental details need to be provided throughout the manuscript. In particular the authors utilize BASELINE to assess the CDR selective pressure and state that they observe a selective reduction in IgG1 but not other class switched compartments. However, in Fig 2h IgG2 in the ICU cohort also seems to be diminished. Can they provide the specific parameters utilized to determine the significance of their findings? How many cells was this analysis based on for the various isotypes in HD versus ICU-COVID patients? Were the differences statistically significant? Given that this analysis is one of their key findings and is being utilized in several figures, this information would help the reader better assess the different comparisons that they perform. Was any in-depth analysis conducted in the non-ICU COVID cases?

4) The comparison of the differences in break in tolerance between those in COVID-19 and the ones commonly observed in autoimmune patients is very intriguing. As the authors and others have shown, the key signals driving the expansion of these cells in autoimmune disease are TLR7 engagement as well as the presence of T cell cytokines like IL-21 and IFN γ . Could the lack of TFH cells, and/or the presence of antibodies targeting these cytokines in COVID-19 patients limit the T cell help that these cells can receive and make them rely primarily on TLR7 driven signals? Could this contribute to the transient and more limited nature of these autoreactive responses and be an important difference with autoimmune disease? In this regard are the anti-CarP antibodies in ICU-3 that increase during the recovery period still predominantly IgG1 or do they include different IgG isotypes?

5) Is there any benefit from this relaxed break of tolerance? Could these cross-reactive antibodies serve as a rapid response to control infection with autoimmunity being a "side effect" of rapidly responding? Could this break in tolerance be beneficial for host survival? The authors find 70% of the EF IgG responses to be anti-viral, do these also wane?

Minor comments:

1) While one recognizes the challenges of investigating the ASC compartment in COVID-19 patients, these cells are notoriously fragile and might be affected by freezing the samples (which are being utilized for all of the studies). Have the authors evaluated whether there is any differential loss of Ig producing capabilities upon freezing/thawing of samples in their MENSA assay by comparing fresh versus frozen samples (which could be conducted in samples from HD subjects)?

2) In the legend of Fig. 1 the g and h panels are reversed.

Referee #3 (Remarks to the Author):

A. Summary of the key results

In this manuscript, the authors studied the evolution of BCR specificity during severe COVID-19. This report builds from previous findings published by this group and others^{1,2}, which showed that severe COVID-19 is characterized by an exaggerated extrafollicular B cell and ASC response and expanded DN2 B cells previously shown to contribute to autoimmunity in SLE³. In this work, the authors again show that severe COVID-19 patients display an expansion of IgG1 ASC and EF B cells, and that these IgG1 ASC abundantly secrete RBD specific IgG. Again, similar to their previous work¹, the authors show that expanded IgG1 ASC display decreased SHM and elevated usage of autoreactive IGHV4-34 with preservation of the FR1 patch. Building off of this, the authors show that these low SHM IgG1 ASC are derived from the naïve B cell compartment as opposed to the memory compartment. From data derived from a single patient, authors show these IgG1 ASCs are polyreactive, specific for both SARS-CoV2 antigens and intracellular and extracellular self-antigens. In two separate larger cohorts, the authors show a correlation between severe disease and inflammatory markers with presence of select autoantibodies against, most notably, carbamylated proteins and nuclear antigens. Finally, one of the most interesting findings of this paper is the contraction and disappearance of expanded low SHM IgG1 ASC, exclusion of these clones from the memory B cell compartment (in one patient), and decreases in certain autoantibody titers at 6 months after acute infection.

B. Originality and significance

Several findings presented in the manuscript, particularly in figures 1 and 2, are not novel or particularly differentiated from the authors' previous report¹. For example, both papers show that severe COVID-19 is characterized by a dominance of EF B cells and expansion of IgG1 ASC with decreased SHM. Both papers show that severe COVID-19 patients display elevated usage of IGHV4-34 with increased prevalence of the FR1 patch that is normally eliminated through SHM. Thus, major segments of the current paper are largely confirmatory of the previous work. While it is nice to show that the findings are generalizable and supported by multiple studies, it seems much of this data

could be moved to supplemental figures or supported by references.

Aspects of other points made in this paper, are somewhat novel, though not particularly unexpected based on preexisting data. For example, the elevated frequencies of autoantibodies against RF^{4,5}, nuclear antigen⁶, cardiolipin⁷, and extracellular proteins⁸ in COVID-19 patients have already been reported and, in some studies, shown to correlate with severity of infection^{6,8}. Findings showing antibodies against GBM, of which only isolated cases have been reported, and carbamylated proteins are novel and interesting.

Other aspects of the paper are novel, interesting, and important. For example, data in one patient showing IgG1 low SHM ASCs are polyspecific to self- and SARS-CoV-2 antigens provides an interesting and plausible bridge between the aberrant B cell abnormalities and the presence of extensive intracellular and extracellular autoantibodies in patients with severe disease. One major concern, however, is the generalizability of these findings, given that they are only shown in one individual. Second, and equally notable, the results showing contraction of the low SHM IgG1 ASC response and exclusion of these cells from the memory compartment is highly significant, novel, and perhaps the strongest aspect of this work. However, again some of these conclusions are based on 1 or 2 patients, making the generalizability of these findings uncertain.

C. Data & methodology: validity of approach, quality of data, quality of presentation

Most of the work here is performed using sound techniques, some of which are novel. A general criticism of the data presentation is that it is hard to follow exactly which and how many patient samples are being analyzed. Below are comments regarding the authors' approach:

1. One of the most novel claims of this work is that IgG1 antibody from low mutation frequency ASC are polyspecific for SARS-CoV2 and common autoantigens such as cardiolipin, nuclear antigen, B cells, and glomerular basement membrane, in addition to extracellular proteins enriched for immune and lung specificity (figure 4). Although this is an interesting and scientifically plausible finding, the results here were produced from a single patient without a control of either healthy donor(s) or patient(s) with mild COVID-19. Thus, it is not possible to conclude that the IgG1 ASC polyspecificity observed in this individual is representative of severe COVID-19 or whether this is even a unique phenomenon to severe vs mild COVID-19. The authors partially support the generalizability of the single cell polyreactivity findings by showing that elevated reactivity to RF and ANA is correlated with severe COVID-19 in two separate large cohorts. However, the elevated presence of these autoantibodies alone does not support the claims of their origin put forth in figure 4. At the bare minimum, this conclusion needs to be supported by additional severe COVID-19 patient(s) and mild COVID-19 control(s). Furthermore, the authors state in the paper text that they performed single cell BCR repertoire analysis on 4 severe COVID-19 patients. Why only one of them was chosen for figure 4 is unclear and needs to be further explained.

2. The CDI recombinant protein binding assays lack an appropriate negative control (antibodies derived from healthy donor(s) or mild COVID-19). While the findings here are thematically congruent with other reports, there is no way to assess whether the protein reactivities found in this single severe COVID-19 patient are unique or enriched in severe COVID-19. Similarly, there is no way to assess whether the enriched pathways identified using enrichment pathway analysis are enriched

relative to any reference. For example, the pathways identified as significant in this single sample may be biased by the specific recombinant proteins present in the analysis platform. Without a control as reference, this data lacks meaning or context.

3. Related to the point above, the CDI protein arrays are comprised of proteins expressed in the yeast cytoplasm as GST-fusions, yet many proteins highlighted (IFN α , CD49, L-selectin, SLAMF7) are secreted glycoproteins, raising questions about their fidelity in approximating the target antigen. As such, this array should only be used as a screening tool; interesting candidate autoantigens need to be confirmed with gold-standard ELISA or similar assays using validated recombinant proteins expressed from mammalian cells.

4. The authors state that inclusion criteria for their second autoantibody cohort (52 critically ill patients from Atlanta ICUs) specified patients that “had received autoantibody testing as part of routine clinical care at the discretion of their treating physicians.” These inclusion criteria may upwardly bias the prevalence of autoantibodies in this group, as these tests are not typically performed in the absence of a history of autoimmunity or a clinical presentation suggestive of an autoimmune process. The authors should provide more clinical details of this cohort to enable a determination of whether the disease course and medical history of this cohort is representative or skewed towards an autoimmune population.

D. Appropriate use of statistics and treatment of uncertainties

The majority of the work here is performed using appropriate statistical analysis. Below are minor points:

1. Please quantify findings in figures 3D and 3E. It is unclear what is going on here, particularly in ICU-1 in figure 3D; while the authors claim the IgG1 ASCs are predominantly coming from the CD27-naïve compartment, it appears by eye that the memory and CD27- compartments equally contribute. A numerical metric here would help to prove their claims.

2. The following figures lack statistical tests: 3A, 3C, 6D

E. Conclusions: robustness, validity, reliability

The following conclusions are well supported by the data:

1. Severe COVID-19 is characterized by dominant EF and IgG1 ASC populations with low mutation frequency and enriched usage of IGHV4-34.

2. Severe COVID-19 is characterized by elevated autoantibodies against RF, ANA, and Carbamylated proteins, and these are correlated with inflammatory markers such as CRP

While these conclusions are well supported, they lack significant novelty as discussed above.

The following conclusions are partially, but insufficiently supported by the data:

1. Antibodies from IgG1 low SHM ASC are polyreactive to SARS-CoV2 antigens and intracellular and extracellular autoantigens. While this data is interesting and plausible given previous reports showing elevated autoantibodies in severe COVID-19, the data is generated from a single patient and lacks an appropriate reference control, either mild COVID-19 or healthy individual(s). As such, definitive conclusions are not possible with the present data. For example, the authors speculate on the meaning of enriched pathways identified in enrichr pathway analysis as evidence that B cells develop autoreactive specificity for immediately available antigens in milieu of the blood/immune or lung compartments. While this is an interesting interpretation of the data, there is no solid evidence that these pathways are differentially enriched in severe COVID-19 relative to control.

2. Relaxed peripheral tolerance observed in acute disease resolves upon recovery. This finding is another highly interesting and novel aspect of the paper. It is confusing, why the authors flip between showing 4 patients versus selecting 2 of the 4 for display, when they state in the text that they followed 4 patients with single cell analysis. Data for all 4 patients should be at least available in the supplementary figures. Furthermore, while authors claim decreased IgG1 ASC and return of normal SHM levels represents reversal of relaxed peripheral tolerance observed in disease, they do not show full data on all autoantibodies tested during acute disease. This data is needed to complete the narrative of his paper regarding long term outcomes of relaxed B cell peripheral tolerance. Specifically, autoantibody persistence is an important indicator of whether this relaxed tolerance has temporary versus long term implications. For example, if there is chronic persistence of autoantibodies months after resolution of acute disease, this would suggest conversion of a subset of IgG1 ASC to long-lived plasma cells which could be present in niches (ie bone marrow) not sampled in the peripheral blood in this study.

3. Polyreactive IgG1 ASC are excluded from the memory compartment. This is an important and valuable conclusion with great implications as to whether COVID-19 may result in long term breaches in humoral tolerance. However, the only evidence for this is connectivity analysis showing the IgG1 clonotypes were underrepresented in the memory compartment at 6 months. While this is interesting, again this analysis is derived from only one patient and may not be representative of the average outcome after COVID-19. Given the tremendous heterogeneity in patient outcomes (e.g., some patients entirely recover whereas others develop diverse manifestations of post-acute sequelae of COVID-19, PASC), a more comprehensive evaluation across a varied spectrum of patients is warranted.

The following conclusions are not well supported by the data:

1. Anti-carbamylated protein antibodies as a biomarker for relaxed peripheral tolerance in acute Covid19. The results showing 40% of severe Covid19 patients display reactivity to carbamylated proteins are interesting. However, there are several factors that undermine the claim of carbamylated protein reactivity is a biomarker for loss of peripheral tolerance checkpoints in severe COVID-19: 1) the lack of these findings in a secondary validation cohort; 2) lack of receiver operator analysis for prediction of loss of peripheral tolerance checkpoints in another cohort; 3) lack of any

shown connection between presence of anti-carbamylated protein autoantibodies in figure 5 to loss of peripheral tolerance as shown in figures 2, 3, and 4. In other words: did the patients who had multiple autoreactivities and carbamylated protein reactivity in figure 5 also display expansions in low mutation IgG1 ASCs that the authors claim underly this relaxed peripheral tolerance? Also did they have antibodies of a similar profile to those found in the proteomics array of figure 4?

E. Suggested improvements: experiments, data for possible revision

Overall this paper puts forth interesting and highly relevant results regarding peripheral tolerance in severe COVID-19. These results bridge previous reports of aberrant B cell responses and autoantibody prevalence in COVID-19. However, the credibility of the claims made here suffer from the very low number of analyzed samples. For example, Figure 4, one of the most important and novel figures of the paper, has only one patient with no reference, making any results from this figure effectively uninterpretable. While results from figure 4 make sense given the current literature, the authors should include data from the other 3 patients whom they performed single cell BCR analysis on. If the single cell data was not sufficient (lack of cells, reads, etc), the authors should perform additional studies on new acute samples and controls.

Similarly, another key figure of the paper, figure 6, alternates inexplicably between including 1 patient, to 4 patients, to 2 patients. The authors should consistently show data for all 4 patients and not cherry-pick between them. (Parenthetically, even an n=4 patients seems like a fairly small number to derive sweeping conclusions.) The connectivity analysis showing that the IgG1 ASC compartment was underrepresented in the memory compartment at 6 months was another extremely relevant and exciting finding. However, again this analysis should be expanded to include more than 1 patient.

Additional experiments to connect the autoantibody data in figure 5 with the rest of the paper would also be additive. For example, it would be helpful to assess the persistence of all autoantibodies assayed at 6 months, and how this correlates with contraction and/or exclusion of the IgG1 ASC compartment.

References

1. Woodruff, M.C. et al. Extrafollicular B cell responses correlate with neutralizing antibodies and morbidity in COVID-19. *Nat Immunol* 21, 1506-1516 (2020).
2. Hoehn, K.B. et al. Cutting Edge: Distinct B Cell Repertoires Characterize Patients with Mild and Severe COVID-19. *J Immunol* (2021).
3. Jenks, S.A. et al. Distinct Effector B Cells Induced by Unregulated Toll-like Receptor 7 Contribute to Pathogenic Responses in Systemic Lupus Erythematosus. *Immunity* 49, 725-739.e726 (2018).
4. Xu, Chen, et al. "Prevalence and Characteristics of Rheumatoid-Associated Autoantibodies in Patients with COVID-19." *Journal of Inflammation Research* 14 (2021): 3123
5. Lingel, Holger, et al. "Unique autoantibody prevalence in long-term recovered SARS-CoV-2-infected individuals." *Journal of Autoimmunity* (2021): 102682.
6. Lerma, L. Angelica, et al. "Prevalence of autoantibody responses in acute coronavirus disease 2019 (COVID-19)." *Journal of translational autoimmunity* 3 (2020): 100073.

7. Hossri, Sami, et al. "Clinically significant anticardiolipin antibodies associated with COVID-19." *Journal of critical care* 59 (2020): 32-34.

8. Wang, Eric Y., et al. "Diverse functional autoantibodies in patients with COVID-19." *Nature* (2021): 1-6.

Author Rebuttals to Initial Comments:

To the Reviewers,

On behalf of the authors, I would like to extend our gratitude to the reviewers for their timely and detailed reviews. We are encouraged that all three found several aspects of the work to be important, novel, and worthy of publication. However, we also understand that the reviewers raised important concerns around pieces of the manuscript surrounding the inclusion of important controls, decisions around data analysis and presentation, and scope of discussion.

To address these concerns, the authors have substantially altered the manuscript to more fully substantiate its claims, and specifically address the concerns outlined by the reviewers below. In addition to general manuscript updates, clarifications, and reformatting, the authors have:

1. Consolidated figures 1 and 2 to avoid redundancy with previously published work and streamline the novel findings of the manuscript.
2. Added an entirely new cohort of mild/moderate patients with COVID-19 for single-cell analysis for the purposes of repertoire comparison (n = 4).
3. Added a new cohort of ICU-admitted patients with bacterial pneumonia-induced ARDS as controls (n = 28) for severe disease.
4. Expanded our numbers of ICU-COVID-19 patient groups (n = 50) for kinetic analysis of autoreactivity emergence.
5. Added a new cohort of ICU-recovered patients for kinetic analysis of autoreactivity contraction (n = 40).
6. Added a second set of antibodies from an independent patient to reinforce the manuscript's main cellular and molecular findings.
7. Significantly modified data analysis and presentation to resolve concerns that we have selectively included data.
8. Provided increased clarity surrounding the individual patients included in the study and their underlying demographics.
9. Expanded the scope of discussion to include important points brought up by the reviewers, and to address the expanded impact that the new data additions including the generalizability of these findings to other severe infections and the implications of this work in the emergence of post-acute sequelae of COVID-19 (PASC).
10. Proposed a title change to the manuscript to more appropriately highlight novel aspects of the work as pointed out by the reviewers.

In total, this revision represents a significant update of the initially submitted manuscript. Thanks to the reviewer's comments, we believe that it represents a substantially improved manuscript with greater scope than the initial submission. We have addressed each comment individually below, and provided references to the new manuscript to aid the reviewers in identifying and assessing the relevant manuscript alterations.

Again, we deeply appreciate the reviewer's time and consideration.

Regards,

Iñaki Sanz, M.D.
Mason Lowance Professor of Medicine and Pediatrics
Chief, Division of Rheumatology
Director, Lowance Center for Human Immunology
Point-by-point response

Author responses in bold

Referee #1 (Remarks to the Author):

Overall-

In this study from Woodruff et al., the authors have focused on antibody secreting cells in patients with severe COVID-19 and their association with an extra-follicular B cell response. They have examined the repertoire of these cells and focused on the propensity of some of these cells to generate autoantibodies.

Viral (and other) infections have long been known to lead to autoantibody generation. One of the landmark studies that aimed at elucidating the underlying basis of this phenomenon was published by Rolf Zinkernagel in 2002 (PMID: 12627229), and there have been many other studies showing that viral infections lead to a break in tolerance, but the model from Zinkernagel remains the only one with strong published support from multiple groups. These models are neither considered nor discussed in this manuscript.

Sanz and colleagues (and others) have shown in earlier studies that extrafollicular B cells may contribute to autoantibody generation in autoimmune diseases. There have been a number of reports describing a plethora of autoantibodies in acute COVID-19, and also a few reports about the activation of virus-specific extrafollicular B cells. These authors previously made a useful contribution (reference 8) showing that extrafollicular B cell profiles and repertoires in severe COVID-19 were similar to those seen in autoimmunity. Based on their previous work and other studies in the field it has already been frequently inferred that extrafollicular B cell activation in infectious settings and autoimmunity breaks tolerance. This is in contrast to germinal center B cell responses where presumably tolerance checkpoints are generally functional and apoptosis and failure of selection by self-antigen may be major mechanisms by which self-reactive post-GC B cells fail to accumulate.

The studies presented in this current manuscript are technically solid and well executed. Extensive studies have been undertaken, ASCs have been more comprehensively looked at and some new information has certainly been provided. This study would have been far more valuable and interesting had the authors directly attempted to adequately consider and test at least one of the two major models that exist in the field. Is the model from Zinkernagel, already supported by data in the mouse LCMV context, wherein auto-reactive B cells promiscuously present viral antigens to T cells, also applicable to COVID-19? Very likely it is and there is little reason to believe this is not the case, but this really was not tested, though it could have been. The alternative major model is molecular mimicry. The one example provided in this manuscript that makes the likely rare case for molecular mimicry is not well developed.

A related question that arises and which I do not expect these authors to address is - why do only a very small fraction of COVID-19 patients exhibit clinically relevant and persistent auto-antibody responses?

The authors appreciate the reviewer's perspective on the work and thank them for their thorough review. We are encouraged that the reviewer highlighted the work as "technically solid and well-executed", with "new information certainly ... provided".

The reviewer's general point is well-taken as we should have taken the opportunity in the initial submission to highlight the implications of our work on long-standing models of viral-induced autoreactivity – whether molecular mimicry or polyclonal B cell activation by promiscuous presentation of non-viral peptides to specific TH cells. In the latter model, it was proposed that the viral (LCMV-induced), autoreactivity, would result from the switch to IgG of autoreactive pre-immune IgM B cells. It should be noted however, that this model does not preclude a role for molecular mimicry as the latter property might still be at play in the process of selection and pathogenic targeting of self

tissues, whether or not the autoreactivities might have been initially induced through the polyclonal mechanism. In our work, we actually propose and present strong evidence for a model which might combine both mechanisms, or instead, be separate from either one. Our central finding laid out through the manuscript is the un-opposed expansion of autoreactive ASC from their naive precursors through an extrafollicular pathway devoid of tolerogenic checkpoints and which in most patients but possibly not all, appears to self-correct. Hence, we believe that the significance of the finding would stand irrespective of the relative contribution of the two mechanisms.

However, in highlighting what we believe are those central findings, we failed to highlight aspects of our study that directly address the concepts of molecular mimicry in COVID-19, and strongly argue against its dominance as the driver of induced autoantibody production in severe disease. This is a unique feature of our experimental approach, and it has now been bolstered by additional data using severe-disease controls, kinetic analyses of autoreactivity onset, and refined analysis of the (now expanded) monoclonal antibody testing. In light of these new data, and at the reviewer's suggestion, the discussion has been significantly altered to include a formal discussion of these models, and the implications of the current work on their interpretation.

Specific reviewer concerns are addressed below:

Some specific concerns:

1. In Figure 1, the point is made that some ASCs downregulate IgG and so antigen specific BCR capture does not accurately capture all ASCs. ASCs are derived from activated B cells and overall, the results in the figure are consistent with what is already known in the field.

We thank the reviewer for their comment, and generally agree that B cell receptor downregulation is commonly held as a feature of antibody-secreting cell responses. However, it is also a feature that has been periodically downplayed in its importance over the course of the emerging COVID-19 literature (Dugan et. al, 2021), and the details of this downregulation in human ASCs are, to our knowledge, not comprehensively documented – particularly within the domain of acute human antiviral response. In particular, the differences in downregulation between class-switched, versus unswitched ASCs is both less-well recognized, and also critical in understanding the antigen-specific B cell flow approaches that have been used to capture “total” antigen-specific B cell responses throughout the pandemic.

We understand, and do agree, that this is a minor contribution to this overall work presented here and, by itself, may not be overly novel in concept. However, we believe that its importance lies in the justification of the current study to outline how our approach differs from some of the existing literature, and in contextualizing and solidifying these general observations. As a result, while we have significantly altered Figure 1 (see below) to reduce redundancy with previously published work, we believe that this particular observation remains important enough to retain.

In Figure 1C, a correlation is shown between the fraction of B cells that are DN2 B cells and the fraction of cells that are ASCs. While these results suggest a connection between (essentially activated) extra-follicular B cells and ASCs, isn't it likely that if SARS-CoV-2 antigen-specific memory B cells had been gated on that they would also correlate with ASC numbers?

It is possible that antigen-specific memory might also correlate in these patients, yet not likely as significantly, as our own unpublished data and others indicate that, as it might be expected, the peaks of the memory response may happen later and are uncoupled from the peak of the plasmablast

response. Our previous analysis of these data showed no overall correlation with DN2 B cell expansion and (total) switch memory B cell predominance, with an overall *negative* correlation with unswitched B cell memory populations. Instead, the clearest activation correlates in these patients were between the ASC compartment and extrafollicular intermediates including DN2 B cells.

However, as the reviewer correctly points out below, these correlations have been established previously and may not be appropriate for inclusion as a main figure. We have moved this analysis to the supplement to continue to provide context for the nature of these ASCs responses, but streamline the manuscript towards the presentation of more novel, and ultimately more interesting data.

2. In Figure 1D, using the terminology “GC B cells” is problematic and should be avoided. Yes, most of the memory B cells studied here are likely GC derived. The authors however seem to discount long-established evidence that class switching and some SHM can occur in the absence of germinal centers as established by Takemori, Rajewsky and others (PMID 24610726, 23027924, 12354385), and that switched memory B cells can often be extrafollicular in origin. This is likely to be the case for antigen-specific memory B cells in COVID-19 and the authors do not assess these cells in their studies. The comparisons are likely largely being made between mainly virus-specific ASCs that have been compared to a large variety of historically accumulated memory B cells. Nevertheless, the point that extrafollicular B cell responses dominate in COVID-19 is well taken – but again this has been established before.

This point is well-taken. The use of the “GC B cell” terminology in this context is, indeed, problematic for the reasons that the reviewer indicates. As above, we hoped that this figure would provide context to the overall B cell responses observed in these patients. However, on review that the use of this metric here it is neither well defended nor particularly valuable to the overall work presented here. We have removed the panel from the manuscript and instead introduced the concepts by reference to previous work by our group and others.

3. In Figure 2, the comparisons are between 3 healthy controls and 7 severe COVID-19 patients. The authors emphasize that ASCs in severe COVID-19 are largely IgG1+ cells; it is known that virus specific responses in this disease are biased towards the most prominent IgG isotype, IgG1. Given that circulating ASCs from healthy controls will likely largely reflect responses to commensal microbes at mucosal sites, while ASCs in severely ill COVID-19 patients disproportionately reflect an immune response to an acute viral infection that is known to cause viremia, the shift towards the most prominent IgG isotype in the context of an acute and severe viral infection (and this is already known from other studies) is not particularly remarkable. Clearly some non-IgG1 class switched ASCs are also likely to be virus-specific, so the other class switched cells are possibly just relatively more mucosal commensal specific and these distinctions are not air-tight. The increase in IgHV4-34 cells in the recently activated pool is interesting but given the breadth of antigens recognized by antibodies containing this VH gene it is not too surprising that there might be some expansion in COVID-19 of B cells that rearrange this V gene.

Response to this, and the subsequent comment is combined and provided below.

4. There are over a dozen reports that SHM is very limited in activated SARS-CoV-2 antigen-specific B cells including plasmablasts. So, while the SHM results in Figure 3 are also largely to be expected and further confirm that extrafollicular B cells selectively expand in COVID-19, the inclusion of and comparison with memory B cell SHM responses should be questioned. ASCs in the circulation are likely to represent recently activated cells. For the comparison with memory B cells to be “fair” the authors should have used appropriately labeled probes for a few SARS-CoV-2 antigens and purified antigen-specific memory B cells. The comparison made in Figure 3 between ASCs and memory B cells are not really meaningful.

While we appreciate the reviewer's perspective, we respectfully disagree with their characterization of the emerging IgG1 ASC compartment as predictable, even if indeed, IgG1 dominance might be expected. The central feature of our work is the identification of a plastic IgG1 ASC that newly emerges early during the acute response, bears unique molecular signatures and enhanced autoreactivity (demonstrated by both VH4-34 usage and direct mAb analysis), and contracts after the resolution of the infection. Early on, the contribution of the early IgG1 ASC dominates the global IgG1 ASC compartment in contrast to the late IgG1 ASC that recovers the same mutational and selection features as other class-switched ASC whether during acute memory responses or at steady-state. Thus, we believe the focus on IgG1 ASC is both justified and essential for the model proposed.

We would also argue that it is quite novel. The reviewer is correct in pointing out that broad characterization of the B cell repertoire has been executed elsewhere in varying degrees of detail. However, in most studies, (Xuefeng et al, Galson et al, Nielson et al, etc.), B cell repertoire analysis has been carried out on total PBMCs, which fail to isolate contributions of specific B cell compartments to the overall response. In others (Sakharkar et al, Dugan et. al., etc.), reliance on antigen-specific staining would have eliminated the vast majority of ASC responses. Altogether, even studies containing sequencing libraries with the ability to pinpoint these responses (Montague et. al.) have often failed to report the specific features identified here, and certainly do not provide the context of those features as they relate to emerging autoreactivity.

Rather than a broad analysis of the B cell repertoire in COVID-19, our goal instead is to provide a focused look at features of the ASC emerging in the early phase of severe disease that are ultimately linked in later figures to the transient autoreactivity that emerges in these patients. We now have also added an entirely new scVDJ analysis of mild/moderate patients (Fig 1e-i), to deeply characterize the repertoire of a dominant emerging B cell compartment that is specific to severe disease, incompletely understood, and highly relevant to both the antiviral and autoreactive responses.

We also agree that B cells containing IGHV4-34 have broad targeting potential; indeed, that is exactly the point. The breadth of targeting of the IGHV4-34 gene is described as fundamentally autoreactive in nature, and in ways independent of CDR specificity. It is for this reason that these cells are generally censored from fully-developed immune responses (Pugh-Bernard et. al.), with their specific expansion found primarily in the context of autoimmunity. The observations here are not that a few clones with 4-34 receptors are expanding, it's that they are expanding in frequency *in relation* to other V-genes in the compartment – in some patients more than doubling in frequency. This, in the context of a seeming absence of enforced clonal redemption to remove autoreactive potential (Reed et. al.), and a reduction of selective pressure across the compartment strongly suggests a relaxed peripheral tolerance environment. It is the characterization of these features, in total, that justifies the downstream investigation into the actual observed autoreactivity of these clones, and their links to emerging autoreactivity in these patients.

Finally, the reviewer is correct in pointing out that by comparing to the memory compartment in these patients, much of the compartment in the comparison will consist of pre-existing memory. This is intended – the point of these data is not to fully describe the antigen-specific B cell memory compartment in COVID-19 as others have done. It is instead to highlight the dynamic nature of these responses and emphasize that, while these low selection responses rapidly come to dominate the overall antibody secreting potential of the compartment, they do not necessarily come to dominate the memory compartment in the same way. To emphasize this point, we introduce new analyses to highlight the connections, or lack thereof, between the ASC and memory compartments in the

emerging humoral response (Fig 2d,e). In those analyses, we find that within the 4 patient single cell cohort, only 3 show *any* clonal connection between the ASC and memory compartment (Fig 2d). Of those three, connectivity of low selection ASCs to the memory was essentially non-existent (Fig 2e). This, combined with a lack of connectivity of all but 2 clonotypes across the entire dataset with recovery time points strongly indicates that this compartment should be considered and evaluated independently of the emerging memory.

5. The AVY patch comparisons in Figure 3c are again biased, though they are consistent with the fact that the majority of accumulated memory B cells against a very wide range of antigens likely largely emanated from germinal centers and in these cells the AVY patch in the fairly promiscuous rearranged VH4-34 gene is likely to be mutated.

We generally agree that this comparison from the ASC to memory compartment is asymmetric but would reiterate that that is intended – the emerging ASC compartment of patients with critical disease and is not necessarily reflected in the contemporaneous memory. This distinction becomes important when discussing longevity of these responses later in the manuscript and provides an early justification for why emerging autoreactivity might be expected to wane.

In response to reviewer 3 comments on the reliance of our initial analysis on data from individual patients rather than emphasizing analysis of the group as a whole: While all patients in the dataset contained sufficient cell numbers to provide good data on the emphasis of the low-selection, IgG1 compartment, only two contained sufficient numbers of IgG1, IGHV4-34 clonotypes (>15) to provide sufficient confidence in the AVY analysis. While we believe these data, and they are consistent with the model we propose, we have chosen to remove them from the main figures and instead include them as supplemental information (Supplemental figure 3) to alleviate concerns over data ‘cherry-picking’ in defending the work’s main message.

6. SHM has been well established to be limited in activated antigen-specific B cells in COVID-19 Is it surprising then that the expansion of B cells to SARS-CoV-2 antigens in individuals infected for the first time or in non-vaccinated individuals are largely naïve B cell derived? The connectivity data nonetheless represents novel information, though it does not necessarily yield new insights.

We appreciate the reviewer’s perspective and generally agree with the expected role of naive B cells. In fact, our studies were meant and designed to interrogate the human primary B cell response and its properties in COVID infection. This is important for multiple reasons including, as mentioned by the reviewer, that the accumulation of somatic hypermutation in class-switched cells continues to be attributed to GC-memory derived in multiple papers including some dedicated to classifying human memory responses. It should also be noted that as shown in our work and others, while the average mutation rate is clearly decreased in IgG1 ASC (both in acute SLE and COVID-19), this response also contains a fraction of highly mutated cells that most likely represent polyclonal, non-specific activation of pre-existing memory cells. Thus, while it might be predicted, it remains equally important to formally establish the presence, expansion and behavior of the naive-derived component representing the primary response to COVID-19. That is indeed, one central element of the work and one that to our knowledge had not been established before.

The inclusion of the data was also important to address the belief amongst a variety of prominent groups that the observed autoreactive responses in these patients, due to the rapid kinetics of their observation, must be derived from pre-existing memory. In contrast, we show (alongside our previous work) that the EF response can readily generate these antibodies within two weeks or less. The current

work, shows that the majority of the low-mutation population are indeed antigen-specific (whether anti-viral, autoreactive or both), thereby providing formal demonstration of the characteristics of the primary immune response that had been previously lacking. Our observations, in conjunction with the data showing RBD-targeting alongside autoreactive targeting, provides clear evidence that this is not the case; both SARS-CoV-2 specific and self-specific clonotypes are emerging together in this low-selection compartment from a naive origin. However, as the reviewer has suggested that this observation may not be surprising to the reader, we have chosen to remove these data from the main figure and instead place them in supplemental figure 3.

7. What could enhance these studies? I have two suggestions. One example for bi-specificity comes from VH4-34 that is already overrepresented in the naïve B cell population and contributes to the binding of many different autoantigens. It is not too surprising that some BCRs containing rearranged VH4-34 find an epitope in one of the 29 proteins of SARS-CoV-2. While I do not believe that autoimmunity in COVID-19 will be significantly linked to molecular mimicry (see below) nevertheless it would be useful to fine map this epitope and establish whether or not a specific nucleocapsid-specific BCR that contains rearranged VH4-34 is truly autoreactive.

The other alternative would be to study a few specific autoreactive BCRs in detail -say against carbamylated proteins or GBM. Even though SHM is low in these BCRs, it is not zero. It is theoretically possible that the unmutated common ancestor (UCA) for each BCR remains autoreactive, and that possibility could be tested. It could be shown whether these UCAs do or do not interact with any known protein of SARS-CoV-2. These results would lend support to the Zinkernagel model and suggest that as for other pathogens, antigen overload can lead to auto-reactive naïve B cells capturing and presenting viral peptides non-specifically and thus induce a promiscuous T-dependent B cell response that triggers autoimmunity.

We appreciate the reviewers' careful evaluation of this work, their perspective on its potential impact, and guidance on how to improve it. As stated above, the authors agree that an opportunity was missed in the writing of the initial manuscript to discuss the impact that these data have on the concept of molecular mimicry, particularly as it relates to COVID-19.

We regret that a key feature of the antibodies tested was not appropriately emphasized – many (57/107) of the monoclonals produced and tested were indeed germline in configuration in both the heavy and light chain, having accumulated no mutations and speaking directly to the reviewers point. Within that group, we identify monoclonals that are highly RBD-specific with no observed autoreactivity (Fig 4a, mAb-13), monoclonals with reactivity to both virus and self antigen (Fig4a, mAbs 3,5,7), and monoclonals with reactivity only to self-antigen (Fig4a, mAb 6). Antibodies with zero mutations across both heavy and light chains have now been indicated in the heatmap to aid in interpretability. We also observe monoclonal antibodies reactive against the same self antigen, but with dual specificity to *different viral antigens* (Fig 4a, mAbs 5,6,7). Combined, these two observations strongly argue against molecular mimicry as the root of these self-targeted responses as the reviewer points out.

Further, in response to reviewer 2's request (below), we have now included an entirely new demographically-matched cohort of ICU patients admitted due to acute respiratory distress syndrome (ARDS) as a result of confirmed bacterial pneumonia. The results of autoreactivity testing in these patients was surprising, and essentially phenocopied patients recruited with severe COVID-19 with similar levels of overall autoreactivity, ANA and CarP positivity, and even anti-GBM responses (Main table 1, Fig 3f). This observation has sweeping implications, not the least of which is that molecular mimicry is, in light of those data, is difficult to defend. We have highlighted these points as needed in

the initial presentation of the data, as well as in a significantly altered discussion surrounding the impact of this work on previously established models.

Referee #2 (Remarks to the Author):

It is now increasingly recognized that aberrant immune responses to SARS-CoV-2 can be a major contributor to the pathogenesis that accompanies the development of severe disease in COVID-19 patients. The Sanz group has been at the forefront in demonstrating that one of the key features observed in many critically ill COVID-19 patients is the inappropriate activation of a population of T-bet driven B cells known to accumulate in SLE (which they term DN2s, are enriched in VH4-34, and share similarities to murine ABCs) as well as a marked expansion of ASCs. Another recently recognized aspect of SARS-CoV-2 related pathogenesis is the observation that disease can be accompanied by the production of a wide-range of autoantibodies (from anti-PL Abs to anti-IFN Abs to Abs targeting the exoproteome) a feature that again preferentially accompanies severe disease. Given that DN2s are believed to differentiate into ASCs primarily via an EF route and to be major autoAb producers in SLE, the authors in this manuscript employ several elegant and novel approaches to extend their original analysis of COVID-19 patients described in their Nature Immunology manuscript to establish closer connections between these B cells subsets and the production of both antiviral and potentially pathogenic autoantibodies (primarily of the IgG1 isotype). In particular, their analysis aims to uncover the “origins, breadth, and resolution” of the transient autoreactivity that is observed in severe COVID-19.

While the production of autoantibodies has long been recognized to transiently accompany many infections in addition to COVID-19 (e.g. malaria, Chikungunya virus, hepatitis, Lyme disease, etc) and hence this particular aspect of the story is not as innovative, the impact and novelty of the manuscript lies in the in-depth analysis of the ASC compartment and the combined molecular and serological approaches utilized to understand the mechanism of the appearance of this phenomenon during COVID-19. Furthermore, some of the findings, such as the detection of anti-carbamylated antibodies in severe COVID-19 patients, could potentially have important clinical implications. Despite a great degree of enthusiasm for the work, however, the studies need to be strengthened in several areas.

The authors appreciate the reviewer’s perspective on the work and their thorough review of the manuscript. We appreciate the reviewer’s description of the study design as ‘elegant’ and ‘novel’. We agree that much of the novelty of the current manuscript lies in its integration of cellular and molecular data into the broader concepts of clinical autoreactivity observed in severe COVID-19, and the authors are pleased to hear that the reviewer shares our enthusiasm for the work. However, we understand the reviewer’s concerns with the initial submission, and have addressed them comprehensively below:

1) While the authors take advantage of the patient cohort described in their Nature Immunology manuscript most of the studies deal with a very small (and at times extremely small) subset of these patients. Thus, some of their analysis could be profoundly skewed by the demographics and clinical characteristics of the subjects on which the in-depth analysis is performed. The authors should thus provide additional clinical information on the patients investigated in each of the figures. More detailed information about timing of the analysis in each figure would also be helpful to the reader. For instance, which ones are ICU-1 and ICU-2 in Fig 2a? How do their demographics compare to the 3 HD controls? Was there a change in the clinical course of the ICU-1 patient that might have contributed to the change in the ANA pattern from nuclear to homogenous in a 4-day span and the increase in other autoAb titers in Fig. 4d?

We appreciate the reviewer’s request for clarity in the disease states and demographics of these patients, and regret that it was not made clearer in the first submission. In the Nature Immunology manuscript, our ability to obtain demographically-matched healthy cohorts was significantly hampered

by the environment surrounding the early phases of the pandemic. The current study does not share those limitations, and in an effort to make these data more transparent and accessible, an additional patient demographics table specific to the patients included in the single cell analysis has been provided as supplemental table 3. In addition, Fig 1e (previously Fig 2a) now includes naming of all patients presented elsewhere in the work to serve as a reference should the reader choose to go back and look at individual patient data. Importantly, and to the reviewer's specific question, patient ICU-2 (34yo AA M) is well matched to patient HD-2 (36yo AA M) which we hope will alleviate concerns about interpretability of the findings. Patient ICU-1 is slightly older at 53yo.

As to the clinical course of patient ICU-1, after reviewing the patient's medical records, there is no single major event that could be reliably linked to the change in ANA staining pattern, due in large part to the large number of confounding variables associated with the severe disease course. The patient was admitted and intubated on d7 PSO. Unfortunately, fine assessment of kidney abnormalities (eg the presence/absence of hematuria or proteinuria) is not routinely measured in this setting, however, in the following week developed acute respiratory stress syndrome, coagulopathy secondary to COVID-19, altered mental status, and was treated with a variety of medications which broadly fall into the categories of supportive care, putative treatments of COVID-19 (such as remdesivir), and prophylactic medications (such as heparin). Whether the worsening course of the disease was linked to pathogenic autoantibodies is important and will require extensive study into the pathogenic nature of the self-targeted antibodies that we have identified which we hope to investigate thoroughly in future studies.

2) The finding of broad autoreactivity is consistent with other recent studies and obviously of great clinical interest. The ability of the investigators to link this finding to the EF-derived IgG1 ASCs is thus one of the most interesting aspects of the manuscript. However, while the investigators provide some evidence for these connections additional information should be included to further support this claim. Specifically,
a. The authors should assess whether the isotype of the plasma autoantibodies detected in their ICU COVID patients is indeed IgG1. This should be easily testable for at least some of the autoantibodies like the anti-CarP antibodies.

The authors appreciate the reviewer's perspective on the potential clinical importance of the work and understand their desire to pinpoint the clinical serology to this specific antibody subclass. To this end, we have selected five patients that were identified through our general screen as having clinically positive levels of HEp-2-targeted IgG antibodies as determined by ELISA. We tested these patients against the two most prominent IgG subclasses present in the repertoire – IgG1 and IgG2. Consistent with our model, IgG1 provided both positive signal across the group, and higher signal than the paired IgG2 response. These data have now been added to the manuscript in Supplemental figure 3 as further justification to investigate the specificities of individual clonotypes within the IgG1⁺, low selection ASC compartment of patients ICU-1 and ICU-2, and are addressed in the results section accordingly.

b. The presence of anti-CarP (and potentially anti-GBM) antibodies could be an important biomarker not only of acute peripheral tolerance breaks but also of the extent of lung damage in these patients. However, the authors cannot exclude that presence of these antibodies could be preexisting in some of these patients based on their age, smoking status etc. Hence availability of (any) pre-infection samples would be very valuable to assess this possibility. One would furthermore presume that the ICU-COVID (but not the OUT-C) patients were on mechanical ventilation, which could be a key contributing feature to the findings. Thus, samples from non-COVID ICU patients on mechanical ventilation should be included as a control. Ideally one would also assess samples from intubated patients with other viral/bacterial infections to evaluate whether these findings are specific for COVID-19 patients. Again providing additional information on the clinical course of these patients (e.g. did they have any evidence of acute renal damage?) would be very helpful.

The authors agree with the reviewer that a fully longitudinal dataset of these patients, inclusive of pre-infection samples would be a valuable addition to the manuscript. Unfortunately, none of the patients collected from the ICUs were patients that had been previously enrolled in our protocols. As a result, this avenue of investigation is unfortunately unavailable. However, while we cannot rule out that some of these patients may have had previously existing autoantibodies, should that have been the case, the cellular derivation of those autoantibodies would be from memory cells and not from naive B cells – the central finding in our study. Our data shows conclusively the development of newly derived autoreactive B cell clonotypes with germline B cell receptor configurations and no attachment to the B cell memory alongside SARS-CoV-2-specific antibodies. Indeed, the identification of clonotypes simultaneously specific for RBD and GBM (Fig4a, mAb 5) strongly argues the naive derivation of these responses irrespective of any additional pre-existing memory response. Further, their general resolution upon recovery (Fig 6a), is strongly suggestive that these responses are developing acutely and then resolving over time. This is bolstered by new data (Fig 3l) showing the kinetics of the onset of increased reactivity against HEp-2 over time in these patients as a group.

The reviewer's point is well-taken that it is likely impossible to account for all the confounders in a single critically ill patient which is why we have taken great care to focus in this manuscript instead at discrete, objective immunologic endpoints rather than trying to associate clinical outcomes (such as the presence or absence of renal failure) with immunologic phenomena. This is in spite of the fact that many of the autoantibodies contained herein are associated with (and, in some cases, biomarkers for) clinical diseases. We trust that the presentation of these data will allow for large, adequately powered, multi-center, randomized trials to account for confounding variables and to see if breaks in immune tolerance are associated with clinical outcomes or are instead a byproduct of severe COVID-19

However, to address the question of generalizability directly, we have now included in our analysis an entirely new demographically-matched cohort of ICU patients admitted due to acute respiratory distress syndrome (ARDS) as a result of confirmed bacterial pneumonia. These patients were also mechanically ventilated and in the acute phase of infection. The similarities between autoreactive profiles between the two groups is stark (Table 1), and beyond the implications that these data have on molecular mimicry as outlined above, has significant implications on the entire fields of infectious disease and critical care. The presence of anti-CarP protein responses, ANAs, and anti-GBM antibodies in highly similar frequencies in almost perfectly matched cohorts is strongly suggestive that the findings are not specific to COVID-19 and are likely to be identified in a variety of highly inflammatory respiratory infections. Thus, our detailed cellular studies provide a needed mechanistic explanation for an important phenomenon and open the door to future investigations of B cell tolerance breakdown in acute severe infections and open the door to B cell modulating interventions during potential therapeutic windows. Whether these findings are linked to mechanical ventilation is an intriguing possibility that will require further investigation in patients without generalized inflammation.

c. While the authors do attempt to broaden their findings to an additional cohort of patients, this analysis, as they admit, primarily included clinically validated autoantibodies like ANA and RF (which is often an IgM and seems to be the main difference between high and low CRP patients) and thus falls short of supporting their key claim that this autoreactivity is linked to IgG1 producing EF-derived ASCs. Given the very small number of ICU-COVID patients in which their in-depth analysis was conducted, gaining additional support for their main findings (e.g. by assessing the production/kinetics of IgG1 autoantibodies) in additional cohorts of COVID-19 patients (which unfortunately are not lacking) of different disease severity is very important.

The authors understand the reviewer's desire for comparative groups of different disease severities. We would highlight similar requests by reviewer 3 below and understand that the initial manuscript did not appropriately demonstrate this phenomenon as specific for patients with severe/critical disease. To this end, the revised manuscript includes a completely new cohort of 4 patients with mild/moderate COVID-19 collected and assessed via single cell repertoire analysis (Fig 1e-i). Detailed analysis of the ASC repertoires of these patients reveals that while they display some of the same trends in IgG1 class switching in the ASC compartment, patients with less severe disease do not experience the same influx of low-selection clones that dominate the repertoire of patients with more serious disease.

In addition, to further validate the identification of an emerging autoreactive response, we have included an additional 90 patients into our study – 50 patients in the acute phase of disease and 40 recruited from Post-acute sequelae of COVID-19 (PASC) clinics between 2 and 14 months post symptom onset. With these new data, kinetic analyses of HEp-2 reactivity were performed across the cohort, revealing an onset of increased levels of autoreactivity in hospitalized patients between 12 and 14 days post symptom onset that persist for several months before declining across the cohort (Figs 3l and 6e). Again, these data strongly suggest a new onset of IgG-specific clinical autoreactivity in a subset of patients, and that while they appear to wane over time, can still persist for several months beyond recovery from the acute phase of infection. These data have clear implications in the study of PASC, in general, and a detailed analysis of those patients, alongside mild/moderate patients that also experience ongoing symptoms is also underway.

3) More experimental details need to be provided throughout the manuscript. In particular the authors utilize BASELINE to assess the CDR selective pressure and state that they observe a selective reduction in IgG1 but not other class switched compartments. However, in Fig 2h IgG2 in the ICU cohort also seems to be diminished. Can they provide the specific parameters utilized to determine the significance of their findings? How many cells was this analysis based on for the various isotypes in HD versus ICU-COVID patients? Were the differences statistically significant? Given that this analysis is one of their key findings and is being utilized in several figures, this information would help the reader better assess the different comparisons that they perform. Was any in-depth analysis conducted in the non-ICU COVID cases?

The authors regret that this analysis lacked transparency in the initial submission and agree with the reviewer that a more statistically-robust representation of those data were important in the interpretability of the response. A more detailed methods section has been provided, and the manuscript text has been updated for transparency. In addition the figures including these analyses (Figs 1k, 2a/b, 5e/f) have been replaced or supplemented with new visualizations to display the 95% confidence intervals associated with the analysis to aid in interpretation and statistical validity. Further, the addition of the non-ICU cohort in the repertoire analysis should help alleviate concerns that this phenomenon may not be specific to patients with severe disease.

4) The comparison of the differences in break in tolerance between those in COVID-19 and the ones commonly observed in autoimmune patients is very intriguing. As the authors and others have shown, the key signals driving the expansion of these cells in autoimmune disease are TLR7 engagement as well as the presence of T cell cytokines like IL-21 and IFN γ . Could the lack of TFH cells, and/or the presence of antibodies targeting these cytokines in COVID-19 patients limit the T cell help that these cells can receive and make them rely primarily on TLR7 driven signals? Could this contribute to the transient and more limited nature of these autoreactive responses and be an important difference with autoimmune disease? In this regard are the anti-CarP antibodies in ICU-3 that increase during the recovery period still predominantly IgG1 or do they include different IgG isotypes?

The authors agree that the specific signatures of these patients, now paired with the new cohort of ARDS patients with similar autoimmune signatures, is extremely interesting and requires further investigation as to how extrafollicular microenvironments and autoantigen availability during acute severe infection may influence the specificity of autoantibody targeting. As the reviewer correctly points out, a lack of TFH cells (Kaneko et. al.) and disruptive anti-cytokine targeting (Wang et. al.) has now been well documented in these patients and could certainly contribute to the overall germinal center loss (Kaneko et. al.) that has been identified in terminal infections. Whether the collapse of germinal centers initiates the reliance of responses on extrafollicular pathways, or positive feedback within the EF pathway in a hyperinflammatory state forces the collapse of GCs is an important requiring further investigation and directed study – likely in model systems where these components can be carefully controlled. In either case, the authors certainly believe that this emphasis on EF pathway initiation, which has been previously shown to specialize in the generation of short-term plasmablast-based responses and typically precedes mature GC reactions, is likely to contribute to the transient nature of these responses. However, to the reviewer’s last point, there do seem to be patients (such as ICU-4) where these responses may not be as transitory. Newly included data from 40 additional patients recruited from COVID-19 recovery clinics around the Emory University network suggests a general waning of autoreactivity over the year following recovery, however there are also clear exceptions which must be followed up on, such as patient ICU-4 (Fig 6d-f). It should be noted however, that EF reactions can also generate long-term responses and even contribute to long-lived plasma cells. These fundamental questions are now facilitated by our results and should be addressed in larger and longer future studies.

5) Is there any benefit from this relaxed break of tolerance? Could these cross-reactive antibodies serve as a rapid response to control infection with autoimmunity being a “side effect” of rapidly responding? Could this break in tolerance be beneficial for host survival? The authors find 70% of the EF IgG responses to be anti-viral, do these also wane?

Our best interpretation of the available data is that the reviewer’s model is correct. It is our belief that, due to an upstream mechanism outside of the scope of the current manuscript (immunosuppression, genetic immunodeficiency, etc.), early infection is poorly controlled and results in a hyperinflammatory environment favoring the EF response pathway. This pathway, while predisposed to autoreactivity, is nonetheless capable of producing high levels of neutralizing antibodies as identified in our previous work (Woodruff et. al.). Viewed in this way, the reviewer would be correct – emerging autoreactivity would be a tradeoff of a rapid response pathway intended first and foremost to reduce viral burden. In line with that model, the answer to the reviewers last question is yes – while we emphasize the loss of the autoreactive clonotypes identified in acute infection, we also report (with increased clarity in the revised text) the loss of viral-targeted clonotypes as well.

Minor comments:

1) While one recognizes the challenges of investigating the ASC compartment in COVID-19 patients, these cells are notoriously fragile and might be affected by freezing the samples (which are being utilized for all of the studies). Have the authors evaluated whether there is any differential loss of Ig producing capabilities upon freezing/thawing of samples in their MENSA assay by comparing fresh versus frozen samples (which could be conducted in samples from HD subjects)?

We understand the reviewers' concerns about investigating ASCs that have been previously frozen, and regret that the methodology surrounding the MENSA analysis was not clear in the initial submission of the manuscript. The MENSA samples are performed on fresh cells directly isolated from

patient samples and are never frozen. The methods section outlining this protocol has been updated for clarity surrounding this point.

2) In the legend of Fig. 1 the g and h panels are reversed.

The authors thank the reviewer for bringing our attention to this oversight. It has been corrected in the revised version of the manuscript.

Again, we thank the reviewer for their insightful comments, and hope that the substantially revised manuscript and new patient cohorts help alleviate their initial reservations.

Referee #3 (Remarks to the Author):

A. Summary of the key results

In this manuscript, the authors studied the evolution of BCR specificity during severe COVID-19. This report builds from previous findings published by this group and others^{1,2}, which showed that severe COVID-19 is characterized by an exaggerated extrafollicular B cell and ASC response and expanded DN2 B cells previously shown to contribute to autoimmunity in SLE3. In this work, the authors again show that severe COVID-19 patients display an expansion of IgG1 ASC and EF B cells, and that these IgG1 ASC abundantly secrete RBD specific IgG. Again, similar to their previous work¹, the authors show that expanded IgG1 ASC display decreased SHM and elevated usage of autoreactive IGHV4-34 with preservation of the FR1 patch. Building off of this, the authors show that these low SHM IgG1 ASC are derived from the naïve B cell compartment as opposed to the memory compartment. From data derived from a single patient, authors show these IgG1 ASCs are polyreactive, specific for both SARS-CoV2 antigens and intracellular and extracellular self-antigens. In two separate larger cohorts, the authors show a correlation between severe disease and inflammatory markers with presence of select autoantibodies against, most notably, carbamylated proteins and nuclear antigens. Finally, one of the most interesting findings of this paper is the contraction and disappearance of expanded low SHM IgG1 ASC, exclusion of these clones from the memory B cell compartment (in one patient), and decreases in certain autoantibody titers at 6 months after acute infection.

The authors thank the reviewer for their careful evaluation of this work. We are encouraged that they identified novel areas in the work presented and expressed interest in respect to the contraction of this unique compartment. However, we also understand the reviewer's concerns about the presentation of concepts available from previously available literature (by our group and others) and have addressed those specific concerns below. The authors have invested significant effort and resources to provide additional appropriate controls based on the reviewer's suggestions, and generally broaden our analysis and data presentation. We believe that the reviewer will find the revised manuscript to be significantly more defensible, with a broadened scope and ultimately greater impact.

B. Originality and significance

Several findings presented in the manuscript, particularly in figures 1 and 2, are not novel or particularly differentiated from the authors' previous report¹. For example, both papers show that severe COVID-19 is characterized by a dominance of EF B cells and expansion of IgG1 ASC with decreased SHM. Both papers show that severe COVID-19 patients display elevated usage of IGHV4-34 with increased prevalence of the FR1 patch that is normally eliminated through SHM. Thus, major segments of the current paper are largely confirmatory of the previous work. While it is nice to show that the findings are generalizable and supported by

multiple studies, it seems much of this data could be moved to supplemental figures or supported by references.

The authors appreciate the reviewer's comments regarding Figures 1 and 2 of the original manuscript and understand their point. This point was also consistent with Reviewer 1's perspective (above), and we agree that while our intent was to provide context to the ASC compartment that the rest of the manuscript describes, some of the findings are confirmatory in nature. As laid out in response to Reviewer 1, these two figures have been consolidated into a single figure outlining phenotype of the ASC compartment in a statistically rigorous manner.

The Reviewer is correct in that our previous work highlighted the low mutation frequency and IgG1 predisposition of ASCs from a single patient identified within our compiled ICU cohort. In addition, we identified increased titers of IGHV4-34-based, 9G4 idiotype-reactive antibodies in the blood of severe patients (although no repertoire-based analysis of this increase was performed). Further, we highlighted 2 ASC clonotypes in the patient expressing IGHV4-34 genes and suggest that the overall data from that patient were consistent with the described EF response that formed the backbone of that work. While this was an important finding that needed to be communicated during the early phase of the pandemic, it was nonetheless, limited to a single patient, did not include mild/moderate controls, and while consistent with the model we continue to believe to be true, were not amenable to statistical analysis (and were reported as such).

Here, we solidify and contextualize those observations. As per the Reviewer's comments (below) we have added an entirely new mild patient cohort to aid in interpretability and have applied statistical rigor to the observations that we were previously only able to speculate. Following consolidation of figures 1 and 2, the remaining panels now contain statistically valid characterizations of the IgG1+ low-selection compartment with healthy donor and mild COVID-19 comparisons unique to the current work.

Aspects of other points made in this paper, are somewhat novel, though not particularly unexpected based on preexisting data. For example, the elevated frequencies of autoantibodies against RF4,5, nuclear antigen6, cardiolipin7, and extracellular proteins8 in COVID-19 patients have already been reported and, in some studies, shown to correlate with severity of infection6,8. Findings showing antibodies against GBM, of which only isolated cases have been reported, and carbamylated proteins are novel and interesting.

The authors agree that features of clinical autoreactivity in COVID-19 have indeed been previously reported starting from the earliest days of the pandemic. Indeed, the preprint that is associated with the current manuscript (presenting data similar to Figure 3g) was uploaded in October, 2020 on the heels of early reports of anti-phospholipid and interferon-alpha directed antibodies which are referenced in that manuscript. However, as previously discussed, definitive evidence for the cellular origin of the COVID-associated autoantibodies has been lacking and in fact, several prominent publications have postulated a memory source of pre-formed autoantibodies. These conclusions have been based on purely theoretical considerations, namely the presumed lack of sufficient time for the generation of switched autoantibodies within 10-15 days after symptom onset. While this is a reasonable consideration as human memory cells do contain a significant frequency of autoreactive B cells (Scheid et. al.) that could be expanded in the context of generalized inflammation, our data conclusively show that: 1) the timing is fully consistent with a naive origin for de novo autoimmunity; and 2) naive-derived IgG1 ASC with low mutation and even full germline configuration, are greatly enriched in COVID-19 triggered autoreactivity.

As the reviewer points out below, we believe the strength of the current work to be in its ability to provide context to these findings within the broader lens of B cell response development. It is through broad screening of these patients that anti-carbamylated and anti-GBM responses were identified, and while some results of those screens were consistent with previously published literature, we believe that it is still important to provide broader context to those data – particularly in light of the newly provided data from patients with ARDS due to bacterial pneumonia (see Reviewer 2 comment 2b).

Other aspects of the paper are novel, interesting, and important. For example, data in one patient showing IgG1 low SHM ASCs are polyspecific to self- and SARS-CoV-2 antigens provides an interesting and plausible bridge between the aberrant B cell abnormalities and the presence of extensive intracellular and extracellular autoantibodies in patients with severe disease. One major concern, however, is the generalizability of these findings, given that they are only shown in one individual. Second, and equally notable, the results showing contraction of the low SHM IgG1 ASC response and exclusion of these cells from the memory compartment is highly significant, novel, and perhaps the strongest aspect of this work. However, again some of these conclusions are based 1 or 2 patients, making the generalizability of these findings uncertain.

The authors appreciate the Reviewer’s identification of pieces of the work as ‘novel, interesting, and important’. As above, we agree that a strength of this work is to extend and contextualize the observations of autoreactivity now common in COVID-19 and help cement those observations into more fundamental B cell developmental processes. We have gone to significant lengths to alleviate legitimate by the reviewer about the generalizability of the work including:

- 1. Adding a new cohort of mild patients with COVID-19 for single B cell repertoire analysis**
- 2. Creating a new set of monoclonal antibodies from an independent donor for specificity testing**
- 3. Adding statistical analysis to previously anecdotal representations of the data**
- 4. Adding reporting of the full autoreactive courses of the 4 patients at the core of this analysis**
- 5. Ensuring that all analysis in the main figures is statistically valid and unreliant on anecdote.**

We believe that, in large part due to the reviewer’s comments, the revised manuscript is significantly improved over the previous version of the work and addresses all major concerns as identified below.

C. Data & methodology: validity of approach, quality of data, quality of presentation

Most of the work here is performed using sound techniques, some of which are novel. A general criticism of the data presentation is that it is hard to follow exactly which and how many patient samples are being analyzed. Below are comments regarding the authors' approach:

1. One of the most novel claims of this work is that IgG1 antibody from low mutation frequency ASC are polyspecific for SARS-CoV2 and common autoantigens such as cardiolipin, nuclear antigen, B cells, and glomerular basement membrane, in addition to extracellular proteins enriched for immune and lung specificity (figure 4). Although this is an interesting and scientifically plausible finding, the results here were produced from a single patient without a control of either healthy donor(s) or patient(s) with mild COVID-19. Thus, it is not possible to conclude that the IgG1 ASC polyspecificity observed in this individual is representative of severe COVID-19 or whether this is even a unique phenomenon to severe vs mild COVID-19. The authors partially support the generalizability of the single cell polyreactivity findings by showing that elevated reactivity to RF and ANA is correlated with severe COVID-19 in two separate large cohorts. However, the elevated presence of these autoantibodies alone does not support the claims of their origin put forth in figure 4. At the bare minimum, this conclusion needs to be supported by additional severe COVID-19 patient(s) and mild COVID-19 control(s). Furthermore, the authors state in the paper text that they performed single cell BCR

repertoire analysis on 4 severe Covid19 patients. Why only one of them was chosen for figure 4 is unclear and needs to be further explained.

The authors appreciate the reviewer's perspective on monoclonal antibody testing data and understand that the original manuscript left open questions about the finding's generalizability. We also agree that screening of these clonotypes for viral-, and auto-reactivity is a critical feature of the work and have made several significant modifications including the incorporation of important new data, to address this important point.

First, the Reviewer is correct (here and elsewhere) that lack of a mild COVID-19 patient cohort in the original manuscript significantly hampered its interpretability. To this end, we have now included data from 4 additional patients with mild COVID-19 for single B cell repertoire analysis of the ASC compartment. Importantly, those patients are now included in the general repertoire characterization of Figure 1, and, consistent with the autoreactivity data trends displayed in figure 3, do not display the same IgG1-focused, low-mutation ASC compartment arising in ICU patients at similar time points post infection. This phenomenon, based on the controls the reviewer requested, does appear to be specific to, or at least much more frequently observed in, the ICU-C cohort.

Patient ICU-1 was selected for the initial production of antibodies because they were highly autoreactive, showed strong activation of the EF response, displayed significant alterations in both isotype selection and mutation frequency (Figure 1 e,g), and contained high enough cell numbers of ASCs in the repertoire data to perform a relatively broad analysis of clonotype specificity. We reasoned that 50 monoclonal antibodies would provide sufficient breadth to understand the general targeting tendencies of a compartment of ASCs uniquely emphasized in the overall ICU-C cohort, while still providing an opportunity to identify rare autoreactive events. A control set of antibodies was not manufactured simply because the compartment did not exist in our healthy control cohort. These cells are similarly absent in the new mild cohort.

However, as a key point of novelty in this manuscript, and in response to the Reviewer's desire to confirm generalizability of the initial finding further, we have manufactured a second set of 53 monoclonal antibodies (ICU-2, Fig 4a) for antigen testing. In confirmation of the previous findings, the second set of antibodies (again derived from clonotypes unique to the ICU-C cohort) show almost identical levels of antiviral targeting, autoreactivity, and dual-reactivity. We believe that confirmation of this phenotype in a second patient adds significant weight to the findings we have presented here, and we thank the reviewer for their suggestion.

In reading the reviewer's comment, the authors would also offer a point of clarification important in the discussion and interpretation of these results. While we stand behind and fully support a model of polyreactivity being a component of the autoreactivity observed in this compartment, they do not believe that this is strictly necessary for autoreactivity to arise. Indeed, as pointed out above (Reviewer 1, comment 7), we believe that this relaxation of peripheral tolerance is a feature resulting from an outsized EF B cell response dependency and entirely independent of the viral antigen being targeted. This view is supported by the inclusion of the new bacterial-pneumonia-induced ARDS cohort which shares no common pathogenic antigen, and yet results in similar clinical autoreactivities as observed in severe COVID-19. We have added extensive discussion surrounding molecular mimicry and its relevance (see Reviewer 1 comments), significantly improving the scope and impact of the work.

2. The CDI recombinant protein binding assays lack an appropriate negative control (antibodies derived from healthy donor(s) or mild COVID-19). While the findings here are thematically congruent with other reports,

there is no way to assess whether the protein reactivities found in this single severe COVID-19 patient are unique or enriched in severe COVID-19. Similarly, there is no way to assess whether the enriched pathways identified using enrichr pathway analysis are enriched relative to any reference. For example, the pathways identified as significant in this single sample may be biased by the specific recombinant proteins present in the analysis platform. Without a control as reference, this data lacks meaning or context.

Comments 2 and 3 are addressed comprehensively, below.

3. Related to the point above, the CDI protein arrays are comprised of proteins expressed in the yeast cytoplasm as GST-fusions, yet many proteins highlighted (IFNa, CD49, L-selectin, SLAMF7) are secreted glycoproteins, raising questions about their fidelity in approximating the target antigen. As such, this array should only be used as a screening tool; interesting candidate autoantigens need to be confirmed with gold-standard ELISA or similar assays using validated recombinant proteins expressed from mammalian cells.

The authors agree – the initial presentation of the CDI array data, while consistent with the previously established literature, was poorly represented and defended. Importantly, the three GSEA plots used for validation of the Enrichr analysis showed no significant enrichment in the HD plasma control – data that should have been included in the initial submission. However, the authors are inclined to agree that validation and further testing (at significant additional cost) would be necessary to properly validate these results and report them alongside the validated clinical autoreactivity reporting. Even then, the resulting data would not be particularly novel as this data is largely confirmatory of the existing literature. As a result, we have chosen to remove these data from the revised manuscript and focus on the clinical autoreactivity that remains the important focus of the work.

4. The authors state that inclusion criteria for their second autoantibody cohort (52 critically ill patients from Atlanta ICUs) specified patients that “had received autoantibody testing as part of routine clinical care at the discretion of their treating physicians.” These inclusion criteria may upwardly bias the prevalence of autoantibodies in this group, as these tests are not typically performed in the absence of a history of autoimmunity or a clinical presentation suggestive of an autoimmune process. The authors should provide more clinical details of this cohort to enable a determination of whether the disease course and medical history of this cohort is representative or skewed towards an autoimmune population.

We understand the reviewer’s cause for concern in a potential selection bias in this cohort and appreciate the opportunity to clarify the testing surrounding these patients. During the pandemic, especially during the summer of 2020, there was significant heterogeneity in practice patterns for patients admitted to the intensive care unit with COVID-19 in the United States. After our initial work in Nature Immunology outlining SLE-like EF responses in patients with critical illness, several physicians at our institution began obtaining autoimmune serologies on patients with COVID-19 who were admitted to the ICU in the acute phase of their illness. These labs were intended by the treating physicians to guide referral to Rheumatology or Long COVID clinics if they were fortunate enough to recover from their severe illness and were ordered at the discretion of the treating physician. We therefore designed a retrospective observational study to capture these results to add evidence to our hypothesis that breaks in immune tolerance are common in patients with severe COVID-19.

The reviewer is correct that there is bias in our study design within this cohort, and certainly more rigorous, well-designed, multi-center trials are needed to evaluate autoreactivity in critical illness and its possible pathologic consequences. However, given the clinical situation in Atlanta, Georgia in the summer of 2020, there is unlikely to have been significant selection bias simply because of the homogenous nature of patient presentations and treatments available at that time. The physicians who

ordered the autoimmune serologies at our institutions seemed to have placed the orders on all new admissions to their units (however, because this is a retrospective study, we cannot confirm the precise reasons that these labs were ordered). Even if the treating physicians were using clinical discretion to order these labs for some patients but not others, the vast majority of patients who required ICU admission during the summer peak were largely the most severe because ICU beds were being prioritized for patients who were likely to develop respiratory failure requiring intubation. Assuming that breaks in peripheral tolerance are only present in the sickest patients, the minimal differences in severity between patients admitted to the ICU at that time would have made selection bias difficult (ie either all patients would have been selected for testing, or none of them would have been). Importantly, all patients were screened through chart review for any evidence of pre-existing autoimmune disease; patients with documentation of autoreactivity were removed from the analysis.

D. Appropriate use of statistics and treatment of uncertainties

The majority of the work here is performed using appropriate statistical analysis. Below are minor points:

1. Please quantify findings in figures 3D and 3E. It is unclear what is going on here, particularly in ICU-1 in figure 3D; while the authors claim the IgG1 ASCs are predominantly coming from the CD27- naïve compartment, it appears by eye that the memory and CD27- compartments equally contribute. A numerical metric here would help to prove their claims.

The authors agree that the alluvial plots are difficult to interpret visually, and quantitation has been added to the panel to quantify the number of connections observed. However, in the spirit of the Reviewer's overall comments (below), the authors believe that these data, although interesting, contribute to the overall impression that data may have been 'cherry picked' for presentation due to a low number of samples. As a result, we have reanalyzed the data with all four patients in the ICU single cell cohort and retained only analyses in the main figure that support the manuscript in a statistically valid and rigorous manner. However, the authors still believe that these data are important in further detailing the ASC phenotypes of ICU-1 and ICU-2 from which the monoclonal antibodies are derived. As a result, the updated data are now included in Supplemental figure 3.

2. The following figures lack statistical tests: 3A, 3C, 6D

As above, the authors understand the Reviewer's desire to have statistical analyses associated with each observation in the main figure set. The authors have altered to analysis and presentation in these, and other data panels to ensure that all data is either directly statistically analyzed, or immediately followed by a statistical analysis of the full cohort if representative data is provided.

E. Conclusions: robustness, validity, reliability

The following conclusions are well supported by the data:

1. Severe COVID-19 is characterized by dominant EF and IgG1 ASC populations with low mutation frequency and enriched usage of IGHV4-34.

2. Severe COVID-19 is characterized by elevated autoantibodies against RF, ANA, and Carbamylated proteins, and these are correlated with inflammatory markers such as CRP

While these conclusions are well supported, they lack significant novelty as discussed above.

We appreciate the reviewer's acknowledgement of the robustness of the data surrounding the repertoire analysis and clinical autoreactivity screens. Please see above for complete discussion surrounding the repertoire analysis (Reviewer 3, section B).

In terms of clinical autoreactivity, the documentation of anti-carbamylated protein responses in these patients is, to our knowledge, entirely novel as is the identification of anti-GBM responses. In the revised version of the manuscript, through the inclusion of an additional 50 ICU patient, we validate that finding in a separately collected and analyzed cohort. Further, we identify similar signatures in bacteria-induced ARDS patients suggesting a broader scope of these findings even beyond COVID-19. Finally, identification of persistent signature in our newly developed PASC cohort argues for continued potential relevance of these features of disease beyond the acute phase of infection. While we understand that more work must be done, the new cohorts included in this manuscript as a result of the Reviewer's comments have substantially boosted the novelty and potential impact of these findings.

The following conclusions are partially, but insufficiently supported by the data:

1. Antibodies from IgG1 low SHM ASC are polyreactive to SARS-CoV2 antigens and intracellular and extracellular autoantigens. While this data is interesting and plausible given previous reports showing elevated autoantibodies in severe COVID-19, the data is generated from a single patient and lacks an appropriate reference control, either mild COVID-19 or healthy individual(s). As such, definitive conclusions are not possible with the present data. For example, the authors speculate on the meaning of enriched pathways identified in enrichr pathway analysis as evidence that B cells develop autoreactive specificity for immediately available antigens in milieu of the blood/immune or lung compartments. While this is an interesting interpretation of the data, there is no solid evidence that these pathways are differentially enriched in severe COVID-19 relative to control.

The authors thank the reviewer for their comment. It is addressed above in Reviewer 3, Section C – points 2,3.

2. Relaxed peripheral tolerance observed in acute disease resolves upon recovery. This finding is another highly interesting and novel aspect of the paper. It is confusing, why the authors flip between showing 4 patients versus selecting 2 of the 4 for display, when they state in the text that they followed 4 patients with single cell analysis. Data for all 4 patients should be at least available in the supplementary figures. Furthermore, while authors claim decreased IgG1 ASC and return of normal SHM levels represents reversal of relaxed peripheral tolerance observed in disease, they do not show full data on all autoantibodies tested during acute disease. This data is needed to complete the narrative of his paper regarding long term outcomes of relaxed B cell peripheral tolerance. Specifically, autoantibody persistence is an important indicator of whether this relaxed tolerance has temporary versus long term implications. For example, if there is chronic persistence of autoantibodies months after resolution of acute disease, this would suggest conversion of a subset of IgG1 ASC to long-lived plasma cells which could be present in niches (ie bone marrow) not sampled in the peripheral blood in this study.

We regret that the initial manuscript lacked clarity around these important points and have revised the manuscript accordingly.

The impetus for highlighting individuals in the original manuscript was to provide examples representative of the overall cohort. However, in doing so, the Reviewer is correct that we may have

inadvertently failed to provide proper statistical support for the overall conclusions – particularly in reference to the recovery figure. To address this, the authors have taken several steps:

1. We have split the recovery figure (original figure 6) into two separate and independent main figures
 - a. Figure 5 now addresses the resolution of the low-selection IgG1 ASC repertoire. We have removed any analysis that relies on a single patient, and all concepts are now addressed across the cohort with appropriate statistical assessment. Connectivity, or lack, thereof, between acute and recovery phases is now addressed explicitly in the text for all 4 patients.
 - b. Figure 6 now comprehensively displays autoreactivity across all four ICU patients at all available time points. Importantly, all activities that resulted in a positive test are displayed for each patient with results discussed in detail.
2. As discussed above, we have added in additional patient cohorts to further explore the kinetics and resolution of these emerging autoreactive responses. In particular, the inclusion of 40 new PASC patients recovered from hospitalizing infections shows a tapering effect consistent with our initial interpretation of the data. However, as in patient ICU-4, it also suggests that there may be patients, as the Reviewer suggests, with autoreactive clonotypes now converted into longer term memory. The demands significant additional followup outside of the constraints of the current manuscript, and the authors look forward to addressing it further.

3. Polyreactive IgG1 ASC are excluded from the memory compartment. This is an important and valuable conclusion with great implications as to whether COVID-19 may result in long term breaches in humoral tolerance. However, the only evidence for this is connectivity analysis showing the IgG1 clonotypes were underrepresented in the memory compartment at 6 months. While this is interesting, again this analysis is derived from only one patient and may not be representative of the average outcome after COVID-19. Given the tremendous heterogeneity in patient outcomes (e.g., some patients entirely recover whereas others develop diverse manifestations of post-acute sequelae of COVID-19, PASC), a more comprehensive evaluation across a varied spectrum of patients is warranted.

We agree that a comprehensive evaluation of autoreactivity across a cross-section of disease severity is certainly important. We should point out that while indeed formal connectivity (or lack thereof), between IgG1 ASC and IgG1 memory by clonotype analysis was limited, the results need to be taken in the larger context of our comparative study of the frequency of low-mutation/low-selection IgG1 cells in the acute ASC versus the recovery ASC and memory compartment. On that basis, we believe that the central finding of the emergence of a new autoreactive compartment and its subsequent contraction upon resolution of the infection is robust through the analysis of multiple patients

With the inclusion of the new PASC cohort, we have shown a general tapering of autoreactivity in the year following acute infection (Fig 6e,f). However, as the reviewer points out, there is tremendous heterogeneity in patient outcomes in COVID-19, and a screen of autoreactivity would not be sufficient to defend the emergence of long-term autoreactivity or its role in symptom persistence or even developing autoimmunity. We are pleased that the observations resulting from the focus from this manuscript – the emergence of a low-selection, short-lived, autoreactive IgG1+ ASC compartment in patients with severe COVID-19 – are consistent with the new recovery-focused data and look forward to building on those data in a more comprehensive way outside the scope of the current manuscript.

The following conclusions are not well supported by the data:

1. Anti-carbamylated protein antibodies as a biomarker for relaxed peripheral tolerance in acute Covid19. The results showing 40% of severe Covid19 patients display reactivity to carbamylated proteins are interesting. However, there are several factors that undermine the claim of carbamylated protein reactivity is a biomarker for loss of peripheral tolerance checkpoints in severe COVID-19: 1) the lack of these findings in a secondary validation cohort; 2) lack of receiver operator analysis for prediction of loss of peripheral tolerance checkpoints in another cohort; 3) lack of any shown connection between presence of anti-carbamylated protein autoantibodies in figure 5 to loss of peripheral tolerance as shown in figures 2, 3, and 4. In other words: did the patients who had multiple autoreactivities and carbamylated protein reactivity in figure 5 also display expansions in low mutation IgG1 ASCs that the authors claim underly this relaxed peripheral tolerance? Also did they have antibodies of a similar profile to those found in the proteomics array of figure 4?

The authors agree that the confident use of the word ‘biomarker’ was premature, due in part to a lack of a validation cohort and a lack of analysis surrounding predictive value. Pending validation, ‘candidate biomarker’ would have been a more appropriate term. In our revised manuscript, we have now included an additional 50 patients for the purposes of inclusion in the kinetic analysis (Fig 3I) and have run additional analyses testing for anti-CarP reactivity (reviewer figure 2, below). The data are highly consistent with our initial cohort, suggesting that these results are consistent between patient groups – at least within the Atlanta area. This, with the addition of both the ARDS and PASC cohorts provides strong evidence that these are, indeed, routine signatures of autoreactivity in these patients.

However, while these signatures may be consistent within our datasets, the true use of these clinical

tests as biomarkers should be validated with much larger, multi-site studies which we believe are now justified by the current work.

As these studies have not yet been undertaken, we have softened our language throughout the manuscript to emphasize the positive findings of the work but ensure that it is clear that this is still a ‘candidate’ biomarker, albeit one with strong signal across our analyses.

Reviewer Figure 2 – Anti-CarP responses in the original manuscript cohort (left), and revised manuscript cohort (right). Titers displayed as U/ml

E. Suggested improvements: experiments, data for possible revision

Overall this paper puts forth interesting and highly relevant results regarding peripheral tolerance in severe COVID-19. These results bridge previous reports of aberrant B cell responses and autoantibody prevalence in COVID-19. However, the credibility of the claims made here suffer from the very low number of analyzed samples. For example, Figure 4, one of the most important and novel figures of the paper, has only one patient with no reference, making any results from this figure effectively uninterpretable. While results from figure 4 make sense given the current literature, the authors should include data from the other 3 patients whom they performed single cell BCR analysis on. If the single cell data was not sufficient (lack of cells, reads, etc), the authors should perform additional studies on new acute samples and controls.

Similarly, another key figure of the paper, figure 6, alternates inexplicably between including 1 patient, to 4 patients, to 2 patients. The authors should consistently show data for all 4 patients and not cherry-pick between them. (Parenthetically, even an n=4 patients seems like a fairly small number to derive sweeping conclusions.) The connectivity analysis showing that the IgG1 ASC compartment was underrepresented in the memory compartment at 6 months was another extremely relevant and exciting finding. However, again this analysis should be expanded to include more than 1 patient.

Additional experiments to connect the autoantibody data in figure 5 with the rest of the paper would also be additive. For example, it would be helpful to assess the persistence of all autoantibodies assayed at 6 months, and how this correlates with contraction and/or exclusion of the IgG1 ASC compartment.

The authors thank the Reviewer for their comment which is reflective of their valid points, above. Please refer to the above responses, particularly Reviewer 3, comments C-1, E-2.

In general, the authors thank the reviewer for their valuable insights and depth of review. Their suggestions have significantly contributed to the improvement of this work in both scope and impact. We look forward to their review of the revised manuscript.

References

1. Woodruff, M.C. et al. Extrafollicular B cell responses correlate with neutralizing antibodies and morbidity in COVID-19. *Nat Immunol* 21, 1506-1516 (2020).
2. Hoehn, K.B. et al. Cutting Edge: Distinct B Cell Repertoires Characterize Patients with Mild and Severe COVID-19. *J Immunol* (2021).
3. Jenks, S.A. et al. Distinct Effector B Cells Induced by Unregulated Toll-like Receptor 7 Contribute to Pathogenic Responses in Systemic Lupus Erythematosus. *Immunity* 49, 725-739.e726 (2018).
4. Xu, Chen, et al. "Prevalence and Characteristics of Rheumatoid-Associated Autoantibodies in Patients with COVID-19." *Journal of Inflammation Research* 14 (2021): 3123
5. Lingel, Holger, et al. "Unique autoantibody prevalence in long-term recovered SARS-CoV-2-infected individuals." *Journal of Autoimmunity* (2021): 102682.
6. Lerma, L. Angelica, et al. "Prevalence of autoantibody responses in acute coronavirus disease 2019 (COVID-19)." *Journal of translational autoimmunity* 3 (2020): 100073.
7. Hossri, Sami, et al. "Clinically significant anticardiolipin antibodies associated with COVID-19." *Journal of critical care* 59 (2020): 32-34.
8. Wang, Eric Y., et al. "Diverse functional autoantibodies in patients with COVID-19." *Nature* (2021): 1-6.

Reviewer Reports on the First Revision:

Referees' comments:

Referee #1 (Remarks to the Author):

In this revised manuscript from Woodruff et al., the data are very interesting and the key findings as I see them may be summarized as follows:

a) In patients with severe COVID-19, both viral antigen-specific and auto-reactive antibody secreting cells, largely of the IgG1 subclass and expressing low surface BCR levels are generated extra-follicularly. The authors seek to assume that viral antigen specific ASCs and auto-reactive ASCs in this pool are often different cells but firm evidence for such a contention does not exist.

b) Most of these antibody secreting cells have Igs with germline sequences; about half bind to epitopes on the three SARS-CoV-2 proteins tested, but they sometimes bind to these antigens with low affinity.

c) About 25% of all these ASC clones encode self-reactive antibodies -about half of the self-reactive clones are simultaneously capable of binding one of the three SARS-CoV-2 proteins assayed

d) About 30% of these ASCs have rearranged the known poly-reactive VH4-34 gene, and some of these VH4-34 Igs are self-reactive.

e) With clinical recovery, the IgG1 ASC pool of clones diminishes and auto-reactivity dissipates

Overall, I do find the data to be interesting and well documented. To my mind, however, these data reinforce the long-held view, that in severe infections, B cell tolerance is broken. This has been shown repeatedly in mice and humans but generally without the detailed analyses that this manuscript provides. These data nevertheless do not explain in a mechanistic way why or how tolerance is broken in an extra-follicular response in the context of a highly inflammatory milieu potentially and/or in the presence of Th1 cell expansion. My main concern is with the overall interpretation and what, to me, seems to be a slightly questionable statement made in that interpretational context.

It seems very well within the realm of possibility that almost every ASC clone, if not every ASC clone that was shown to be auto-reactive, may actually have been SARS-CoV-2 specific. After all, ONLY 3 of the 29 protein antigens of SARS-CoV-2 were utilized in serological assays yet about half of all the ASC clones made IgG1s that bound to these proteins. Perhaps the most parsimonious interpretation of the data is that most, if not all, the induced IgG1 ASC clones were specific for SARS-CoV-2 antigens (only three proteins were assayed for serologically) and they expand during severe viral infection; most are also intrinsically self-reactive.

So, while tolerance is broken, and I have no argument with that, I am not sure why this is NOT molecular mimicry. If anything, it argues strongly for normally quiescent self-reactive clones being

activated by numerous viral antigens in the inflammatory milieu of severe COVID-19. The specific autoantibodies induced may reflect cross-reactivity with SARS-CoV-2 epitopes.

On page 9 the authors argue against molecular mimicry stating:

“.....or can have no discernible affinity to any SARS-CoV-2 protein (Fig 4a, mAb 15)”. The word “any” was italicized by the authors. There is a need for more caution in making such a statement since only a fraction of the antigenic epitopes of the virus were interrogated.

Shiv Pillai

Referee #2 (Remarks to the Author):

The authors have done a commendable job in recruiting additional cohorts to extend their initial findings and in providing additional clinical, technical, and statistical details. The presence of a very similar profile of “autoreactivity” in ARDS patients with bacterial pneumonia is indeed very striking and suggests that development of these profiles is not pathogen-specific but potentially damage-driven and, as the authors indicate, could have important clinical implications. There are, however, few remaining concerns, which are outlined below:

1) While the authors have extended their analysis to OUT-C patients, several of the panels in Fig. 1 do not clearly evaluate the differences between OUT-C and ICU-C patients. For instance, are there statistical differences between OUT-C and ICU-C in Fig. 1F? Can the OUT-C patients be included in the analyses of Fig. 1L? Can a similar BASELine selection analysis to that shown in Fig. 1K be conducted in OUT-C patients? Addition of these comparisons would strengthen the authors’ claims that their findings are a hallmark of severe COVID-19.

2) Supplementary Fig 3a comparing differences in IgG1 and IgG2-specific ANA reactivity suggests a trend but does not achieve statistical significance (if the p value provided is correct). Thus, at present, these data do not strongly support the authors’ claims. Could this analysis be expanded to additional samples?

Author Rebuttals to First Revision:

Referees' comments:

Referee #1 (Remarks to the Author):

In this revised manuscript from Woodruff et al., the data are very interesting and the key findings as I see them may be summarized as follows:

a) In patients with severe COVID-19, both viral antigen-specific and auto-reactive antibody secreting cells, largely of the IgG1 subclass and expressing low surface BCR levels are generated extra-follicularly. The authors seek to assume that viral antigen specific ASCs and auto-reactive ASCs in this pool are often different cells but firm evidence for such a contention does not exist

b) Most of these antibody secreting cells have Igs with germline sequences; about half bind to epitopes on the three SARS-CoV-2 proteins tested, but they sometimes bind to these antigens with low affinity.

c) About 25% of all these ASC clones encode self-reactive antibodies -about half of the self-reactive clones are simultaneously capable of binding one of the three SARS-CoV-2 proteins assayed

d) About 30% of these ASCs have rearranged the known poly-reactive VH4-34 gene, and some of these VH4-34 Igs are self-reactive.

e) With clinical recovery, the IgG1 ASC pool of clones diminishes and auto-reactivity dissipates

Overall, I do find the data to be interesting and well documented. To my mind, however, these data reinforce the long-held view, that in severe infections, B cell tolerance is broken. This has been shown repeatedly in mice and humans but generally without the detailed analyses that this manuscript provides. These data nevertheless do not explain in a mechanistic way why or how tolerance is broken in an extra-follicular response in the context of a highly inflammatory milieu potentially and/or in the presence of Th1 cell expansion. My main concern is with the overall interpretation and what, to me, seems to be a slightly questionable statement made in that interpretational context.

The authors would like to thank Reviewer 1 for their continued effort on this manuscript. We agree with the points of interest that they have highlighted and are glad to hear that they continue to find the data interesting and well-documented. We agree that, while we have documented the nature of the tolerance breakdown seen in these patients in great detail, the molecular mechanisms and microenvironmental requirements driving EF-response bias and tolerance breakdown remains of great interest to the field. We believe that the data presented here will serve as an important guidepost for those ongoing studies.

We want to point out that while indeed, the triggering of transient serological autoreactivity has been described in several human infections, the cellular basis of this phenomenon has, to the best of our knowledge, never been previously addressed. In that regard, our work provides not only original findings but also a first mechanistic insight. Indeed, as shown by Nussenzweig and others, a large degree of autoreactivity and polyreactivity is present in both, the naive and the memory B cell compartment of healthy subjects but not in the pre-formed plasma cell compartment, thereby explaining the absence of serum autoantibodies through a memory-based late censoring checkpoint. Accordingly, the appearance of de novo serological autoreactivity could be due either to polyclonal bystander stimulation of autoreactive memory cells triggered by third signals released by severe inflammation (including TLR9/7 ligands; IFNs; IL-6 and IL-10 among others), or to newly activated naive B cells triggered in an antigen-specific fashion. While naive B cell expansions could also be enhanced by non-specific third signals (both T-dependent and T-independent), they typically need to be triggered also through the BCR. In all, our model provides strong evidence for the naive-derived mechanism and future studies will be needed to determine the contribution of several extra-follicular TH-like candidates, including ThP and TH10, with the latter population being more efficient to stimulate naive B cells.

While we agree that some clones are low affinity, it is important however to note that even low-affinity provided measurable binding in a dose-response fashion. In contrast, multiple previous studies of memory-derived high-affinity autoantibodies have documented loss of any detectable binding upon reversion to a germline configuration. Such studies have been used to postulate in several human autoimmune diseases that meaningful autoreactivity is generated de novo in the GC from non-autoreactive naive B cells. Our work clearly demonstrates that this is not necessarily the case as the low-affinity clones were clearly autoreactive and could potentially seed or form GC. More importantly, a significant fraction of the near-germline IgG1 clones, had considerable affinity (sub-nanomolar), and encoded autoantibodies highly specific for autoimmune diseases (such as GBM antibodies). These findings highlight the significance of the autoreactivity measured whose pathogenic potential is also highlighted by our previous demonstration of fast isotope switching and substantial hypermutation of naive-derived clones within days of the infection. These observations are also in keeping with several infectious mouse models (including salmonella infection dominated by extrafollicular responses with affinity maturation as in the work of Di Niro and Shlomchik). Of interest, this and other infections have the ability to either suppress or disrupt GC as shown for human severe COVID-19 by Kaneko et al.

It is important to note that our results provide the possibility of identifying subsets of patients in whom suppression of naive B cells could ameliorate autoimmune manifestations during acute COVID without the undesired consequences of universal B cell depletion. Such modulated interventions, such as belimumab which is known to regulate autoreactive naive B cells without impacting pre-formed memory could also be effective in preventing chronic autoimmunity (PASC?), in the appropriate patients.

It seems very well within the realm of possibility that almost every ASC clone, if not every ASC clone that was shown to be auto-reactive, may actually have been SARS-CoV-2 specific. After all, ONLY 3 of the 29 protein antigens of SARS-CoV-2 were utilized in serological assays yet about half of all the ASC clones made IgG1s that bound to these proteins. Perhaps the most parsimonious interpretation of the data is that most, if not all, the induced IgG1 ASC clones were specific for SARS-CoV-2 antigens (only three proteins were assayed for serologically) and they expand during severe viral infection; most are also intrinsically self-reactive.

The reviewer is correct. It is formally possible that most, if not all ASC clones we have identified may bind some component of SARS-CoV-2. This indeed, would nicely support the idea of BCR-induced primary B cell activation as opposed to memory expansion and validates our approach as it is likely that other studies demonstrating that a majority of highly mutated plasmablasts are infrequently SARS-CoV2-specific in acute COVID would reflect non-specific expansion of pre-formed memory cells. In the end, both the current extent of testing as well as the likelihood that additional testing might show an even larger frequency and overlap of viral and self-reactivity, demonstrate a striking level of cross-reactivity.

So, while tolerance is broken, and I have no argument with that, I am not sure why this is NOT molecular mimicry. If anything, it argues strongly for normally quiescent self-reactive clones being activated by numerous viral antigens in the inflammatory milieu of severe COVID-19. The specific autoantibodies induced may reflect cross-reactivity with SARS-CoV-2 epitopes.

As indicated above, the reviewer's point is well-taken. We agree that molecular mimicry could be a mechanism of cross-reactivity and now make this point more frontally in the manuscript. At its atomic definition, much of the cross reactivity that we identify could be explained by mirrored shared epitopes on viral and self-antigens although that would require multiple instances of molecular mimicry between several viral and multiple self-antigens. Our intent was to highlight the fact that autoreactivity in COVID-19 does not seem to stem from a single pathogenic protein driving autoreactive responses to a consensus self-antigen as has been previously described in rheumatic fever, Epstein-Barr, and other infectious disease commonly invoked under the molecular mimicry model. It does not seem to be the case, for example, that the spike protein of

SARS-CoV-2 is more or less capable of driving autoreactive responses than nucleocapsid, and neither seem to drive towards any particular autoreactive target.

In all, we would suggest that our read-out measures cross-reactivity and cannot differentiate between the two, which need not necessarily be synonymous. Indeed, cross-reactivity does many times reflect binding two different antigens that don't share an epitope (conformational or otherwise), through separate parts of the antigen-binding site with different contributions of the HCDR3, light chains and even frameworks (in particular FR1 for VH4-34 and FR3 for many other VH genes). This binding promiscuity would be more likely with the long and heavily charged HCDR3 characteristic of autoreactive B cells

We have now addressed this in the text by acknowledging the potential role and significance of molecular mimicry in the context of our interpretation of the data. We thank the reviewer for the comment and believe that the conclusions around these data are now clearer in the text.

On page 9 the authors argue against molecular mimicry stating:

“.....or can have no discernible affinity to any SARS-CoV-2 protein (Fig 4a, mAb 15)”. The word “any” was italicized by the authors. There is a need for more caution in making such a statement since only a fraction of the antigenic epitopes of the virus were interrogated.

This is correct – the language referenced was an overstatement of the available data. That sentence has been reworked to more accurately reflect the antigens that have been screened for reactivity.

Referee #2 (Remarks to the Author):

The authors have done a commendable job in recruiting additional cohorts to extend their initial findings and in providing additional clinical, technical, and statistical details. The presence of a very similar profile of “autoreactivity” in ARDS patients with bacterial pneumonia is indeed very striking and suggests that development of these profiles is not pathogen-specific but potentially damage-driven and, as the authors indicate, could have important clinical implications. There are, however, few remaining concerns, which are outlined below:

The authors thank reviewer 2 for their continued time and effort evaluating this manuscript. We agree – the inclusion of the ARDS data places an important context on the COVID-19 autoreactivity literature generated to-date, and holds important future implications both in scientific and therapeutic investigation.

1) While the authors have extended their analysis to OUT-C patients, several of the panels in Fig. 1 do not clearly evaluate the differences between OUT-C and ICU-C patients. For instance, are there statistical differences between OUT-C and ICU-C in Fig. 1F? Can the OUT-C patients be included in the analyses of Fig. 1L? Can a similar BASELINE selection analysis to that shown in Fig. 1K be conducted in OUT-C patients? Addition of these comparisons would strengthen the authors' claims that their findings are a hallmark of severe COVID-19.

We appreciate the reviewer's contribution to building the strength of the data presented. To answer the reviewer's question, there was no statistical difference between the OUT-C and ICU-C group in figure 1f. While we believe that these documented phenomena are indeed highly emphasized in the ICU patient group, we also believe that an intermediate phenotype (at least in IgG1 skewing) is possible in more mild manifestations of disease and have used language in the results to that effect. At the reviewer's suggestion, the OUT-C group has now been added to Fig 1l (now Extended data 2c), showing a highly similar connectivity between IgG1 and IgM as healthy controls. Unfortunately, yet highly informative, the main difference for IgG1 ASC between the two groups is the very low numbers found of low-mutation cells in the OUT cohort, thereby making a BASELINE comparison between the groups not robust or informative.

2) Supplementary Fig 3a comparing differences in IgG1 and IgG2-specific ANA reactivity suggests a trend but does not achieve statistical significance (if the p value provided is correct). Thus, at present, these data do not strongly support the authors' claims. Could this analysis be expanded to additional samples?

We appreciate the reviewer's perspective on these data and regret that it was poorly presented in the previous manuscript. While a paired-t test seemed appropriate, it poorly reflects the non-linear nature of OD values – particularly when the values being compared are below the detectable range as is the case in the IgG2 group. Indeed, the value of the data is gained in a much simpler presentation – in 5 patients with positive serological ANAs by ELISA, all 5 displayed ANA reactivity in the IgG1 compartment significantly above the assay background (3x SD of the mean background value). None of the five showed displayed reactivity in the IgG2 compartment above assay background. The figure has been altered to reflect that fact and aid interpretation.

Reviewer Reports on the Second Revision:

Referees' comments:

Referee #1 (Remarks to the Author):

No further comments

Shiv Pillai

Referee #2 (Remarks to the Author):

The authors have answered my previous concerns. There are no additional comments.

[Redacted]

[Redacted]

[Redacted]

[Redacted]

[Redacted]

[Redacted]

[Redacted]

[Redacted]

[Redacted]